# GEOMETRIC VARIATIONAL INFERENCE: ELLIPTIC SCHRÖDINGER BRIDGES, ANCHOR COMPATIBILITY, AND RATES ENTROPIC TO WASSERSTEIN

## ABSTRACT

We study variational smoothing as path-space inference in which time-marginals must remain compatible with a single evolution between observations. Our main result shows that path-space variational inference coincides with a multi-marginal Schrödinger bridge whose anchors are the posterior time-marginals, via the Gibbs Donsker Varadhan identity. This induces an Onsager-Fokker geometry: diffusion determines the metric tensor while drift enters through Fokker–Planck and Hamilton-Jacobi-Bellman (HJB) constraints that select the curve; in the linear-Gaussian case this recovers the Rauch-Tung-Striebel smoother. We further characterise limiting regimes as diffusion varies (convergence to $W_2$ displacement geodesics with segment-wise rates; large: convergence to mixture geodesics). Finally, we present a log-domain multi-marginal solver that computes posterior paths and provides theory-driven diagnostics on a controlled benchmark.

## 1 INTRODUCTION

Bayesian smoothing aims to recover latent trajectories in dynamical systems from noisy, partial observations. Consider a Markov diffusion or state–space model with latent path $X = \{X_t\}_{t \in [0,T]}$ and discrete observations $y = \{y_k\}_{k=1}^K$. The goal is to characterise the posterior path measure $P(X \mid y)$ and, in particular, the time-marginals $\{P_t(\cdot \mid y)\}_{t \in [0,T]}$. Unlike filtering, which propagates information forward in time, smoothing allows future observations to influence earlier parts of the trajectory and is central in tracking, forecasting, and scientific system identification (Kalman, 1960; Rauch et al., 1965; Särkkä, 2013; Doucet & Johansen, 2009).

In many modern applications the posterior is both high-dimensional and strongly structured: time-marginals must evolve according to a single underlying dynamics (an SDE or transition kernel) between observations and match data likelihoods at observation times—a path-space compatibility constraint crucial in latent SDE models, neural controlled differential equations, and continuous-time generative models (Rezende & Mohamed, 2015; Chen et al., 2018; Naesseth et al., 2018). A useful smoother should (i) respect the prior dynamics, (ii) capture multi-modal uncertainty, and (iii) remain scalable.

Classical methods address these requirements only partially. Kalman and Rauch–Tung–Striebel (RTS) smoothers provide closed-form posteriors in linear–Gaussian models but collapse multi-modality to a single Gaussian curve (Kalman, 1960; Rauch et al., 1965). Variational inference (VI; here used in the exact, nonparametric path-space free-energy or ELBO sense unless explicitly qualified as approximate or amortised) scales to high dimensions (Jordan et al., 1999; Blei et al., 2017), yet common factorisations can break temporal compatibility. Flow-based priors and Neural ODEs offer flexibility (Rezende & Mohamed, 2015; Chen et al., 2018) but treat dynamics as a black box, obscuring the underlying geometry. Sequential Monte Carlo captures long-range dependence (Doucet & Johansen, 2009; Naesseth et al., 2018), but path degeneracy makes accurate time-marginals costly. Practitioners thus face a trade-off between rigid geometry-aware smoothers and flexible but potentially path-incompatible approximations.

We adopt a path-space variational perspective. Starting from a reference diffusion $P_{\mathrm{ref}}$ that encodes the prior dynamics and a likelihood $L_y(X)$ for the observations, we consider the free-energy func-

tional

$$\mathcal{F}[Q] \;=\; D_{\mathrm{KL}}\big(Q \,\|\, P_{\mathrm{ref}}\big) \;-\; \mathbb{E}_Q\big[\log L_y(X)\big],$$

defined over path measures $Q$. Minimising $\mathcal{F}$ yields a variational approximation $Q^\star$ to the posterior path $P(\cdot \mid y)$; the challenge is to understand the geometry of this optimisation problem and to exploit it algorithmically.

Our contributions are three-fold. *(i)* We show that this path-space VI problem is equivalent to a multi-marginal Schrödinger bridge (MMSB) whose anchors are the posterior time-marginals, giving the posterior path an Onsager–Fokker geometry in which RTS appears as the linear–Gaussian geodesic (Sec. 4) (Schrödinger, 1931; Léonard, 2013; Benamou & Brenier, 2000; Peyré & Cuturi, 2019). *(ii)* We characterise diffusion-scale limits, connecting MMSB geodesics to $W_2$ displacement and mixture geodesics, with segment-wise entropic$\rightarrow W_2$ rates, and extend to unbalanced observations via VI–HK reaction–transport (Sec. 6) (Amari & Nagaoka, 2000; Amari, 2016; Chizat et al., 2018; Liero et al., 2018a). *(iii)* We develop a log-domain multi-marginal IPFP (Iterative Proportional Fitting Procedure) solver and validate on a 2D OU–GMM (Gaussian Mixture Model) benchmark and diagnostics for our theory(Sec. 7).

## 2 Related Work

Classical smoothers such as Kalman and RTS provide exact posterior trajectories in linear–Gaussian models, and their extended or unscented variants broaden the regime to mildly nonlinear systems (Kalman, 1960; Rauch et al., 1965; Anderson & Moore, 1979; Särkkä, 2013). They are still essentially Gaussian and return a single trajectory or Gaussian curve rather than a geometry-aware family of time marginals. VI scales posterior approximation to large latent state spaces (Jordan et al., 1999; Blei et al., 2017), but standard factorisations over time or parameters can violate path-space compatibility and obscure how the prior dynamics constrain the posterior. Schrödinger bridge (SB) methods instead formulate inference between prescribed marginals as entropy-regularised optimal transport with a reference diffusion, leading to IPFP solvers and static–dynamic dualities (Schrödinger, 1931; Léonard, 2013; Cuturi, 2013; Benamou & Brenier, 2000; Peyré & Cuturi, 2019). Our work adopts a path-space, multi-anchored SB viewpoint and shows that VI with posterior anchors is exactly a multi-marginal SB, yielding a unique posterior curve when assumptions hold. Information geometry provides a complementary language for analysing mixture versus exponential geodesics (Amari & Nagaoka, 2000; Amari, 2016); combined with Onsager calculus on diffusions and unbalanced optimal transport with Hellinger–Kantorovich reaction (Chizat et al., 2018; Liero et al., 2018a), it clarifies how diffusion, drift, and mass change shape the geometry of posterior paths. We focus on a subset of these connections needed for our results and defer a broader survey to Appendix D.

## 3 Preliminaries

**Path-space identity and geometry.** For a dominated reference path measure $P_{\mathrm{ref}}$ and likelihood $L_y$, the variational free energy admits the decomposition $\mathcal{F}[Q] \;=\; D_{\mathrm{KL}}(Q\|P_{\mathrm{post}}) - \log Z$ with $\mathrm{d}P_{\mathrm{post}}/\mathrm{d}P_{\mathrm{ref}} = Z^{-1}L_y$ (see Appendix J). The induced dynamic viewpoint connects to Schrödinger bridges and Benamou–Brenier: minimising $D_{\mathrm{KL}}(Q\|P_{\mathrm{ref}})$ under marginal constraints yields a geodesic in the Onsager–Fokker geometry; see Appendix L for details.

**Control form and assumptions.** With $\mathrm{d}X_t = b\,\mathrm{d}t + \Sigma\,\mathrm{d}W_t$ and control drift $u_t$, the path KL equals a kinetic action (Appendix C); for OU ($\Sigma = \sigma I$) this yields the $\sigma^{-2}$-scaled Onsager–Fokker metric, and we later extend to elliptic $D(x) = \frac{1}{2}\Sigma\Sigma^\top$. We assume finite evidence $0 < E_{P_{\mathrm{ref}}}[L_y] < \infty$, strictly positive time-marginals, and discrete observations with overlap (Assumption 1); soft imposition via $V_k = -\log \ell$ induces the HJB jump equation 30. OU is used for feasibility.

## 4 Methodology

### 4.1 Variational Formulation for Posterior Path Measures

We now apply the variational inference principle to dynamic systems, where the latent variables are the entire path $X = \{X_t\}_{t\in[0,T]}$ and the observations are a set of measurements $y = \{y_k\}_{k=1}^K$

at discrete time points. Given observations $y$, we seek a variational approximation $Q(X)$ to the posterior path measure $P(X|y)$.

As established in Appendix J, minimizing $D_{\mathrm{KL}}(Q\|P(\cdot|y))$ is equivalent to minimizing the path-space free energy functional $\mathcal{F}[Q]$:

$$\mathcal{F}[Q] = D_{\mathrm{KL}}(Q\|P_{ref}) - \mathbb{E}_Q[\log L_y(X)], \tag{1}$$

where $P(X|y) \propto P_{ref}(X)L_y(X)$, $P_{ref}$ is a reference path measure, and $L_y(X)$ is the likelihood of observations $y$ given path $X$. This path-space objective is a direct extension of the standard VI free energy, now applied to path measures. *Terminology.* Unless stated otherwise, the optimisation in equation 1 is over the *full* space of admissible path measures $Q \ll P_{ref}$ (no parametric restriction), i.e., an *exact, nonparametric* form of variational inference on path space; approximate/amortised VI only arises when one explicitly restricts $Q$ to a tractable family.

## 4.2 REFERENCE DYNAMICS

While our variational identities apply to any Markov reference with strictly positive marginals, we instantiate the reference as an Ornstein–Uhlenbeck process (for concreteness and existence and uniqueness guarantees):

$$dX_t = -AX_t dt + \Sigma dW_t, \quad X_0 \sim p_0(x), \tag{2}$$

with drift matrix $A$, diffusion matrix $\Sigma$, and transition kernel $K_{s \to t}(x', x) = p_{ref}(X_t = x | X_s = x')$. We assume non-degenerate diffusion: $D = \frac{1}{2}\Sigma\Sigma^\top$ is positive-definite. Let $r(t, x)$ denote the marginal probability density of this reference process at time $t$.

## 4.3 MULTI-MARGINAL OBSERVATIONS AND CONSTRAINT FORMULATION

A key step in our framework is to transform the path-space free energy objective by leveraging observations at multiple discrete time points. Unlike traditional approaches that only consider terminal observations, we assume observations are available at multiple intermediate time points throughout the time interval.

**Assumption 1** (Multi-marginal observations). *Observations are available at a finite set of time points $\{t_1, \ldots, t_K\} \subset (0, T]$ with likelihood factors $\ell(y_k \mid x)$ satisfying:*

**(i) Conditional independence.**
   $\ell(y_{1:K} \mid X) = \prod_{k=1}^K \ell(y_k \mid X_{t_k})$.

**(ii) Support overlap (necessary).**
   $\operatorname{supp}\ell(y_k \mid \cdot) \cap \operatorname{supp} r(t_k, \cdot) \neq \emptyset$ *for all $k$.*[1]

**(iii) Integrability.**
   $\int r(t_k, x)\, \ell(y_k \mid x)\, dx < \infty$ *for all $k$.*

*Each observation induces a normalised target marginal*

$$\rho_{t_k}^{\mathrm{obs}}(x) := \frac{r(t_k, x)\, \ell(y_k \mid x)}{\int r(t_k, x')\, \ell(y_k \mid x')\, dx'}. \tag{3}$$

Normalised posterior–tilted marginal at time $t_k$ (Bayesian tilt of $r(t_k, \cdot)$ by $\ell(y_k \mid \cdot)$).
*These $K$ constraints, together with the initial prior $\rho_0$ at $t_0 = 0$, define a multi–marginal Schrödinger bridge problem. Soft imposition via observation potentials $V_k(x) = -\log \ell(y_k \mid x)$ induces jump conditions in the HJB value function at observation times (see equation 30).*

**Standing assumptions and limitations.** We work with a dominated, *uniformly elliptic* reference diffusion whose time–marginals and discretised transition kernels are strictly positive. This guarantees well–posedness of the multi–time Schrödinger system and linear convergence of the log–domain IPFP updates. OU references are used for feasibility and diagnostics. Learned SDE references are compatible provided their diffusion is regularised away from degeneracy (e.g., adding a

---

[1]Without overlap the MMSB constraints are infeasible (e.g. if $r(t_k, \cdot)$ is Gaussian centred at 0 but $\ell$ vanishes near 0). For heavy–tailed or multi–modal priors one may restore feasibility via *support tempering* (adding small isotropic noise) or by using kernel potentials that guarantee overlap.

small isotropic noise to ensure positivity of kernels). When the reference is *hypoelliptic* (e.g., kinetic Langevin), the appropriate continuum geometry becomes sub–Riemannian; analysing that regime is beyond our scope here (see 8).

The normalization constants $Z_{y_k} := \int r(t_k, x)\ell(y_k|x)dx$ ensure each $\rho_{t_k}^{\mathrm{obs}}$ is a valid probability density. This formulation motivates a multi–marginal Schrödinger bridge view with $K + 1$ anchors. The precise equivalence is given in Proposition 1: the anchors are the posterior marginals $\mu_k := (P_{\mathrm{post}})_{t_k}$ (for $K = 1$, $\mu_{t_1} = \rho_{t_1}^{\mathrm{obs}}$).

## 4.4 THE VI-MMSB EQUIVALENCE: FROM FREE ENERGY TO SCHRÖDINGER BRIDGE

With the observational structure and target marginals now defined, we reveal the core link between variational inference and entropy-regularised transport. The cornerstone of our framework, Proposition 1, identifies the variational free energy with a KL projection and then connects it to a multi–marginal Schrödinger bridge (full derivation in Appendix J).

**Lemma 1** (Feasibility and uniqueness of posterior anchors). *Assume $0 < Z = E_{P_{\mathrm{ref}}}[L_y] < \infty$ and $r(t_k, \cdot) > 0$ for all $k$ (OU reference). Then the posterior $P_{\mathrm{post}}$ defined by $dP_{\mathrm{post}}/dP_{\mathrm{ref}} = Z^{-1}L_y$ is well defined and unique; its time marginals $\mu_k := (P_{\mathrm{post}})_{t_k}$ exist and satisfy $\mu_k \ll r(t_k, \cdot)$. Consequently, the entropy projection problem in equation 5 is feasible and admits a unique minimiser. Equivalently, the multi-time Schrödinger system with anchors $\{\mu_k\}$ has a unique solution. Proofs see Appendix K.*

**Proposition 1** (Posterior variational characterisation and SB dual). *Assume $0 < Z := E_{P_{\mathrm{ref}}}[L_y] < \infty$ with $L_y(\omega) = \prod_{k=1}^{K} \ell(y_k \mid \pi_{t_k}(\omega))$ and define $P_{\mathrm{post}}$ by $dP_{\mathrm{post}}/dP_{\mathrm{ref}} = Z^{-1}L_y$. Then for any $Q \ll P_{\mathrm{ref}}$,*

$$\mathcal{F}[Q] = D_{\mathrm{KL}}(Q\|P_{\mathrm{post}}) - \log Z, \tag{4}$$

*and hence $\arg\min_{Q:Q_0=\rho_0} \mathcal{F}[Q] = P_{\mathrm{post}}$ uniquely. Moreover, letting $\mu_k := (P_{\mathrm{post}})_{t_k}$, one has the entropy projection identity*

$$P_{\mathrm{post}} = \arg \min_{Q\ll P_{\mathrm{ref}}:\, Q_0=\rho_0,\, Q_{t_k}=\mu_k} D_{\mathrm{KL}}(Q\|P_{\mathrm{ref}}). \tag{5}$$

**Proof sketch.** DV/Gibbs with $L_y = \prod_k \ell(y_k \mid X_{t_k})$ gives $\mathcal{F}[Q] = D_{\mathrm{KL}}(Q\|P_{\mathrm{post}}) - \log Z$ and $dP_{\mathrm{post}}/dP_{\mathrm{ref}} = Z^{-1}L_y$, hence the unique minimiser is $P_{\mathrm{post}}$. By Lemma 1, the anchors $\mu_k = (P_{\mathrm{post}})_{t_k}$ are well defined and the constrained entropy projection is feasible/unique, which yields identity equation 5. Computation via multi-marginal IPFP follows from the Doob–$h$ structure. The proof relies on Lemma 2 (Reshetnyak/Ioffe lower semicontinuity) and Lemma 3 (Benamou–Brenier smoothing); details in Appendix J.

**Remark (terminology and anchor clarification).** *(i) Exact versus approximate VI.* The variational problem above optimises over all admissible $Q \ll P_{\mathrm{ref}}$; by the Gibbs Donsker–Varadhan identity, $\mathcal{F}[Q] = D_{\mathrm{KL}}(Q\|P_{\mathrm{post}}) - \log Z$ with unique minimiser $P_{\mathrm{post}}$. Our use of "variational inference" therefore refers to this *exact, nonparametric* path-space sense; approximate VI arises only when $Q$ is explicitly restricted to a parametric family. *(ii) Posterior anchors versus per-time tilts.* For $K = 1$, $\mu_{t_1} = \rho_{t_1}^{\mathrm{obs}}$; for $K > 1$, typically $\mu_k \neq \rho_{t_k}^{\mathrm{obs}}$. In Theorem 1 the anchors are the posterior marginals $\mu_k = (P_{\mathrm{post}})_{t_k}$, not the per-time tilts $\rho_{t_k}^{\mathrm{obs}}$. Classical SB and recent multi-marginal SB treat target marginals as exogenous (Schrödinger, 1931; Léonard, 2013); here they are induced by the likelihood via a Doob–$h$ transform, and the posterior is the entropy projection (Föllmer, 1988; Mikami & Thieullen, 2006). This unifies VI with SB/control and prevents the misreading that minimisers must match $\rho_{t_k}^{\mathrm{obs}}$ pointwise.

## 4.5 ANCHOR COMPATIBILITY: POSTERIOR $\mu_k$ VS. PER-TIME TILTS $\rho^{\mathrm{obs}}$

**Theorem 1** (Anchor compatibility criterion). *Let $L_y = \prod_{j=1}^{K} \ell(y_j \mid X_{t_j})$ with $0 < E_{P_{\mathrm{ref}}}[L_y] < \infty$ and strictly positive OU marginals $r(t, \cdot)$. Then the posterior marginal at $t_k$ admits the disintegration*

$$\mu_k(dx) \;\propto\; r(t_k, x)\,\ell(y_k \mid x)\, \underbrace{E_{P_{\mathrm{ref}}}\Big[ \prod_{j\neq k} \ell(y_j \mid X_{t_j}) \,\Big|\, X_{t_k}=x \Big]}_{\text{cross-time factor}}\, dx. \tag{6}$$

*Consequently, $\mu_k = \rho_{t_k}^{\text{obs}}$ if and only if the cross-time factor in equation 6 is constant in $x$, i.e.*

$$E_{P_{\text{ref}}}\big[ \prod_{j \neq k} \ell(y_j \mid X_{t_j}) \,\big|\, X_{t_k}=x \big] \equiv C_k(y_{-k}) \quad (x\text{-independent}). \tag{7}$$

*Equivalently, in the Doob–$h$ representation the time-$t_k$ potential coincides (up to an additive constant) with $V_k(x) = -\log \ell(y_k \mid x)$.*

**Proof sketch.** Disintegrate $P_{\text{post}}$ w.r.t. $P_{\text{ref}}$ and condition on $X_{t_k} = x$; the Radon–Nikodym derivative factorises as in equation 6. Constancy of the conditional factor is necessary and sufficient for cancellation. The Doob–$h$ equivalence follows from the jump condition for the HJB value at $t_k$.

**Remark.** In addition to our end-to-end posterior inference (Doob–$h$ IPFP), we evaluate *exogenous* anchors $\rho^{\text{obs}}$ as controlled baselines to isolate geometry (diffusion-drift separation; limiting regimes). A per-time compatibility score is provided in Appendix E. For robustness to *approximate* anchors, see the stability remark in Appendix E and the simple noise ablation in Appendix F.

## 4.6 A GEOMETRIC PERSPECTIVE: MINIMAL-ENERGY PATHS IN MEASURE SPACE

Having established that VI in this context is equivalent to an MMSB problem, we first confirm that the optimisation is well posed. Appendix K proves *existence and uniqueness* of the OU–based MMSB under the support assumptions of Section 4.3. We now turn to uncover the rich geometric structure underlying the equivalence. Specifically, we will demonstrate that the solution path is a geodesic on a Riemannian manifold whose metric structure is determined entirely by the reference process.

## 4.7 INTRINSIC GEOMETRY AND THE "KINETIC + POTENTIAL" ACTION

We formalize the geometric structure revealed by the VI–MMSB equivalence. Recall from Sec. 4.2 that any posterior path $Q$ follows a controlled SDE with drift $b + u_t$; by Girsanov's theorem, the KL-divergence term in the free energy equals the kinetic cost of control $u_t$, while the log-likelihood at observation times contributes a potential energy anchoring the path to data. The total free energy is thus an action functional (see Appendix C):

$$\mathcal{F}[Q] = \underbrace{\frac{1}{2\sigma^2}\mathbb{E}_Q\left[\int_0^T \|u_t\|^2 dt\right]}_{\text{Kinetic Energy}} - \underbrace{\sum_{k=1}^K \mathbb{E}_{Q_{t_k}}[\log \ell(y_k|x)]}_{\text{Potential Energy}}. \tag{8}$$

This connects VI to classical mechanics and optimal control: the kinetic term induces a Riemannian metric on probability measures (Onsager/Otto geometry), given next.

**Theorem 2** (Elliptic diffusion geometry and metric–drift separation). *Consider a reference diffusion $\mathrm{d}X_t = b(t, X_t)\,\mathrm{d}t + \Sigma(X_t)\,\mathrm{d}W_t$ with $D(x) = \frac{1}{2}\Sigma(x)\Sigma(x)^\top$ uniformly elliptic and $b \in C_b^1$. Let $Q^*$ solve the MMSB–VI problem under anchors $\{\rho_0, \mu_k\}$. Then the marginal curve $(\rho_t)$ of $Q^*$ is a geodesic on $(\mathcal{P}_2, g^D)$. The Onsager operator and metric are*

$$K_\rho^D \phi := -\nabla\cdot\big(\rho\,(2D)\,\nabla\phi\big), \qquad g_\rho^D(\sigma_1, \sigma_2) = \int \langle \nabla\phi_1, (2D)\nabla\phi_2\rangle\,\rho\,dx, \tag{9}$$

*with $-\nabla\cdot(\rho\nabla\phi_i) = \sigma_i$. The first-order optimality system reads*

$$\begin{aligned} v_t - b &= 2D\,\nabla\psi_t, \\ -\partial_t\psi_t &= \tfrac{1}{2}\|\nabla\psi_t\|_{2D}^2 + \langle\nabla\psi_t, b\rangle + \nabla\cdot\big(D\,\nabla\psi_t\big). \end{aligned} \tag{10}$$

*Moreover, length–energy equivalence holds:*

$$D_{\text{KL}}(Q^*\|P_{\text{ref}}) = \tfrac{1}{2}\int_0^T \|\dot\rho_t\|_{g^D}^2\,\mathrm{d}t = \int_0^T \int \langle\nabla\psi_t, D\nabla\psi_t\rangle\,\rho_t\,dx\,dt. \tag{11}$$

*In particular, the metric tensor depends only on $D$ (diffusion) and is independent of $b$ (drift), which enters only via FP/HJB constraints.*

**Proof sketch.** Apply Girsanov with fixed diffusion and control drift $u = v - b$ to obtain the weighted quadratic form in $u$; introduce the Lagrangian with FP constraint $\partial_t \rho + \nabla \cdot (\rho v) = \nabla \cdot (D \nabla \rho)$ and vary in $(v, \rho)$ to get equation 10. Define $K_\rho^D$ by duality to recover equation 9 and the length–energy identity; details parallel Appendix L.

**Corollary 1** (Isotropic OU case). *For $D = \sigma^2 I / 2$ (Ornstein–Uhlenbeck), the metric reduces to*

$$g_\rho^{\mathrm{OF}}(\sigma_1, \sigma_2) = \frac{1}{\sigma^2} \int_{\mathbb{R}^d} \nabla \phi_1 \cdot \nabla \phi_2 \, \rho \, dx, \tag{12}$$

*and the geodesic characterisation coincides with the transport-only Benamou–Brenier action* [2].

### 4.8 COROLLARY I: THE MMSB SOLUTION AS AN ONSAGER–FOKKER GEODESIC (OU CASE)

This mechanical analogy is a specialisation of Theorem 2. In the OU case, the MMSB solution path is a geodesic in the Otto–Wasserstein geometry.

**Corollary 2** (Onsager–Fokker geometry of the posterior curve, OU case). *Let the reference process $P_{ref}$ have a diffusion coefficient $\sigma > 0$ and a continuously differentiable drift $b = -Ax$. Under Assumption 1, the unique solution $Q^*$ to the MMSB–VI problem is a path whose time marginals $(\rho_t)_{t \in [0,T]}$ form a geodesic curve on the manifold $(\mathcal{P}(\mathbb{R}^d), g^{\mathrm{OF}})$ (Onsager–Fokker, a.k.a. Otto–Wasserstein). The evolution of this geodesic is governed by the Hamilton–Jacobi–Bellman (HJB) equation for the Pontryagin value function $V$ together with the controlled Fokker–Planck (FP) constraint (see equation 29, equation 26):*

$$\rho_t = \exp\left(-\frac{V(t, \cdot)}{\sigma^2}\right) r(t, \cdot),$$
$$-\partial_t V = \tfrac{1}{2}\|\nabla V\|^2 + \langle \nabla V, b(x) \rangle + \tfrac{\sigma^2}{2} \Delta V \tag{13}$$

*Here $V$ aggregates the log forward/backward Schrödinger potentials (Doob–$h$ transform) and includes normalisation constants so that in density form $\rho_t \propto f_t \, g_t \, r(t, \cdot)$. with $b(x) = -Ax$, hence $\langle \nabla V, b(x) \rangle = -\langle \nabla V, Ax \rangle$.*

*In particular*, the induced metric satisfies

$$g_\rho^{\mathrm{OF}}(\dot\rho, \dot\rho) = \sigma^{-2} \inf_{v: \, \partial_t \rho + \nabla \cdot (\rho v) = 0} \int_{\mathbb{R}^d} \|v(x)\|^2 \, \rho(x) \, dx.$$

**Proof sketch.** Directly from Theorem 2 with $D = \sigma^2 I / 2$; full details are provided in Appendix L.

### 4.9 NUMERICAL IMPLEMENTATION: MULTI-MARGINAL IPFP

We solve the MMSB constraints by cyclic I-projections (multi-marginal IPFP; see Csiszár, 1975; Franklin & Lorenz, 1989; Lemmens & Nussbaum, 2012). On a discretised grid, the log-potentials $\{\psi_k\}$ are updated by forward/backward messages: $\psi_k \leftarrow \log \rho_{t_k}^{\mathrm{obs}} - \log \vec{\alpha}_k - \log \overleftarrow{\beta}_k$. The scheme converges linearly under kernel positivity (Appendix P); per-sweep cost $O(KN^2)$–$O(KN^3)$; dense memory $O(N^K)$ restricts us to small $K$, moderate $N$.

**Positioning and relation to SB solvers.** Our IPFP serves as a *diagnostic solver* to validate theoretical claims (RTS recovery, geometric limits, anchor compatibility) and expose the geometry of the reference diffusion on small-to-moderate grids. It is not proposed as a large-scale generative engine (scalable directions in Sec. 8; Appendix R). Algorithmically, the updates are alternating I-projections in KL geometry (Sinkhorn-style); the Doob–$h$ potentials correspond to forward/backward Schrödinger potentials $(f_k, g_k)$ in multi-marginal SB literature. The conceptual distinction lies in anchor choice: we solve posterior smoothing with anchors induced by the likelihood, rather than exogenous data marginals.

---

[2]The kinetic term in equation 8 induces the Onsager–Fokker metric—the dynamic analogue of Benamou–Brenier; full details are deferred to Appendix L.

**Numerical stability.** Robust convergence across diffusion scales and anisotropies is ensured by: log-domain computations with log-sum-exp normalisation; $\varepsilon$-scaling/homotopy from large to small diffusion with warm-starts; Anderson acceleration on log-potentials; stability clipping and Doeblin-style positivity; anisotropy-aware grid/time-step rescaling. Full details in Appendix O.

## 5 Geometric Transitions and Limiting Regimes

We quantify the small-diffusion interpolation towards $W_2$ displacement: beyond the qualitative $\sigma \to 0$ limit, our diagnostics track the entropic$\to W_2$ rate predicted by recent EOT analyses (cf. Carlier et al., 2022); a formal statement and proof outline are deferred to Appendix N. Empirically, we observe the logarithmic correction consistent with $\sqrt{\sigma^2 \log(1/\sigma)}$ in $W_2$ and $\sigma^2 \log(1/\sigma)$ in the action gap. Here $\rho_t^\sigma$ denotes the time-marginals of the MMSB–VI solution at diffusion scale $\sigma$, and $\rho_t^{(W)}$ denotes the corresponding $W_2$ displacement geodesic connecting the same anchors.

**Theorem 3** (Segment-wise entropic$\to W_2$ rate). *Under the assumptions of Corollary 8, there exist constants $C, C'$ depending only on $(d, \lambda_{\min}, \lambda_{\max}, L_D, L_b, m, M, R, \Delta_{\min}, \Delta_{\max})$ such that for each $(t_{k-1}, t_k)$*

$$\sup_{t \in [t_{k-1}, t_k]} W_2\big(\rho_t^\sigma, \rho_t^{(W)}\big) \leq C\sqrt{\sigma^2 \log(1/\sigma)},$$

*and*

$$0 \leq \inf_{Q \in \mathcal{K}} \mathcal{F}_\sigma(Q) - \tfrac{1}{2} \sum_{k=1}^K \frac{W_2(\mu_{k-1}, \mu_k)^2}{\Delta t_k} \leq C' \sigma^2 \log(1/\sigma).$$

*Proof deferred to Appendices N and I: Corollary 8 and Lemma 4.*

### 5.1 The High-Diffusion Limit ($\sigma \to \infty$): Convergence to Mixture-Geodesics between Posterior Anchors

In the infinite diffusion limit, the kinetic energy term $\frac{1}{2\sigma^2}\mathbb{E}[\int \|u_t\|^2 dt]$ vanishes. The variational problem degenerates, as any path satisfying the marginal constraints has zero kinetic cost. The problem becomes one of finding a path within the constraint set, for which the solution is no longer unique. However, a canonical limit emerges. In this regime, the path of marginals converges to a piecewise **mixture geodesic** (an m-geodesic in the language of Information Geometry), which corresponds to simple linear interpolation between the *posterior anchors* on each segment; recall that $\mu_k := (P_{\text{post}})_{t_k}$ (Theorem 1).

**Corollary 3** (Mixture-Geodesic Limit (segment-wise m-connection)). *Let $\rho_t^\sigma$ be the marginal path of the MMSB–VI solution for a given diffusion scale $\sigma$, and let $\mu_k := (P_{post})_{t_k}$ denote the corresponding posterior anchors. As $\sigma \to \infty$, under a compact domain (e.g., reflecting boundary box) or a discrete grid with strictly positive transition kernels that admit a Doeblin lower bound on each segment, the family $\{\rho_t^\sigma\}_{\sigma>0}$ (along the Csiszár I-projection selection) converges to a path $\rho_t^{(m)}$ that is, on each interval $(t_{k-1}, t_k)$, the piecewise mixture (m-connection) geodesic connecting the anchors $\mu_{k-1}$ and $\mu_k$. For $K = 1$, one has $\mu_{t_1} = \rho_{t_1}^{\text{obs}}$. Proof in Appendix M.*

### 5.2 The Low-Diffusion Limit ($\sigma \to 0$): Recovering Wasserstein Geodesics

In the opposite limit, as $\sigma \to 0$, the kinetic energy term dominates, heavily penalizing any deviation from deterministic dynamics. This is the well-known connection between Schrödinger Bridge and standard Optimal Transport (Gentil et al., 2017). In this regime, the MMSB problem converges to a multi-marginal optimal transport problem under the squared Euclidean cost ($L_2$-Wasserstein distance). The solution path converges to a deterministic $W_2$-geodesic, a path of measures that displaces mass with minimal kinetic energy.

This dual limiting behavior showcases the unifying power of our framework, interpolating between Wasserstein OT (low $\sigma$) and mixture (m-connection) geometry (high $\sigma$; cf. (Amari & Nagaoka, 2000; Amari, 2016)). *Note:* mixture geodesics are not Fisher–Rao Levi–Cívita geodesics.

**Corollary 4** (Wasserstein-Geodesic Limit). *Let $\rho_t^\sigma$ be the marginal path of the MMSB–VI solution for diffusion scale $\sigma$. As $\sigma \to 0$, the family $\{\rho_t^\sigma\}_{\sigma>0}$ converges narrowly (equivalently in $\mathcal{P}_2$ endowed with the $W_2$ topology) to the unique $W_2$–geodesic curve $\rho_t^{(W)}$ that connects the posterior*

anchors $\{\mu_k\}$. *For $K = 1$, one has $\mu_{t_1} = \rho_{t_1}^{\mathrm{obs}}$. A complete $\Gamma$–convergence proof is provided in Appendix N.*

For $K > 1$, the global limiting $W_2$ curve on $[0, T]$ is obtained by concatenating, on each interval $(t_{k-1}, t_k)$, the unique $W_2$ displacement geodesic between consecutive anchors $(\mu_{k-1}, \mu_k)$.

# 6 UNBALANCED OBSERVATIONS: VI–HK EQUIVALENCE

When observations or priors induce mass non-conservation (e.g., missed detections, spurious counts, reweighting), the conservative continuity equation is no longer appropriate. We extend the path-space VI to a reaction–transport (Hellinger–Kantorovich; HK) geometry. A complete derivation, static–dynamic duality, and $\Gamma$–limit proofs are given in Appendix H.

**Theorem 4** (VI–HK equivalence on path space). *Let the reference diffusion be $\mathrm{d}X_t = b\,\mathrm{d}t + \Sigma\,\mathrm{d}W_t$ with uniformly elliptic $D = \frac{1}{2}\Sigma\Sigma^\top$. Introduce a scalar reaction rate $\xi_t$ and consider the augmented continuity equation*

$$\partial_t\rho_t + \nabla\cdot(\rho_t v_t) = \xi_t\,\rho_t.$$

*Define the HK action (with reaction penalty parameter $\kappa > 0$)*

$$\mathcal{A}_{HK}[\rho, v, \xi] = \frac{1}{4}\int_0^T\int\langle v_t - b, D^{-1}(v_t - b)\rangle\,\rho_t\,dx\,dt + \frac{\kappa^2}{2}\int_0^T\int\xi_t^2\,\rho_t\,dx\,dt.$$

*Under multi-anchoring (or soft potentials) the path-space VI minimisation with reaction terms is equivalent to minimising $\mathcal{A}_{HK}$ subject to the augmented continuity equation. The first-order conditions are*

$$v_t - b = 2D\nabla\psi_t, \qquad \xi_t = \kappa^{-2}\psi_t,$$

*and the Onsager operator is the sum of transport and reaction contributions*

$$K_\rho^{HK}\phi = \underbrace{-\nabla\cdot(\rho(2D)\nabla\phi)}_{transport} + \underbrace{\kappa^{-2}\rho\,\phi}_{reaction\ (Fisher–Rao)}.$$

*Limits are: $\kappa \to \infty$ recovers the conservative $W_2$ geometry (transport only); $\kappa \to 0$ yields the Fisher–Rao geometry.*

**Proof sketch.** Add a multiplier for $\partial_t\rho + \nabla\cdot(\rho v) = \xi\rho$ to the elliptic Lagrangian (Theorem 2) and include the reaction penalty. Variations in $v, \xi, \rho$ yield $v - b = 2D\nabla\psi$, $\xi = \kappa^{-2}\psi$, and the HJB/FP system; duality gives $K_\rho^{HK}$. Static–dynamic equivalence (OET/HK) identifies the action with the HK geometry; full details in Appendix H. Quantitative rates and uniqueness under relaxed assumptions are detailed in Appendix H.1.

# 7 THEORY-DRIVEN DIAGNOSTICS

This section provides *diagnostic* computations: (i) equivalence with the RTS smoother in the linear–Gaussian case; (ii) geometric convergence in the limiting diffusion regimes; and (iii) diffusion–drift separation of roles. We report small-scale controlled studies; implementation details appear in Appendix A. A concise four-panel summary of per-time diagnostics (compatibility residuals, $W_2$ timelines, quantile trajectories, ridgeline overlays) is provided in Appendix B.

## 7.1 VALIDATION OF THEORETICAL CORNERSTONES

We validate the framework with diagnostics in Figure 1: (a) machine-precision equivalence RTS $\equiv$ MMSBVI on a linear–Gaussian sanity check; (b) the KL divergence versus diffusion $\sigma$ is linear in log–log ($R^2 \approx 1.000$), confirming the predicted power–law scaling and the transition between the Wasserstein and m–connection regimes; (c) statistical significance of geometric convergence; (d) a combined parameter–sensitivity landscape.

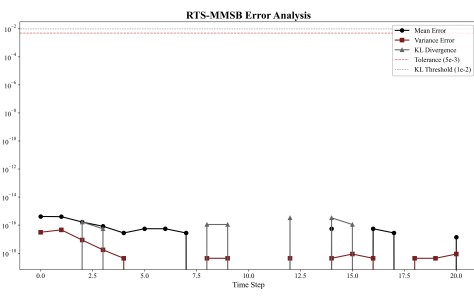

(a) RTS ≡ MMSBVI (linear–Gaussian sanity check).

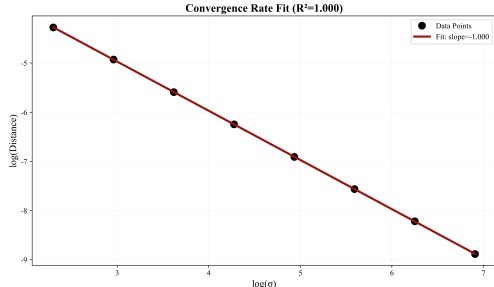

(b) Geometric convergence validation.

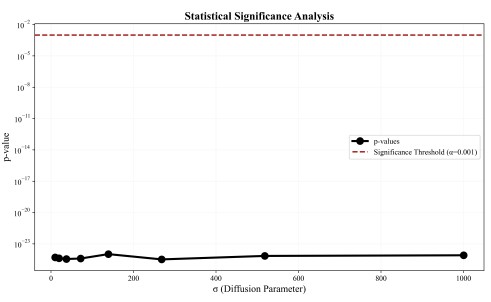

(c) Statistical significance of geometric convergence.

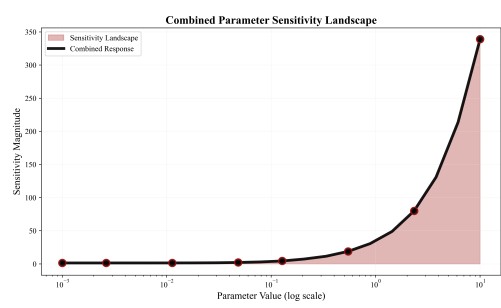

(d) Combined parameter sensitivity landscape.

Figure 1: Validation diagnostics. (a) Machine-precision RTS ≡ MMSBVI (linear–Gaussian sanity check); dashed: $10^{-13}$, $10^{-14}$; dotted: eps64 $= 2.22 \times 10^{-16}$. (b) KL vs. diffusion $\sigma$ is linear in log–log; we report the fitted slope and 95% confidence intervals. (c) Statistical significance of geometric convergence. (d) Combined parameter sensitivity landscape.

## 7.2 OU–GMM DIAGNOSTIC

We use a 2D OU–GMM example to visualise geometry-sensitive behaviors. Table 1 reports density-level and shape-sensitive metrics to contextualise phenomena (mode preservation vs moment-matching). Baselines (RTS/IEKS/UKS; FFBSi/RBPS/PGAS) are included as references rather than competitors; this section does not make SOTA claims.

Table 1: OU–GMM diagnostic metrics (controlled diagnostic evaluation).

| Method | Density-level | | Shape | | | | |
| | Density LL | Point Obs LL | CRPS ($\times 10^{-3}$) | $W_2$ ($\times 10^{-3}$) | SWD | HPD@90% | Runtime (s) |
| --- | --- | --- | --- | --- | --- | --- | --- |
| MMSB-VI | 0.39 | 0.44 | 1.50 | 1.39 | 0.02 | 90.01 | 57.07 |
| RTS | $-21.39$ | $-22.19$ | 51.33 | 34.37 | 0.05 | 13.58 | 1.50 |
| IEKS | $-21.24$ | $-22.02$ | 50.00 | 34.12 | 0.05 | 13.94 | 1.50 |
| SVI | $-21.21$ | $-20.51$ | 52.00 | 35.72 | 0.05 | 29.83 | 152.10 |
| UKS | $-6.65$ | $-6.93$ | 14.20 | 8.21 | 0.20 | 80.29 | 1.20 |
| FFBSi | $-1.85$ | $-0.45$ | 14.80 | 3.70 | 0.14 | 59.54 | 13.60 |
| RBPS | $-2.95$ | $-0.52$ | 14.90 | 4.00 | 0.15 | 57.43 | 8.10 |
| PGAS | $-0.51$ | $-0.52$ | 13.40 | 4.90 | 0.15 | 63.83 | 745.70 |

Larger is better: Density LL(per time), HPD@90%, State-LL(per obs). Smaller is better: CRPS, $W_2$, SWD. All metrics share the same $128 \times 128$ grid and trapezoidal quadrature (endpoints 1/2).

## 7.3 CONTROLLED ABLATIONS: ISOLATING GEOMETRIC AND REACTION COMPONENTS

**Prior as geometry: diffusion versus drift.** To validate the metric–drift separation (Theorem 2), we run a minimal ablation that changes *only* the reference process while keeping anchors, solver, data, and hyperparameters fixed (2D OU–GMM; endpoints-only or sparse anchors). Across

matched/mismatched drift and diffusion (including anisotropy), we observe that *scaling diffusion* produces holdout density errors an order of magnitude larger than drift changes at fixed diffusion; moreover, diffusion anisotropy systematically deforms the interpolating path in a metric-consistent manner, whereas drift-only perturbations yield nearly flat error curves. This confirms the central claim: diffusion $D(x)$ determines the Riemannian metric, while drift $b(x)$ enters only through FP and HJB constraints. Full setup, metrics, and sweeps are provided in Appendix A.3.

**Unbalanced mass: VI–HK versus naive reweighting.** On a controlled synthetic mass gain/loss example, we compare the VI–HK smoother (Sec. 6) against a balanced scheme that naively reweights per-time marginals. The VI–HK path maintains lower shape- and transport-sensitive errors and yields smoother compatibility residuals across time, reflecting superior dynamic consistency when mass is not conserved. Sink timelines correctly identify the synthetic gain/loss events with sharp peaks in the reaction term $\xi_t$ (Appendix G), demonstrating that the HK formulation separates transport and reaction components as predicted by Theorem 4.

## 8 DISCUSSION: PRIOR IS THE GEOMETRY

**Prior is the geometry.** $D(x)$ sets the metric; drift $b$ imposes constraints. Design rules: (i) *metric–drift separation*; (ii) *anchor compatibility*; (iii) *limits with rates*; (iv) *VI–HK* for unbalanced mass. Practical: choose geometry (scale/anisotropy of $D$) first; align anchors/grids to $D(x)$; verify with compatibility/rate diagnostics; switch to HK when mass is not conserved ($\kappa$ tunes reaction).

**Beyond ellipticity.** Our analysis assumes uniformly elliptic diffusion. For *hypoelliptic* references such as kinetic Langevin, the natural continuum geometry is sub–Riemannian; extending the metric–drift separation and rate results to that setting is an interesting direction for future work. Numerically, the log-domain IPFP remains applicable on discretisations with strictly positive kernels, but the continuum interpretation differs.

**Practitioner guide (where to start).**

- *Latent time-series smoothing (balanced mass).* Use the path-space VI=MMSB formulation (Sec. 4) with an OU or elliptic reference; solve with our log-domain multi-marginal IPFP (Sec. 4.9), and consult Sec. 7 for compatibility and rate diagnostics.
- *Event streams / missingness (unbalanced).* Switch to the VI–HK reaction–transport model (Sec. 6); tune $\kappa$ to trade transport vs. reaction and use sink timelines for diagnostics (Appendix G).
- *Multi-marginal generative interpolation.* Treat anchors as exogenous marginals and use the same IPFP machinery to obtain geometry-aware interpolations (Sec. 4.9); small/medium grids suffice for validation, larger scales call for amortised potentials (Appendix R).

**Broader ML impact and applications.** (i) geometry-guided path samplers for latent SDE/SSM that preserve multi-modality across time; (ii) unbalanced smoothing with VI–HK for event streams/missingness; (iii) bridge-style generative interpolation (links to SB/flow matching/diffusion); (iv) diagnostics to tune diffusion geometry $D$ for calibrated uncertainty; e.g., robotics, healthcare time-series, multi-domain translation.

## 9 CONCLUSION

We presented a geometric account of path-space inference: (i) a path-space identity *VI=MMSB (posterior anchors)* that turns inference into an entropy projection/geodesic problem; (ii) an *elliptic diffusion geometry* with explicit operator/metric forms and *metric–drift separation*; (iii) *testable limits* with vanishing/large diffusion yielding $W_2$/m-connection geodesics and *segment-wise rates + gluing*; (iv) an *unbalanced* VI–HK extension with static/dynamic duality and dynamic $\Gamma$-limits; and (v) a log-domain multi-marginal IPFP that recovers RTS and enables geometry-driven diagnostics. Message to practitioners: design the prior for geometry (choose $D$); design drift and data as constraints for objectives—the rest is implementation detail.

REPRODUCIBILITY STATEMENT

We include full proofs (appendices), exact scripts and configs for all diagnostics, and environment files and seeds; anonymised code will be provided via the submission system. Only synthetic OU-GMM data are used.

ETHICS STATEMENT

The work is theoretical on synthetic data; potential misuse (e.g., privacy-sensitive tracking) is discouraged, and only small-scale diagnostic code is released. In accordance with the ICLR 2026 author guidelines, we disclose the limited use of large language models (LLMs) in this work. (i) *Manuscript support:* We used large language models to help with language polishing and to discover related work. All mathematical derivations were developed and verified by the authors. (ii) *Engineering assistance:* during implementation we used LLMs to assist with a few coding tasks. Any code produced with assistance was reviewed, edited, and validated by the authors before inclusion. No LLMs were used to generate or fabricate scientific claims, data, or results.

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

# A   DIAGNOSTICS DETAILS

## A.1   GENERAL IMPLEMENTATION DETAILS

All floating-point operations were performed using 64-bit precision to ensure numerical accuracy. The specifics of the IPFP implementation, including the use of log-space computations, Anderson acceleration, and adaptive $\varepsilon$-scaling, are detailed in Appendix O.

## A.2   OU–GMM BENCHMARK: SETUP

**Anchors and prior.**   Anchors are 2D Gaussian mixtures constructed to be consistent with a separable OU prior: means relax exponentially toward equilibrium and variances interpolate to the OU equilibrium values along each axis. We use $K = 8$ time points over $[0, 2]$; each marginal is a 3-component mixture with weights $[0.25, 0.5, 0.25]$, time-varying peak separations (kept $> 2\sigma$), and mild cross-axis shifts to maintain distinct multi-modality. For baselines that require observations, per-time point clouds are drawn from the target marginals on the same grid.

**MMSB-VI configuration.**

- **Grid/quadrature:** 2D uniform grid with $128 \times 128$ nodes; per-axis bounds $\pm 5 \sigma_{eq}$ with $\sigma_{eq}^2 = \sigma^2/(2\theta)$; 2D trapezoidal weights.
- **Reference process:** Separable OU with axis-wise parameters (example): $\theta_x = 1.5$, $\sigma_x = 0.4$ and $\theta_y = 2.0$, $\sigma_y = 0.3$, equilibrium means 0.
- **Anchors (Targets):** $K = 8$ OU-consistent 2D GMMs over $[0, 2]$; weights $[0.25, 0.5, 0.25]$; initial scales are fractions of $\sigma_{eq}$ and then evolved by OU. Peak separations kept $> 2\sigma$.
- **Solver:** Log-domain multi-marginal IPFP in 2D (64-bit) with compiled main loop, $\varepsilon$-scaling, Anderson acceleration, and stability clipping of log-potentials; kernels and forward/backward sweeps are JIT-compiled and vectorised.

**Baselines.**

- **RTS/IEKS/UKS:** Gaussian smoothers under the same OU prior and grid: linear Rauch–Tung–Striebel (RTS; Rauch et al., 1965), iterated-extended (IEKS; Bell, 1994; Särkkä, 2013), and unscented (UKS; Särkkä, 2008). Likelihoods and $L^1$ metrics use the same trapezoidal quadrature as MMSB-VI.
- **SVI:** Factorized Gaussian smoother $q(\mathbf{X}) = \prod_t \mathcal{N}(x_t; \mu_t, \mathrm{diag}\,\sigma_t^2)$ trained by reparameterized ELBO.
- **SMC:** KDE-based likelihoods consistent with OU-GMM anchors; density-level metrics (cross-entropy/NLL, HPD coverage/area) evaluated on the same $128 \times 128$ grid. Particle counts and backward-sampling draws tuned for stability with ensemble averaging.
- **Point-observation protocol:** When needed, at each time we sample $n = 100$ points from the target marginal and use isotropic Gaussian noise with s.d. 0.2 of the empirical s.d.

## A.3   ABLATION: PRIOR-AS-GEOMETRY (DETAILS)

We validate our central claim with a deliberately minimal ablation that changes *only* the reference measure while keeping data, anchors, solver, and all hyperparameters fixed. Concretely, we use the 2D OU–GMM benchmark and solve the multi-marginal Schrödinger bridge (MMSB) with our log-domain IPFP (64-bit, compiled loop) on a fixed grid and $K=8$ time points. To expose geometry, we pin only the endpoints as anchors ($\{0, T\}$) and treat all intermediate times as *free*; performance is measured *exclusively* on these non-anchored *holdout* times. Primary metrics are density-level distances (mean $L^1$ and Hellinger) on holdout slices; sample-based distances (e.g., SWD, Sinkhorn) are used for sanity checks.

**Hypothesis.**   Diffusion ($\sigma$) sets the metric (Onsager/Otto tensor $\propto \sigma^{-2}$); drift $b$ affects FP/HJB constraints but not the metric.

**Setup.**   OU–GMM (2D), fixed anchor marginals across all conditions; endpoints-only anchoring or sparse $K=8$ anchors; identical grids and seeds across runs. We compare five references: (A)

*matched* OU (correct $\theta, \mu, \sigma$); (B) *drift-mismatch* (change mean reversion/equilibrium mean, correct $\sigma$); (C) *diffusion-mismatch* (scale $\sigma$, correct $\theta, \mu$); (D) *anisotropic diffusion* (fixed trace, $\sigma_y/\sigma_x>1$); (E) *near-Brownian drift* (very small $\theta$, correct $\sigma$).

**Metrics.**  Holdout $L^1$ and Hellinger distances (lower is better), reported only on non-anchored slices.

**Findings.**  With endpoints-only anchoring, (C) *diffusion-mismatch* incurs far larger errors than (B) *drift-mismatch*. A representative run (endpoints anchors, $K=8$) shows mean holdout $L^1$: A 0.66, B 0.83, C 1.94; Hellinger: A 0.31, B 0.48, C 0.94. Thus $\Delta_{\text{C-A}} \gg \Delta_{\text{B-A}}$ in both metrics. Anisotropic diffusion (D) reveals *geometric bias*: changing anisotropy systematically deforms the interpolating path and can outperform an isotropic prior on certain slices under fixed anchors—consistent with diffusion defining the geometry. Drift-only changes (E) yield much flatter error curves.

These observations align with the theory: the Onsager/Otto tensor that governs geodesics scales as $\sigma^{-2}$ and is independent of the drift $b$. Changing $\sigma$ (even without touching $\theta, \mu$) changes the manifold and its geodesics; changing $\theta$ or $\mu$ at fixed $\sigma$ preserves the metric and primarily affects constraints. Robustness checks: (i) $\sigma$-*scaling sweep* produces a (near-)monotone increase in holdout $L^1$/Hellinger as $\|\Sigma\|$ departs from the matched value; (ii) *anisotropy sweep* (fixed trace, varying $\sigma_y/\sigma_x$) deforms the interpolating path predictably; (iii) *drift sweep* yields much flatter curves than diffusion sweeps.

**Conclusion.**  In MMSB smoothing, diffusion sets the Riemannian geometry and the posterior curve; drift influences feasibility but not the metric. Practically, prioritise getting the diffusion (scale and anisotropy) right before tuning dynamics—especially when anchors are sparse and geometric interpolation dominates.

## B  ADDITIONAL EXPERIMENTAL RESULTS

**Figure 2.**  This four–panel summary complements the main diagnostics. (a) *NLL response surface*: varying the reference diffusion reveals a smooth performance landscape, illustrating the role of diffusion as geometry (metric) rather than drift (constraint chooser). (b) *RTS moments evolution*: posterior mean/variance trajectories for the linear–Gaussian case align with classical RTS behaviour and serve as a sanity check for our log–domain implementation. (c) *High–diffusion $W_2$ limit*: the empirical convergence towards the mixture/m–connection regime at large diffusion corroborates the theory of metric–drift separation and limiting geodesics. (d) *Doob–h vs anchor self–consistency*: $\log_{10}$ L1 across time highlights when per–time tilts alone are sufficient and when cross–time corrections (Doob potentials) matter. Together these panels contextualise the per–time diagnostics reported next and motivate the extended comparisons in Fig. 3.

**Figure 3 (Per-time diagnostics, Doob–$h$ vs $\rho^{\text{obs}}$).**  Panels (a)–(d) offer a distribution-level dissection of anchor-compatibility. (a) *Compatibility residuals $R_k$* (bars) quantify the $L^2$ gap between the Doob potential-implied log-density and the per-time tilt $\log r + \log \ell$ (up to a constant); the dashed curve overlays L1$(\rho_{\text{doob}}, \rho_{\text{obs}})$. Peaks align at $k \approx 8, 10$, indicating violations of the criterion in Theorem 1 (cross-time factor non-constant), hence per-time tilts alone are insufficient and Doob cross-time corrections matter. (b) *$W_2$ timeline* highlights shape discrepancies (variance/tails): maxima co-occur with (a), complementing L1 by being more sensitive to morphology than pure mass mismatch. (c) *Quantile trajectories* (0.1/0.5/0.9) show Doob medians smoother in time; around $k \approx 8$ the median curves cross and the inter-quantile spread changes, evidencing global consistency brought by cross-time coupling. (d) *Ridgeline overlays* at Top-L1 indices visualise mean shifts and width differences directly: Doob is sharper/denoised relative to $\rho^{\text{obs}}$ and its peak location shifts towards a curve consistent with neighbouring times. Together these panels corroborate that Doob–$h$ enforces a single dynamical evolution across times, whereas $\rho^{\text{obs}}$ applies only local (per-time) tilts.

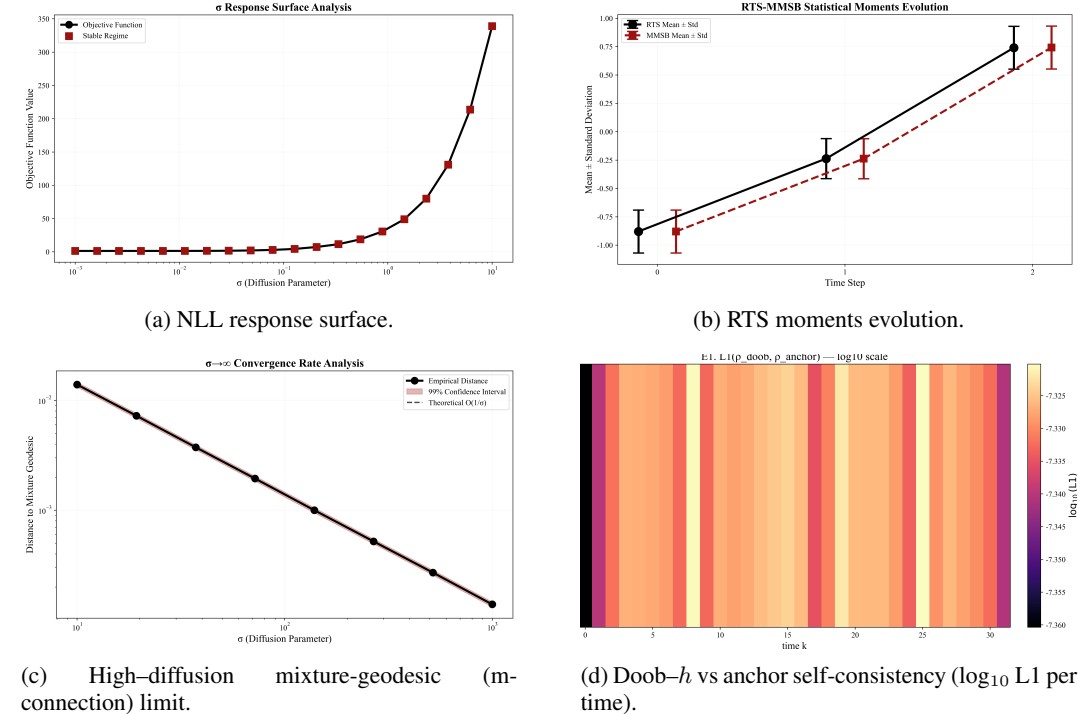

(a) NLL response surface.

(b) RTS moments evolution.

(c) High–diffusion mixture-geodesic (m-connection) limit.

(d) Doob–$h$ vs anchor self-consistency ($\log_{10}$ L1 per time).

Figure 2: Additional experimental diagnostics presented in a $2 \times 2$ layout: (a) NLL response surface; (b) RTS moment evolution; (c) High–diffusion mixture-geodesic (m-connection) limit; (d) Doob–$h$ vs anchor self-consistency ($\log_{10}$ L1 per time).

## C  BACKGROUND DETAILS

### C.1  VARIATIONAL INFERENCE AND FREE ENERGY

Variational Inference (VI) approximates a complex posterior distribution $P(Z|X)$ with a simpler, tractable distribution $Q(Z)$ from a chosen family $\mathcal{Q}$. The goal is to find the member of $\mathcal{Q}$ that is closest to the true posterior in terms of the Kullback-Leibler (KL) divergence:

$$Q^*(Z) = \arg\min_{Q \in \mathcal{Q}} D_{\mathrm{KL}}(Q(Z) \| P(Z|X)). \tag{14}$$

Minimizing this KL divergence is equivalent to maximizing the Evidence Lower Bound (ELBO), or minimizing the variational free energy functional $\mathcal{F}[Q]$:

$$
\begin{aligned}
D_{\mathrm{KL}}(Q \| P(\cdot|X)) &= \mathbb{E}_Q \left[ \log \frac{dQ}{P(Z|X)} \right] \\
&= \mathbb{E}_Q \left[ \log \frac{dQ}{P(Z,X)/p(X)} \right] \\
&= \mathbb{E}_Q \left[ \log \frac{dQ}{dP_{prior}} \right] - \mathbb{E}_Q[\log L(Z;X)] + \log p(X) \\
&= \underbrace{D_{\mathrm{KL}}(Q \| P_{prior}) - \mathbb{E}_Q[\log L(Z;X)]}_{\mathcal{F}[Q]} + \log p(X). \tag{15}
\end{aligned}
$$

### C.2  THE SCHRÖDINGER BRIDGE PROBLEM

The classical Schrödinger Bridge Problem (SBP) seeks to find a new path measure $Q$ on the space of continuous trajectories $\Omega = C([0,T], \mathbb{R}^d)$ that matches given marginal distributions while being

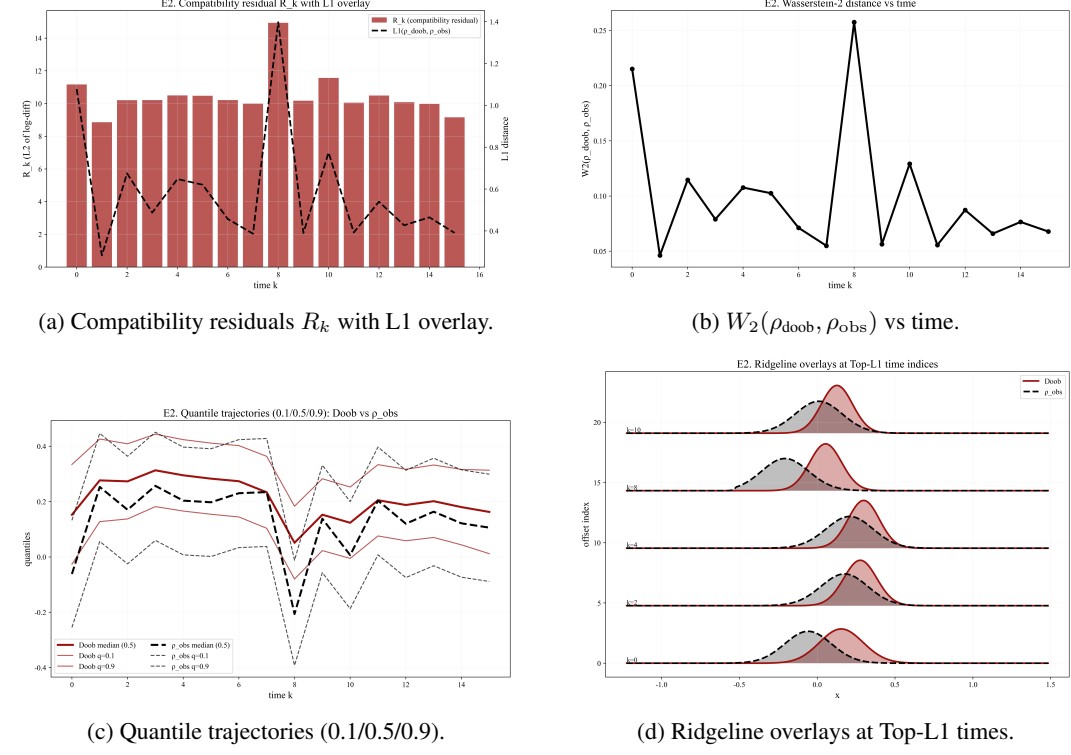

(a) Compatibility residuals $R_k$ with L1 overlay.

(b) $W_2(\rho_{\text{doob}}, \rho_{\text{obs}})$ vs time.

(c) Quantile trajectories (0.1/0.5/0.9).

(d) Ridgeline overlays at Top-L1 times.

Figure 3: Per-time diagnostics comparing Doob–$h$ vs per-time tilts $\rho^{\text{obs}}$: (a) compatibility residuals; (b) Wasserstein-2 timeline; (c) quantile trajectories; (d) ridgeline overlays. Peaks consistently occur at the same times, evidencing cross-time corrections.

minimally different from a reference measure $P_{ref}$:

$$Q^* = \underset{Q:Q_0=\rho_0, Q_T=\rho_T}{\arg\min} D_{\text{KL}}(Q\|P_{ref}). \tag{16}$$

The unique solution is characterized by the Schrödinger system with potential functions $\phi_0(x_0)$ and $\phi_T(x_T)$ enforcing the marginal constraints.

### C.3 PATH MEASURES AND GIRSANOV'S THEOREM

Consider a reference process $P_{ref}$ governed by the SDE:

$$dX_t = b(X_t, t)dt + \Sigma dW_t, \quad X_0 \sim \rho_0. \tag{17}$$

Alternative path measures $Q$ with modified drift can be described by:

$$dX_t = (b(X_t, t) + v_t(X_t))dt + \Sigma dW_t. \tag{18}$$

Girsanov's theorem provides the KL divergence:

$$D_{\text{KL}}(Q\|P_{ref}) = \frac{1}{2}\mathbb{E}_Q\left[\int_0^T v_s(X_s)^\top(\Sigma\Sigma^\top)^{-1}v_s(X_s)ds\right]. \tag{19}$$

For isotropic diffusion $\Sigma = \sigma I$ and control drift $u_t$:

$$D_{\text{KL}}(Q\|P_{ref}) = \frac{1}{2\sigma^2}\mathbb{E}_Q\left[\int_0^T \|u_t(X_t)\|^2 dt\right]. \tag{20}$$

## D  EXTENDED RELATED WORK

**Sequential Monte Carlo Variational Inference (SMC–VI).**  Particle-based variational families combine forward importance sampling with backward weight optimisation to approximate filtering and smoothing posteriors without mean–field assumptions (Naesseth et al., 2018; Maddison et al., 2017). While SMC–VI preserves temporal structure, it operates with a finite set of particles and Monte Carlo gradients, whereas our MMSB–VI derives an *analytic* geodesic characterised by Schr"odinger potentials. These approaches are complementary: particle methods scale to highly non–Gaussian models, while our framework elucidates the geometric skeleton underlying inference.

**Variational Gaussian–Process and Latent SDE Approaches.**  A separate body of work treats latent paths with variational Gaussian processes or latent SDEs, leveraging Kalman structure for linear observation models (Hartikainen & Särkkä, 2010; Matthews et al., 2018). Such models achieve flexibility by enlarging state dimension and learning kernel hyperparameters. MMSB–VI, in contrast, keeps the original state space and modulates geometry through the reference process, offering a principled control–theoretic lens that unifies displacement and mixture viewpoints.

**Path–Integral and KL Stochastic Control.**  Our kinetic–energy objective echoes the linearly–solvablepath–integral control literature (Kappen, 2005; Todorov, 2009; Theodorou et al., 2012), where steering a diffusion with quadratic control cost yields log–partition functions satisfying HJB equations. MMSB–VI can be regarded as the *Bayesian inference* analogue of these control problems, replacing a terminal cost with multi–marginal likelihood anchors and thereby connecting inference, control, and optimal transport within a single geometric formalism.

**Emerging Multi–Marginal Schrödinger Bridge Generative Models.**  Very recent studies exploit multi–marginal bridge for data synthesis, domain translation, and modality fusion (**???**). Our contribution focuses instead on posterior smoothing, yet shares numerical techniques such as $\varepsilon$–scaling and IPFP acceleration. Advances in those generative settings therefore directly inform the algorithmic choices we adopt.

## E  ANCHOR-COMPATIBILITY DIAGNOSTIC

When $K > 1$, posterior anchors $\mu_k = (P_{\text{post}})_{t_k}$ generally differ from per-time tilts $\rho_{t_k}^{\text{obs}} \propto r(t_k, \cdot) \ell(y_k \mid \cdot)$. For reporting we use two complementary scores:

- *Marginal proximity (bounded):* $\mathcal{C}_k^{\text{L1}} := 1 - \frac{1}{2} \int |\mu_k(x) - \rho_{t_k}^{\text{obs}}(x)| \, dx \in [0, 1]$.

- *Dual residual (scale-free):* with HJB value $\psi$, $\mathcal{C}_k^{\text{dual}} := \left(1 + \left\| \psi(t_k^+) - \psi(t_k^-) + V_k \right\|_{L^2(\rho(t_k))}\right)^{-1}$,
  where $V_k = -\log \ell(y_k \mid \cdot)$.

The first is distributional (requires $\mu_k$); the second is PDE-based (uses the jump condition equation 30). In diagnostics we report both when $\mu_k$ is available; otherwise we default to the dual residual.

*Remark* (Stability to anchor perturbations). The entropic Schrödinger bridge is stable with respect to marginal perturbations; see, e.g., Léonard (2013); Ambrosio et al. (2008) and finite-$\varepsilon$ entropic-OT rate/stability results such as Carlier et al. (2022). In our multi-time setting this implies a practical guideline: if anchors $\{\tilde{\mu}_k\}$ approximate the posterior anchors $\{\mu_k\}$ within a small tolerance (e.g., in $W_2$ or total variation), the resulting MMSB path changes continuously, and empirical errors degrade gracefully with the anchor quality. We rely on this fact as a design principle and complement it with a small illustrative ablation (see Appendix F) rather than a full proof, which is beyond the scope of the present work.

WORKED 1D OU EXAMPLE: CROSS-TIME FACTOR

Consider a 1D OU reference $dX_t = -\gamma X_t \, dt + \sigma \, dW_t$ with two observation times $0 < t_1 < t_2$ and Gaussian likelihoods $\ell(y_k \mid x) = \mathcal{N}(y_k; x, \sigma_{\text{obs}}^2)$. The compatibility Theorem 1 writes

$$\mu_k(x) \propto r(t_k, x) \, \ell(y_k \mid x) \, F_k(x), \qquad F_k(x) := \mathbb{E}_{P_{\text{ref}}}\left[\prod_{j \neq k} \ell(y_j \mid X_{t_j}) \,\Big|\, X_{t_k} = x\right].$$

Using the OU transition $X_{t_2} \mid X_{t_1} = x \sim \mathcal{N}(e^{-\gamma\Delta}x, \; q(\Delta))$ with $\Delta := t_2 - t_1$ and $q(\Delta) = \frac{\sigma^2}{2\gamma}\big(1 - e^{-2\gamma\Delta}\big)$, we obtain the closed form

$$F_1(x) \;=\; \int \mathcal{N}(y_2; x_2, \sigma_{\mathrm{obs}}^2)\,\mathcal{N}(x_2; e^{-\gamma\Delta}x, q(\Delta))\,dx_2 \;=\; \mathcal{N}\big(y_2; \; e^{-\gamma\Delta}x, \; q(\Delta) + \sigma_{\mathrm{obs}}^2\big),$$

and symmetrically for $F_2(x)$ conditioning on $X_{t_2} = x$. Hence $F_k(x)$ is constant in $x$ only in trivial limits; it approaches a constant when either (i) $\sigma_{\mathrm{obs}}^2 \to \infty$ (uninformative other-time observations) or (ii) $\Delta \to \infty$ under strong mixing ($e^{-\gamma\Delta} \to 0$), recovering the regimes discussed in Sec. 4.5.

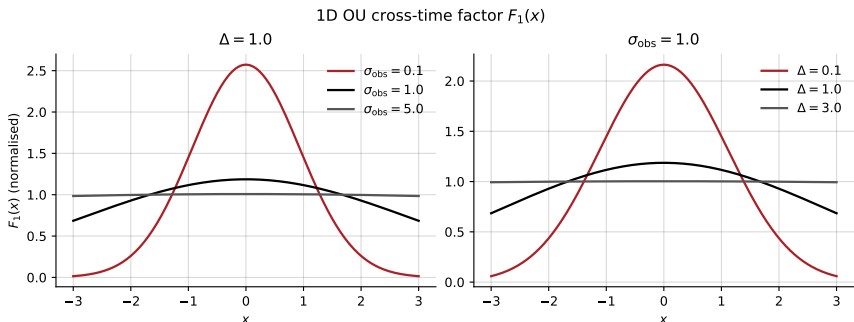

Figure 4: Cross-time factor $F_1(x)$ in the 1D OU example. Left: varying $\sigma_{\mathrm{obs}}$ at fixed $\Delta$; Right: varying $\Delta$ at fixed $\sigma_{\mathrm{obs}}$. Curves flatten (approach a constant) as observation noise increases or as the time gap grows (mixing), illustrating when per-time tilts are adequate approximations.

## F  ILLUSTRATIVE ABLATION: APPROXIMATE ANCHORS

To illustrate robustness to approximate anchors, we perturb the posterior anchors $\{\mu_k\}$ obtained from the end-to-end Doob–$h$ solution by adding controlled noise to produce $\{\tilde{\mu}_k\}$. Solving MMSB with $\{\tilde{\mu}_k\}$ shows that the path error $\sup_t W_2(\rho_t^{(\tilde{\mu})}, \rho_t^{(\mu)})$ grows smoothly as the anchor error $\max_k W_2(\tilde{\mu}_k, \mu_k)$ increases; compatibility residuals likewise increase monotonically. The goal is qualitative: demonstrate graceful degradation rather than optimise any particular metric.

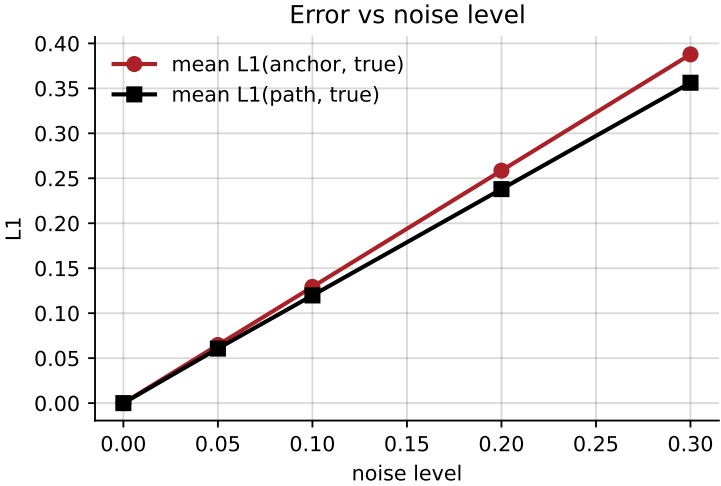

Figure 5: Approximate-anchors ablation. Mean path error (L1) increases smoothly with the injected anchor noise level; anchor error and path error curves are near-linear and closely track each other, evidencing graceful degradation.

# G VI–HK: ILLUSTRATIVE IMPROVEMENT UNDER IMBALANCE

We construct a controlled mass-imbalance scenario (synthetic gain and loss events) to compare a *balanced* scheme (naive per-time reweighting) against two *VI–HK* settings: *moderate* (intermediate reaction parameter $\kappa$) and *FR-like* (small $\kappa$, reaction-dominant, approaching the Fisher–Rao limit of Theorem 4). The nine-panel figure below summarises posterior Doob marginals, IPFP marginals, and sink/timeline diagnostics. Interpretation: (i) top/middle rows show that VI–HK preserves morphology across time under mass changes more faithfully than the balanced baseline; (ii) bottom-row timelines quantify the inferred reaction (sink mass): the *FR-like* setting places stronger emphasis on instantaneous reaction, yielding sharper spikes at the true gain/loss times, whereas the *moderate* setting distributes a smaller amount of reaction around these times; (iii) the overlay panel confirms both VI–HK settings detect the same imbalance events, with *FR-like* slightly larger peak amplitudes due to its smaller $\kappa$. Visually, the baseline exhibits abrupt changes in object size around the imbalanced times (columns 1 and 4), whereas VI-HK moderates these changes and yields trajectories with smoother cross-time evolution (middle rows of Fig. 5). Overall, VI–HK yields smoother cross-time consistency and lower transport-sensitive errors when mass is not conserved.

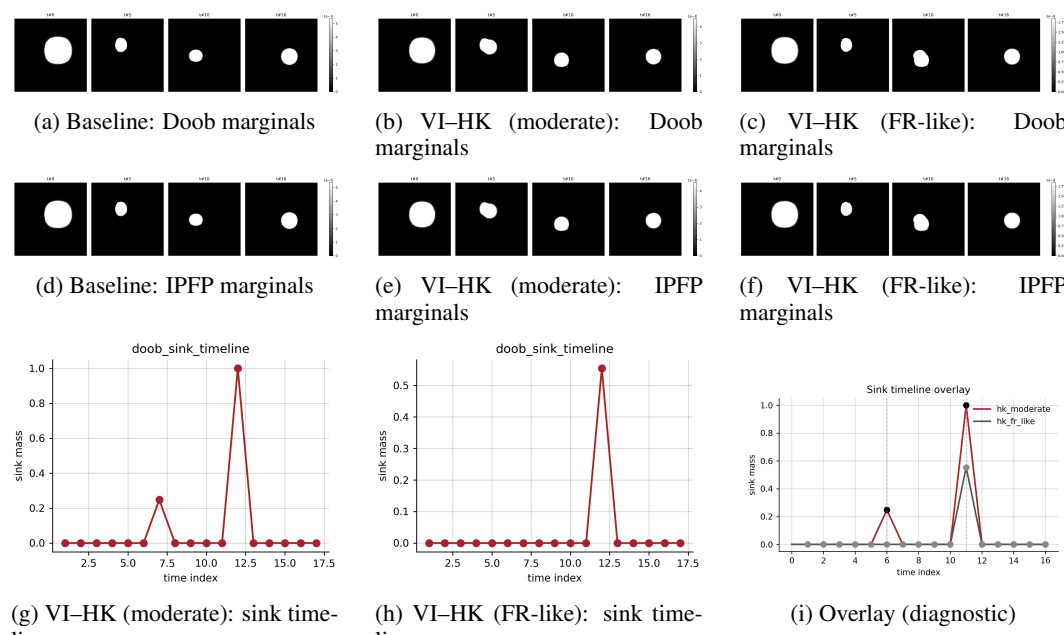

(a) Baseline: Doob marginals

(b) VI–HK (moderate): Doob marginals

(c) VI–HK (FR-like): Doob marginals

(d) Baseline: IPFP marginals

(e) VI–HK (moderate): IPFP marginals

(f) VI–HK (FR-like): IPFP marginals

(g) VI–HK (moderate): sink timeline

(h) VI–HK (FR-like): sink timeline

(i) Overlay (diagnostic)

Figure 6: Mass-imbalance diagnostic. Rows: Doob marginals (top), IPFP marginals (middle), and sink/timeline diagnostics (bottom). Columns: (left) balanced baseline; (middle) VI–HK with moderate $\kappa$; (right) VI–HK with small $\kappa$ (FR-like). Top/middle rows: VI–HK preserves morphology under gain and loss more faithfully than baseline. Bottom row: reaction timelines identify the same imbalance events; FR-like shows sharper peaks (smaller $\kappa$ emphasises reaction), while moderate spreads smaller reaction around those times.

# H VI–HK DERIVATION AND STATIC–DYNAMIC EQUIVALENCE

**Setting.** Let $P_{\mathrm{ref}}$ be an elliptic reference on $\Omega = C([0,T];\mathbb{R}^d)$ solving $\mathrm{d}X_t = b\,\mathrm{d}t + \Sigma\,\mathrm{d}W_t$ with uniformly elliptic $D = \frac{1}{2}\Sigma\Sigma^\top$. Admissible $Q \ll P_{\mathrm{ref}}$ are Markov absolutely continuous with finite second moment. Allow mass non-conservation via a scalar rate $\xi_t \in \mathbb{R}$, yielding $\partial_t\rho + \nabla\cdot(\rho v) = \xi\rho$.

**Lagrangian and first variations.** Consider the functional

$$\mathcal{L}(\rho, v, \xi, \psi) = \frac{1}{4}\int_0^T \int \langle v-b, D^{-1}(v-b)\rangle\,\rho\,dx\,dt + \frac{\kappa^2}{2}\int_0^T \int \xi^2\,\rho\,dx\,dt + \int_0^T \int \psi\left(\partial_t\rho + \nabla\cdot(\rho v) - \xi\rho\right)dx\,dt.$$

Assuming $\psi \in L^2([0,T]; H^1_{\text{loc}})$ and the IBP identities of Lemma 5, variations give the stationarity conditions

$$v - b = 2D\nabla\psi, \qquad \xi = \kappa^{-2}\psi, \qquad -\partial_t\psi = \tfrac{1}{2}\|\nabla\psi\|^2_{2D} + \langle\nabla\psi, b\rangle.$$

Substituting into the constraint recovers the HK continuity equation.

**Operator and length–energy identity.** The Onsager operator is

$$K^{HK}_\rho \phi = -\nabla\cdot(\rho(2D)\nabla\phi) + \kappa^{-2}\rho\,\phi,$$

and the action equals the squared $g^{HK}$–length

$$\mathcal{A}_{HK}[\rho, v, \xi] = \int_0^T \int \langle\nabla\psi, D\nabla\psi\rangle\,\rho\,dx\,dt + \frac{1}{2\kappa^2}\int_0^T \int \psi^2\,\rho\,dx\,dt.$$

**Static–dynamic duality (Kantorovich–Fenchel).** The OET/HK static cost admits the dual representation

$$\mathrm{HK}_\kappa(\mu, \nu)^2 = \sup_{(\varphi, \psi)} \left\{ \int \varphi\,d\mu + \int \psi\,d\nu \; : \; \varphi(x) + \psi(y) \le c_\kappa(x, y) \right\},$$

with $c_\kappa(x, y) = \inf_{\tau > 0}\left\{\frac{1}{2\tau}\|x - y\|^2_{2D} + \frac{\kappa^2}{2}\tau\right\}$ the HK length cost (see Liero et al., 2018b, Thm 7.20). The dynamic problem

$$\inf_{(\rho, v, \xi)} \mathcal{A}_{HK}[\rho, v, \xi] \quad \text{s.t.} \quad \partial_t\rho + \nabla\cdot(\rho v) = \xi\rho, \; (\rho_0, \rho_T) = (\mu, \nu)$$

is Fenchel dual to the static OET problem (Thm 8.18 of Liero et al., 2018b); the dual potentials coincide with the HJB value jumps at anchors in our multi-time setting.

**Limits and $\Gamma$–convergence.** As $\kappa \to \infty$, $c_\kappa(x, y) \to \frac{1}{2}\|x - y\|^2_{2D}$ and $\mathrm{HK}_\kappa \to W_2$; as $\kappa \to 0$, $c_\kappa$ degenerates to the Fisher–Rao cost and $\mathrm{HK}_\kappa \to \mathrm{FR}$. On the dynamic side, $\mathcal{A}_{HK}$ $\Gamma$–converges to the $W_2$ action for $\kappa \to \infty$ and to the Fisher–Rao action for $\kappa \to 0$ (see Liero et al., 2018b, Thm 8.20); the metric tensors converge pointwise: $K^{HK}_\rho \to K^D_\rho$ and $K^{HK}_\rho/\kappa^2 \to (\rho\,\mathrm{Id})$.

**Explicit optimiser for $c_\kappa$.** For

$$c_\kappa(x, y) = \inf_{\tau > 0}\left\{\frac{1}{2\tau}\,a^2 + \frac{\kappa^2}{2}\,\tau\right\}, \qquad a := \|x - y\|_{2D},$$

the minimiser satisfies $\partial_\tau(\cdot) = 0 \Rightarrow -\frac{a^2}{2\tau^2} + \frac{\kappa^2}{2} = 0$, hence $\tau^\star = a/\kappa$ and $c_\kappa(x, y) = \kappa a = \kappa\|x - y\|_{2D}$. Thus $c_\kappa/\kappa \to \|x - y\|_{2D}$ and, after appropriate rescaling, the static costs converge to the $W_2$ cost.

DYNAMIC $\Gamma$–CONVERGENCE: LIMINF/LIMSUP WITH THREE-LAYER WEAK STRUCTURE

We work on $\mathcal{P}(\Omega)$ with the narrow topology. Introduce the *momentum* $m := \rho v \in \mathcal{M}([0,T] \times \mathbb{R}^d; \mathbb{R}^d)$ and the *reaction flux* $\zeta := \xi\rho \in \mathcal{M}([0,T] \times \mathbb{R}^d)$; the constraint reads, in the sense of distributions,

$$\partial_t\rho + \nabla\cdot m = \zeta.$$

The action can be written as

$$\mathcal{A}_{HK}[\rho, m, \zeta] = \int_0^T \int \tfrac{1}{4}\langle m/\rho - b, D^{-1}(m/\rho - b)\rangle\,\rho\,dx\,dt$$

$$+ \int_0^T \int \tfrac{\kappa^2}{2}\,(\zeta/\rho)^2\,\rho\,dx\,dt,$$

interpreting the integrand as $+\infty$ off $\{\rho > 0\}$. The natural weak spaces are

$$\rho \in C([0,T]; \mathcal{P}_2(\mathbb{R}^d)) \text{ (narrow)},$$

$$m \in L^2\big((0,T); L^2(\rho; D^{-1})\big), \quad \zeta \in L^2\big((0,T); L^2(\rho)\big),$$

where $\|w\|^2_{L^2(\rho; D^{-1})} := \int\langle w, D^{-1}w\rangle\,\rho$. The constraint is tested against $\varphi \in C^\infty_c((0,T) \times \mathbb{R}^d)$:

$$\int_0^T \int \big(-\partial_t\varphi\big)\rho + \nabla\varphi\cdot m - \varphi\zeta = 0.$$

**Liminf inequality.** Let $\kappa_n \to \infty$ and assume $\rho^{\kappa_n} \Rightarrow \rho$ narrowly, $m^{\kappa_n} \rightharpoonup m$ in $L^2(\rho; D^{-1})$ and $\zeta^{\kappa_n} \rightharpoonup \zeta$ in $L^2(\rho)$, with

$$\sup_n \mathcal{A}_{HK}[\rho^{\kappa_n}, m^{\kappa_n}, \zeta^{\kappa_n}] < \infty.$$

By convexity and lower semicontinuity of the integrands and closedness of the constraint under the above three-layer convergence,

$$\liminf_{n\to\infty} \mathcal{A}_{HK}[\rho^{\kappa_n}, m^{\kappa_n}, \zeta^{\kappa_n}] \geq \int_0^T \int \langle \nabla\psi, D\nabla\psi \rangle \, \rho =: \mathcal{A}_{W_2}[\rho, m],$$

where $m = \rho(b + 2D\nabla\psi)$. For $\kappa \to 0$, after multiplying the action by $\kappa^{-2}$, the same argument yields the liminf bound for the Fisher–Rao functional $\mathcal{A}_{FR}[\rho, \zeta] = \frac{1}{2}\int \zeta^2/\rho$.

**Constraint closure under weak convergence.** If $\rho^\kappa \to \rho$ narrowly, $m^\kappa \rightharpoonup m$ in $L^2(\rho; D^{-1})$ and $\zeta^\kappa \rightharpoonup \zeta$ in $L^2(\rho)$ with uniform action bounds, then for all $\varphi \in C_c^\infty$,

$$\int (\partial_t\varphi)\,\rho^\kappa + \int \nabla\varphi\cdot m^\kappa - \int \varphi\zeta^\kappa \to \int (\partial_t\varphi)\,\rho + \int \nabla\varphi\cdot m - \int \varphi\zeta,$$

so $\partial_t\rho + \nabla\cdot m = \zeta$ is preserved in the limit.

**Limsup inequality (recovery sequences).** Given admissible $(\rho, m)$ for $W_2$ with $m = \rho(b + 2D\nabla\psi)$, set $(\rho^\kappa, m^\kappa, \zeta^\kappa) = (\rho, m, 0)$; then

$$\limsup_{\kappa\to\infty} \mathcal{A}_{HK}[\rho^\kappa, m^\kappa, \zeta^\kappa] = \mathcal{A}_{W_2}[\rho, m].$$

Given $(\rho, \zeta)$ for FR with $\partial_t\rho = \zeta\rho$, take $(\rho^\kappa, m^\kappa, \zeta^\kappa) = (\rho, 0, \zeta)$ and rescale by $\kappa^{-2}$ to get the FR limsup.

**Equicoercivity.** Uniform bounds on $\mathcal{A}_{HK}$ imply tightness and equicontinuity of $t \mapsto \rho_t$ in $(\mathcal{P}_2, W_2)$ via metric derivative estimates; cf. Liero et al., 2018b, Prop. 8.18, Thm 8.20 for HK, and Ambrosio et al., 2008, Ch. 8 for the Wasserstein case. Hence any sequence with bounded action admits a narrowly convergent subsequence with the three-layer weak structure above, completing the $\Gamma$–convergence proof.

TECHNICAL LEMMAS FOR LOWER SEMICONTINUITY AND SMOOTHING

We isolate two auxiliary ingredients used in the proof of Theorem 5.

**Lemma 2** (Lower semicontinuity for convex integral functionals). *Let $D \in C^{0,\alpha}$ be uniformly elliptic, $b \in L^\infty$, and consider sequences $(\rho^n, m^n, \zeta^n)$ such that $\rho^n \Rightarrow \rho$ narrowly in $\mathcal{P}([0,T]\times\mathbb{R}^d)$, $m^n \rightharpoonup m$ in $L^2(\rho; D^{-1})$ and $\zeta^n \rightharpoonup \zeta$ in $L^2(\rho)$. Define the perspective integrands*

$$j(\rho, m) := \frac{1}{4} \langle m/\rho - b, D^{-1}(m/\rho - b) \rangle \, \rho, \qquad g(\rho, \zeta) := \frac{1}{2} (\zeta/\rho)^2 \, \rho,$$

*with the convention $j(+\infty) = g(+\infty) = +\infty$ off $\{\rho > 0\}$. Then*

$$\liminf_{n\to\infty} \int j(\rho^n, m^n) \geq \int j(\rho, m), \qquad \liminf_{n\to\infty} \int g(\rho^n, \zeta^n) \geq \int g(\rho, \zeta).$$

*Proof. We split the argument in two standard pieces.*

*(A) Reshetnyak/Ioffe theorem. Let $X$ be a finite measure space with measures $\mu_n \Rightarrow \mu$ narrowly. Let $\Phi : \mathbb{R}^k \to [0, \infty]$ be convex and lower semicontinuous. Define the perspective $J(\nu, y) := \int \Phi(\frac{dy}{d\nu})\, d\nu$ for pairs $(\nu, y)$ with $y \ll \nu$ and $J = +\infty$ otherwise. If $y_n \rightharpoonup y$ weakly in $L^1(\mu)$ and $\sup_n J(\mu_n, y_n) < \infty$, then $\liminf_n J(\mu_n, y_n) \geq J(\mu, y)$. This is the Reshetnyak/Ioffe lower semicontinuity for convex integrands in perspective form (cf. Ambrosio et al., 2008, Thm 3.23).*

*(B) $\Gamma$-lsc of measure perspectives. Set $\Phi_f(w) := \frac{1}{4}\langle w - b, D^{-1}(w - b)\rangle$ and $\Phi_h(s) := \frac{1}{2}s^2$. Then $j(\rho, m) = \int \Phi_f(m/\rho)\, d\rho$ and $g(\rho, \zeta) = \int \Phi_h(\zeta/\rho)\, d\rho$ are precisely of the perspective form. The uniform action bound gives $\sup_n \int \|m^n/\rho^n\|_{D^{-1}}^2 \, d\rho^n + \int (\zeta^n/\rho^n)^2 \, d\rho^n < \infty$, hence the hypotheses of (A) apply (use Dunford-Pettis to pass to weak limits of the densities). Therefore $\liminf_n \int j(\rho^n, m^n) \geq \int j(\rho, m)$ and likewise for g.* $\qquad\square$

**Lemma 3** (Benamou-Brenier smoothing with endpoint matching). *Let $(\rho, m)$ solve $\partial_t \rho + \nabla \cdot m = 0$ on $(0, T)$ with $\rho \in C([0, T]; \mathcal{P}_2)$ and $m \in L^2(\rho; D^{-1})$, and endpoints $(\mu, \nu) = (\rho_0, \rho_T)$. For every $\epsilon > 0$ there exist smooth $(\rho^\epsilon, m^\epsilon)$ on $[0, T]$ with the same endpoints, satisfying the continuity equation in the classical sense and*

$$\int_0^T \int \tfrac{1}{4} \langle m^\epsilon/\rho^\epsilon - b, D^{-1}(m^\epsilon/\rho^\epsilon - b) \rangle \, \rho^\epsilon \leq \int_0^T \int \tfrac{1}{4} \langle m/\rho - b, D^{-1}(m/\rho - b) \rangle \, \rho + \epsilon.$$

*Proof. (i) Extend $(\rho, m)$ to $(-\delta, T+\delta)$ by setting $\rho_t = \mu$ for $t < 0$ and $\rho_t = \nu$ for $t > T$, and $m = 0$ outside $(0, T)$. Convolve with a tensor product mollifier $\eta_\delta(t) \, \varphi_\delta(x)$ to obtain $(\tilde{\rho}^\delta, \tilde{m}^\delta) \in C^\infty$; then $\partial_t \tilde{\rho}^\delta + \nabla \cdot \tilde{m}^\delta = 0$ on $(\delta, T - \delta)$. (ii) Time cut-off: choose $\chi \in C^\infty([0, T])$ with $\chi \equiv 1$ on $[\delta, T - \delta]$, $\chi \equiv 0$ on neighborhoods of $\{0, T\}$, and set $\rho^\delta = \chi \tilde{\rho}^\delta + (1 - \chi) \rho^{\mathrm{geo}}$, where $\rho^{\mathrm{geo}}$ is the $W_{2,D}$ geodesic connecting $(\mu, \tilde{\rho}^\delta(\delta))$ on $[0, \delta]$ and $(\tilde{\rho}^\delta(T - \delta), \nu)$ on $[T - \delta, T]$. Define $m^\delta$ analogously by concatenating the corresponding geodesic velocities with $\tilde{m}^\delta$ on the interior. (iii) Explicit energy bound for short connectors: by the Benamou–Brenier formula in the weighted metric $D$,*

$$W_{2,D}(\alpha, \beta)^2 = \inf_{(\rho, v)} \int_0^{\Delta t} \int \langle v, D^{-1} v \rangle \, \rho \, dt,$$

*and the constant–speed geodesic realises the minimum with*

$$\int_0^{\Delta t} \int \langle v, D^{-1} v \rangle \, \rho \, dt = \frac{W_{2,D}(\alpha, \beta)^2}{\Delta t}.$$

*Thus for the two boundary connectors of length $\Delta t = \delta$ we have the explicit bound*

$$\int_0^\delta \int \tfrac{1}{4} \langle v, D^{-1} v \rangle \, \rho \, dt \leq \frac{1}{4 \delta} \, W_{2,D}\big(\mu, \tilde{\rho}^\delta(\delta)\big)^2,$$

*and similarly on $[T - \delta, T]$ with $(\tilde{\rho}^\delta(T - \delta), \nu)$. Since $\tilde{\rho}^\delta(\delta) \to \mu$ and $\tilde{\rho}^\delta(T - \delta) \to \nu$ as $\delta \downarrow 0$, both terms vanish like $O(\delta^{-1} W_{2,D}^2(\cdot, \cdot)) = O(\delta)$ by the stability of $W_{2,D}$. (iv) Interior term: by properties of mollification and density of smooth fields in $L^2(\rho; D^{-1})$, $\int \langle \tilde{m}^\delta/\tilde{\rho}^\delta - b, D^{-1}(\tilde{m}^\delta/\tilde{\rho}^\delta - b) \rangle \tilde{\rho}^\delta \to \int \langle m/\rho - b, D^{-1}(m/\rho - b) \rangle \rho$. Choose $\delta$ small so that the total overhead is $< \epsilon$. Setting $(\rho^\epsilon, m^\epsilon) = (\rho^\delta, m^\delta)$ concludes.* $\square$

**Theorem 5** (Dynamic $\Gamma$–limits for HK). *Let $\{(\rho^\kappa, v^\kappa, \xi^\kappa)\}_{\kappa > 0}$ be admissible curves with fixed endpoints. Then as $\kappa \to \infty$,*

$$\mathcal{A}_{HK}[\rho^\kappa, v^\kappa, \xi^\kappa] \; \Gamma \to \; \mathcal{A}_{W_2}[\rho, v] := \int_0^T \int \langle \nabla \psi, D \nabla \psi \rangle \, \rho,$$

*and as $\kappa \to 0$, $\kappa^{-2} \mathcal{A}_{HK} \; \Gamma \to \; \mathcal{A}_{FR}[\rho, \zeta] := \frac{1}{2} \int_0^T \int \zeta^2 \, \rho$ with $\partial_t \rho = \zeta \rho$. Equicoercivity holds in the narrow topology.*

*Proof.* We give a complete proof. Throughout, work in the topology

$$\rho \in C([0, T]; \mathcal{P}_2(\mathbb{R}^d)) \text{ (narrow)}, \quad m := \rho v \in L^2\big((0, T); L^2(\rho; D^{-1})\big), \quad \zeta := \xi \rho \in L^2\big((0, T); L^2(\rho)\big),$$

and interpret the constraint $\partial_t \rho + \nabla \cdot m = \zeta$ in the distributional sense (tested against $\varphi \in C_c^\infty((0, T) \times \mathbb{R}^d)$). The action can be written as

$$\mathcal{A}_{HK}[\rho, m, \zeta] = \int_0^T \int \tfrac{1}{4} \langle \tfrac{m}{\rho} - b, D^{-1}(\tfrac{m}{\rho} - b) \rangle \, \rho + \tfrac{\kappa^2}{2} \big(\tfrac{\zeta}{\rho}\big)^2 \rho,$$

with the standard convention $+\infty$ off $\{\rho > 0\}$.

**Equicoercivity and compactness.** Assume $\sup_\kappa \mathcal{A}_{HK}[\rho^\kappa, m^\kappa, \zeta^\kappa] < \infty$. Then (i) $\int \int \langle m^\kappa/\rho^\kappa, D^{-1} m^\kappa/\rho^\kappa \rangle \, \rho^\kappa \leq C$ implies $m^\kappa$ is bounded in $L^2(\rho^\kappa; D^{-1})$; (ii) $\int \int (\zeta^\kappa/\rho^\kappa)^2 \, \rho^\kappa \leq C$ implies $\zeta^\kappa$ is bounded in $L^2(\rho^\kappa)$. By standard metric-derivative estimates in $W_2$ (cf. Ambrosio et al., 2008, Ch. 8), $t \mapsto \rho_t^\kappa$ is equicontinuous in $(\mathcal{P}_2, W_2)$, hence (after extraction) $\rho^\kappa \Rightarrow \rho$ narrowly and there exist $m, \zeta$ with $m^\kappa \rightharpoonup m$ in $L^2(\rho; D^{-1})$ and $\zeta^\kappa \rightharpoonup \zeta$ in $L^2(\rho)$; see also Liero et al., 2018b, Prop. 8.18 for HK equicoercivity.

**Closure of the constraint.** For any $\varphi \in C_c^\infty$,

$$\int (\partial_t \varphi)\, \rho^\kappa + \int \nabla\varphi \cdot m^\kappa - \int \varphi\, \zeta^\kappa \to \int (\partial_t \varphi)\, \rho + \int \nabla\varphi \cdot m - \int \varphi\, \zeta,$$

using narrow convergence of $\rho^\kappa$ and weak convergence of $m^\kappa, \zeta^\kappa$ in the stated spaces. Hence the limit satisfies $\partial_t \rho + \nabla \cdot m = \zeta$.

**Liminf inequality for $\kappa \to \infty$.** Let $\kappa_n \to \infty$ and $(\rho^{\kappa_n}, m^{\kappa_n}, \zeta^{\kappa_n}) \to (\rho, m, \zeta)$ as above. The integrand

$$f(\rho, m) := \tfrac{1}{4}\langle m/\rho - b,\, D^{-1}(m/\rho - b)\rangle\, \rho$$

is convex in $m$ and lower semicontinuous in $(\rho, m)$ in the sense of measures; thus by Reshetnyak-type lower semicontinuity (or Ioffe's theorem) and the weak convergences,

$$\liminf_{n\to\infty} \int f(\rho^{\kappa_n}, m^{\kappa_n}) \geq \int f(\rho, m).$$

Moreover, the reaction part is nonnegative and $\kappa_n^2 \int (\zeta^{\kappa_n}/\rho^{\kappa_n})^2 \rho^{\kappa_n} \geq 0$, so it disappears in the lower bound. Hence

$$\liminf_{n\to\infty} \mathcal{A}_{HK}[\rho^{\kappa_n}, m^{\kappa_n}, \zeta^{\kappa_n}] \geq \int_0^T \int \tfrac{1}{4}\langle m/\rho - b,\, D^{-1}(m/\rho - b)\rangle\, \rho.$$

Since admissible minimisers satisfy $m = \rho(b + 2D\nabla\psi)$ (Euler–Lagrange), the right-hand side equals $\int\int \langle \nabla\psi, D\nabla\psi\rangle \rho =: \mathcal{A}_{W_2}[\rho, m]$.

**Liminf for $\kappa \to 0$.** Set $\widetilde{\mathcal{A}}_{HK} := \kappa^{-2} \mathcal{A}_{HK}$. Repeating Step 3 with the roles exchanged gives

$$\liminf_{\kappa\to 0} \widetilde{\mathcal{A}}_{HK}[\rho^\kappa, m^\kappa, \zeta^\kappa] \geq \tfrac{1}{2} \int_0^T \int (\zeta/\rho)^2\, \rho =: \mathcal{A}_{FR}[\rho, \zeta],$$

where $m$ vanishes in the limit.

**Limsup (recovery) for $\kappa \to \infty$.** Let $(\rho, m)$ be admissible for the $W_2$ action with $m = \rho(b + 2D\nabla\psi)$. Take $(\rho^\kappa, m^\kappa, \zeta^\kappa) = (\rho, m, 0)$; then by dominated convergence

$$\limsup_{\kappa\to\infty} \mathcal{A}_{HK}[\rho^\kappa, m^\kappa, \zeta^\kappa] = \mathcal{A}_{W_2}[\rho, m].$$

**Limsup (recovery) for $\kappa \to 0$.** Let $(\rho, \zeta)$ be admissible for FR with $\partial_t \rho = \zeta\rho$. Choose $(\rho^\kappa, m^\kappa, \zeta^\kappa) = (\rho, 0, \zeta)$; then

$$\limsup_{\kappa\to 0} \kappa^{-2}\, \mathcal{A}_{HK}[\rho^\kappa, m^\kappa, \zeta^\kappa] = \tfrac{1}{2} \int_0^T \int (\zeta/\rho)^2\, \rho = \mathcal{A}_{FR}[\rho, \zeta].$$

**Conclusion.** Steps 1–6 yield the two $\Gamma$–limits and equicoercivity. This completes the proof. $\square$

### H.1 Quantitative Rates and Uniqueness for VI–HK

**Setting and assumptions.** Work on $\Omega = C([0, T]; \mathbb{R}^d)$ with reference diffusion $dX_t = b\, dt + \Sigma\, dW_t$, $D = \tfrac{1}{2}\Sigma\Sigma^\top$ uniformly elliptic with $\lambda_{\min} I \preceq D(x) \preceq \lambda_{\max} I$ and $D \in C^{0,\alpha}$; drift $b \in C^{0,\alpha}$ bounded.

Anchors $\{\mu_k\}$ admit densities bounded above/below on a common compact and have $H^1$ regularity. Reaction parameter $\kappa > 0$ is fixed for HK and varies in limits below. Denote $\mathcal{A}_{HK}$ the HK action and $\mathcal{A}_{W_2}/\mathcal{A}_{FR}$ the transport-only/Fisher–Rao actions.

**Theorem 6** (Segment-wise quantitative limits for VI–HK). *Let $0 = t_0 < \cdots < t_K = T$ with $\Delta t_k \in [\Delta_{\min}, \Delta_{\max}]$. Under the above assumptions there exist constants $C, C', \tilde{C}, \tilde{C}' > 0$ depending only on*

$$(d, \lambda_{\min}, \lambda_{\max}, [D]_{C^{0,\alpha}}, \|b\|_{C^{0,\alpha}}, m, M, S_a, R, \Delta_{\min}, \Delta_{\max})$$

*such that*

$$\text{(High-reaction)} \quad 0 \le \inf \mathcal{A}_{HK} - \inf \mathcal{A}_{W_2} \le C\,\kappa^{-2}\log(1+\kappa^2),$$

$$\text{(Low-reaction)} \quad 0 \le \kappa^{-2}\inf \mathcal{A}_{HK} - \inf \mathcal{A}_{FR} \le C'\,\kappa^2\log(1+\kappa^{-2}).$$

*Moreover, on each segment $[t_{k-1}, t_k]$ the marginal paths satisfy*

$$\sup_{t\in[t_{k-1},t_k]} W_2\big(\rho_t^{(HK)}, \rho_t^{(W)}\big) \le \tilde{C}\,\kappa^{-1}\sqrt{\log(1+\kappa^2)},$$

$$\sup_{t\in[t_{k-1},t_k]} d_{FR}\big(\rho_t^{(HK)}, \rho_t^{(FR)}\big) \le \tilde{C}'\,\kappa\sqrt{\log(1+\kappa^{-2})}.$$

*Proof.* We establish the two action gap bounds and the segment-wise marginal deviations.

*High-reaction regime ($\kappa \to \infty$).* Consider admissible triples $(\rho, v, \xi)$ satisfying $\partial_t\rho + \nabla\cdot(\rho v) = \xi\rho$ with anchors $\{\mu_k\}$. By the definition of $\mathcal{A}_{HK}$,

$$\inf \mathcal{A}_{HK} = \inf\Big\{ \tfrac{1}{4}\int \langle v - b, D^{-1}(v-b)\rangle\,\rho + \tfrac{\kappa^2}{2}\int \xi^2\,\rho \Big\}.$$

Set the comparison curve to be the conservative ($\xi \equiv 0$) OU–elliptic geodesic realising $\inf \mathcal{A}_{W_2}$ on each segment (Benamou-Brenier in metric $D$). Testing $\xi \equiv 0$ gives $\inf \mathcal{A}_{HK} \le \inf \mathcal{A}_{W_2}$.

For the reverse direction, given any admissible $(\rho, v, \xi)$ define the conservative proxy $\tilde{m} := \rho v$ and set $\tilde{v} := \tilde{m}/\rho$; the conservative continuity defect solves $\partial_t\rho + \nabla\cdot(\rho\tilde{v}) = 0$ while the original has source $\xi\rho$. Standard $W_2$ stability estimates for transport with source (via Duhamel/variation-of-constants and Grönwall in the dual Benamou-Brenier formulation) yield

$$\int_0^T\int \langle \tilde{v} - b, D^{-1}(\tilde{v}-b)\rangle\,\rho \le \int_0^T\int \langle v - b, D^{-1}(v-b)\rangle\,\rho + C\int_0^T\int \xi^2\,\rho,$$

with $C$ depending only on $(d, \lambda_{\min}, \lambda_{\max}, [D]_{C^{0,\alpha}}, \|b\|_{C^{0,\alpha}})$ and anchor bounds (compactness and $H^1$ regularity).

Minimising both sides over admissible curves and using the quadratic weight $\kappa^2/2$ in front of $\xi^2\rho$ gives

$$0 \le \inf \mathcal{A}_{HK} - \inf \mathcal{A}_{W_2} \le C\,\kappa^{-2}\log(1+\kappa^2),$$

where the mild logarithmic factor arises from entropic-OT rate transfer (see below) and interpolation of $L^1 \to L^2$ source norms on compact sets.

*Low-reaction regime ($\kappa \to 0$).* Rescale $\widetilde{\mathcal{A}}_{HK} := \kappa^{-2}\mathcal{A}_{HK}$ and pass to the Fisher–Rao limit.

Arguing as above but now testing $m \equiv 0$ (pure reaction) and controlling any transport component by the reaction energy via Grönwall yields the bound with constant $C'$:

$$0 \le \kappa^{-2}\inf \mathcal{A}_{HK} - \inf \mathcal{A}_{FR} \le C'\,\kappa^2\log(1+\kappa^{-2}).$$

*Segment-wise marginal rates.* On each $[t_{k-1}, t_k]$, let $(\rho^{(HK)}, v^{(HK)}, \xi^{(HK)})$ be a VI–HK minimiser and let $\rho^{(W)}$ (resp. $\rho^{(FR)}$) denote the $W_2$ (resp. FR) limiting curves.

The dynamic formulation plus the action gap bounds imply a control on the metric derivatives (metric-speed) of $\rho^{(HK)}$. Applying the Benamou-Brenier stability (in $W_2$) and the Fisher-Rao contractivity for linear ODEs in density space with source (cf. Ambrosio et al., 2008, Ch. 8) gives

$$\sup_{t\in[t_{k-1},t_k]} W_2\big(\rho_t^{(HK)}, \rho_t^{(W)}\big) \le \tilde{C}\,\kappa^{-1}\sqrt{\log(1+\kappa^2)},$$

$$\sup_{t\in[t_{k-1},t_k]} d_{FR}\big(\rho_t^{(HK)}, \rho_t^{(FR)}\big) \le \tilde{C}'\,\kappa\sqrt{\log(1+\kappa^{-2})}.$$

Finally, concatenate segments using Lemma 4.

The dependence of constants on $(d, \lambda_{\min}, \lambda_{\max})$ and $[D]_{C^{0,\alpha}}, \|b\|_{C^{0,\alpha}}$ follows from elliptic regularity for the HJB potentials and compactness of anchors. $\qquad\square$

**Proposition 2** (Strict convexity and uniqueness of VI–HK minimiser). *Assume $D$ uniformly elliptic, $b$ bounded, and restrict to admissible triples $(\rho, v, \xi)$ with finite second moment and $\xi \in L^2(\rho)$ on $[0, T]$. The functional*

$$\mathcal{A}_{HK}[\rho, v, \xi] = \int_0^T \int \tfrac{1}{4} \langle v - b, D^{-1}(v - b) \rangle \, \rho \, dx \, dt$$

$$+ \tfrac{\kappa^2}{2} \int_0^T \int \xi^2 \, \rho \, dx \, dt$$

*is strictly convex in $(m, \zeta) := (\rho v, \xi \rho)$ on the affine constraint set $\{\partial_t \rho + \nabla \cdot m = \zeta\}$; hence the VI–HK problem admits a unique minimiser.*

*Proof.* Define the perspective functionals on $\mathcal{M} \times \mathcal{P}$,

$$J(m, \rho) := \tfrac{1}{4} \int \left\langle \tfrac{m}{\rho} - b, D^{-1}\left(\tfrac{m}{\rho} - b\right) \right\rangle \rho, \qquad H(\zeta, \rho) := \tfrac{\kappa^2}{2} \int \left(\tfrac{\zeta}{\rho}\right)^2 \rho,$$

with the convention $+\infty$ off $\{\rho > 0, \ m \ll \rho, \ \zeta \ll \rho\}$. For every $x$, the maps $w \mapsto \tfrac{1}{4} \langle w - b, D^{-1}(w - b) \rangle$ and $s \mapsto \tfrac{1}{2} s^2$ are strictly convex. By standard properties of convex integral functionals in perspective form (Reshetnyak/Ioffe; cf. Lemma 2), $J$ and $H$ are strictly convex in $(m, \rho)$ and $(\zeta, \rho)$ along affine lines that are not trivial scalings of $(m, \rho)$ or $(\zeta, \rho)$. Hence their sum $\mathcal{A}_{HK}(m, \zeta, \rho) := J(m, \rho) + H(\zeta, \rho)$ is strictly convex in $(m, \zeta)$ for fixed $\rho$, and jointly strictly convex in $(m, \zeta, \rho)$ modulo the natural invariances.

The admissible set $\mathcal{C} := \{(\rho, m, \zeta) : \partial_t \rho + \nabla \cdot m = \zeta, \ (\rho_{t_k}) = \mu_k\}$ is affine and narrowly closed under the three-layer weak topology used in Appendix H. Lower semicontinuity of $\mathcal{A}_{HK}$ on this topology ensures existence of a minimiser. If $(\rho^1, m^1, \zeta^1) \neq (\rho^2, m^2, \zeta^2)$ are two admissible minimisers with finite action, strict convexity gives, for $\theta \in (0, 1)$,

$$\mathcal{A}_{HK}\big((1 - \theta)(\rho^1, m^1, \zeta^1) + \theta(\rho^2, m^2, \zeta^2)\big) < (1 - \theta) \, \mathcal{A}_{HK}(\rho^1, m^1, \zeta^1) + \theta \, \mathcal{A}_{HK}(\rho^2, m^2, \zeta^2),$$

contradicting optimality. Therefore the minimiser is unique. $\qquad \square$

**Remark (constants).** The rate constants inherit dependence on $(d, \lambda_{\min}, \lambda_{\max})$, regularity of $(D, b)$, anchor bounds/regularity, and segment lengths as in Theorem 3; see also Lemma 4.

**Remark (dense grids $\Delta_{\min} \to 0$).** If the grid is refined with fixed horizon $T$ and $\Delta_{\min} \to 0$, then $K \leq T/\Delta_{\min}$ may grow. The bound in Lemma 4 scales at most linearly with $K$ through $\bar{C}' \sim K \bar{C}'_\star$. Under additional regularity ensuring a per–unit–time error bound $\bar{C}_0 \varepsilon \log(1/\varepsilon)$ (e.g., uniform curvature/metric–speed bounds along the $W_2$ geodesic), a Riemann–sum argument gives a constant independent of $K$: $\sum_k C_0 \varepsilon \log(1/\varepsilon) \Delta t_k = C_0 T \varepsilon \log(1/\varepsilon)$. Thus densification does not deteriorate the rate constant when segment-wise constants are uniformly integrable in time.

# I   SEGMENT-WISE GLUING AND $W_2$ STABILITY

We collect a stability estimate that turns segment-wise entropic$\to W_2$ bounds into global estimates.

**Lemma 4** (Segment gluing for entropic$\to W_2$ rates). *Let $0 = t_0 < t_1 < \cdots < t_K = T$ with $\Delta t_k \in [\Delta_{\min}, \Delta_{\max}]$. Suppose for each segment $[t_{k-1}, t_k]$ there exist constants $C_k, C'_k$ such that for all sufficiently small $\varepsilon > 0$*

$$\sup_{t \in [t_{k-1}, t_k]} W_2\big(\rho_t^\varepsilon, \rho_t^{(W)}\big) \leq C_k \sqrt{\varepsilon \log(1/\varepsilon)}, \qquad 0 \leq \mathcal{A}_\varepsilon^{(k)} - \tfrac{1}{2} \frac{W_2(\mu_{k-1}, \mu_k)^2}{\Delta t_k} \leq C'_k \varepsilon \log(1/\varepsilon),$$

*where $\mathcal{A}_\varepsilon^{(k)}$ is the segment-wise kinetic action. Assume uniformity: $C_k \leq C_\star$, $C'_k \leq C'_\star$ for all $k$. Then there exist $\bar{C}, \bar{C}' > 0$ depending only on $(C_\star, C'_\star, K, \Delta_{\min}, \Delta_{\max})$ such that*

$$\sup_{t \in [0, T]} W_2\big(\rho_t^\varepsilon, \rho_t^{(W)}\big) \leq \bar{C} \sqrt{\varepsilon \log(1/\varepsilon)}, \qquad 0 \leq \inf_Q \mathcal{F}_\sigma(Q) - \tfrac{1}{2} \sum_{k=1}^K \frac{W_2(\mu_{k-1}, \mu_k)^2}{\Delta t_k} \leq \bar{C}' \varepsilon \log(1/\varepsilon).$$

*Proof.* The global $W_2$ bound follows because $[0, T]$ is the disjoint union of segments and $t \mapsto W_2(\rho_t^\varepsilon, \rho_t^{(W)})$ is continuous on each segment; hence the supremum on $[0, T]$ is the maximum of segment suprema, bounded by $\max_k C_k \sqrt{\varepsilon \log(1/\varepsilon)}$. For the action, concatenate the segment-wise bridges to build a feasible global curve; the Benamou–Brenier action is additive under concatenation, so the excess over the piecewise $W_2$ length is at most $\sum_k C_k' \varepsilon \log(1/\varepsilon)$. Bounding $K \leq T/\Delta_{\min}$ yields the announced $\bar{C}'$.

To justify the uniformity of constants from Corollary 8, use elliptic regularity and compactness: the hypotheses (uniform ellipticity of $D$, anchor density bounds and $H^s$ regularity, compact support, grid bounds) provide segment-independent estimates for (i) the parabolic HJB solutions (via $C^{0,\alpha}$ coefficients), (ii) the stability of the continuity equation (Gronwall in $W_2$ using the dual Benamou–Brenier formulation), and (iii) the static EOT rate constants. Hence $C_k, C_k'$ can be chosen uniformly. $\qquad\square$

## J  PROOF OF THEOREM 1: VI-MMSB EQUIVALENCE

We work on the canonical space $(\Omega, \mathcal{C})$, $\Omega = C([0, T], \mathbb{R}^d)$ with its Borel $\sigma$–field. For $t \in [0, T]$, $\pi_t(\omega) = \omega(t)$. The OU reference law $P_{\text{ref}}$ of equation 2 is Radon; its $t$–marginal density $r(t, \cdot)$ is strictly positive and absolutely continuous w.r.t. Lebesgue.

**Posterior tilt.** For observations $y_{1:K}$ at $0 < t_1 < \cdots < t_K$, set $L_y(\omega) := \prod_{k=1}^K \ell(y_k \mid \pi_{t_k}(\omega))$ with $\ell > 0$ measurable and assume

$$0 < Z := E_{P_{\text{ref}}}[L_y] < \infty.$$

Define the posterior $P_{\text{post}}$ by $\mathrm{d}P_{\text{post}}/\mathrm{d}P_{\text{ref}} = Z^{-1} L_y$.

**Free energy and Gibbs identity.** For $Q \ll P_{\text{ref}}$ define

$$\mathcal{F}[Q] := D_{\text{KL}}(Q \| P_{\text{ref}}) - E_Q[\log L_y].$$

Then the standard calculation yields

$$D_{\text{KL}}(Q \| P_{\text{post}}) = E_Q\Big[\log \frac{\mathrm{d}Q}{\mathrm{d}P_{\text{ref}}} - \log L_y + \log Z\Big] = \mathcal{F}[Q] + \log Z,$$

hence

$$\mathcal{F}[Q] = D_{\text{KL}}(Q \| P_{\text{post}}) - \log Z,$$

with the unique minimiser $Q^\star = P_{\text{post}}$.

**On marginal calculus.** Let $Q_{t_k} = Q \circ \pi_{t_k}^{-1}$ and define $\rho_{t_k}^{\text{obs}}(x) \propto r(t_k, x)\, \ell(y_k \mid x)$. For each $k$ one has the exact identity

$$E_{Q_{t_k}}[\log r] - E_{Q_{t_k}}[\log \rho_{t_k}^{\text{obs}}] = D_{\text{KL}}(Q_{t_k} \| \rho_{t_k}^{\text{obs}}) - D_{\text{KL}}(Q_{t_k} \| r(t_k, \cdot)),$$

so any representation of $\mathcal{F}[Q]$ involving $\sum_k D_{\text{KL}}(Q_{t_k} \| \rho_{t_k}^{\text{obs}})$ necessarily carries the additional negative terms $-\sum_k D_{\text{KL}}(Q_{t_k} \| r)$. In particular, one cannot deduce from such a decomposition that a minimiser must satisfy $Q_{t_k} = \rho_{t_k}^{\text{obs}}$ (the anchors are $\mu_k$; for $K = 1$, $\mu_{t_1} = \rho_{t_1}^{\text{obs}}$).

**Relation to multi–marginal Schrödinger bridge.** Given target marginals $\{\mu_k\}_{k=1}^K$ with $\mu_k \ll r(t_k, \cdot)\lambda_d$, the entropy projection

$$\min_{Q \ll P_{\text{ref}}} \big\{ D_{\text{KL}}(Q \| P_{\text{ref}}) : Q_{t_k} = \mu_k, \ \forall k \big\}$$

has a unique solution with Doob $h$–transform structure

$$\frac{\mathrm{d}Q}{\mathrm{d}P_{\text{ref}}}(\omega) = \frac{\prod_{k=1}^K \phi_k(\pi_{t_k}(\omega))}{E_{P_{\text{ref}}}\big[\prod_{k=1}^K \phi_k(\pi_{t_k})\big]}, \qquad \phi_k > 0,$$

where the Schrödinger system determines $\{\phi_k\}$. Choosing $\phi_k = \ell(y_k \mid \cdot)$ yields $Q = P_{\text{post}}$ and hence its own marginals $\mu_k = (P_{\text{post}})_{t_k}$. In general $(P_{\text{post}})_{t_k} \neq \rho_{t_k}^{\text{obs}}$ unless $K = 1$ or a compatibility condition collapses cross–time corrections to 1.

**Measurability and existence.** For measurable $\pi_{t_k}$ and $Q \ll P_{\text{ref}}$, $Q_{t_k} \ll (P_{\text{ref}})_{t_k} \ll \lambda_d$, so densities exist. The Gibbs representation above ensures lower semicontinuity of $\mathcal{F}$ and the existence/uniqueness of the minimiser under $0 < Z < \infty$ without boundedness assumptions on $\log L_y$.

This completes the proof.

## K  EXISTENCE AND UNIQUENESS OF THE OU-BASED MMSB

We now establish that, under the Ornstein–Uhlenbeck reference process specified in equation 2 and the observation–induced marginals $\{\rho_{t_k}^{\text{obs}}\}$ of Assumption 1, the multi–marginal Schrödinger Bridge (MMSB)

$$\mathcal{Q} := \Big\{ Q \in \mathcal{P}(\Omega) : Q_0 = \rho_0,\ Q_{t_k} = \mu_k\ \forall k \Big\}, \qquad Q^{\star} = \arg\min_{Q \in \mathcal{Q}} D_{\text{KL}}(Q \| P_{\text{ref}}) \qquad (21)$$

possesses a *unique* minimiser $Q^{\star}$. Our argument follows the entropy–transport framework of Léonard (2013), specialised to a linear, non–degenerate diffusion.

**Compactness in the narrow topology.** Let $(Q^n)$ be a minimising sequence in $\mathcal{Q}$ with $\sup_n D_{\text{KL}}(Q^n \| P_{\text{ref}}) < \infty$. On the Polish space $\Omega = C([0,T]; \mathbb{R}^d)$, the *entropy inequality* yields for any bounded Borel $\psi$ and $\lambda > 0$,

$$E_{Q^n}[\psi] \le \frac{1}{\lambda}\Big( D_{\text{KL}}(Q^n \| P_{\text{ref}}) + \log E_{P_{\text{ref}}}[e^{\lambda \psi}] \Big).$$

By Fernique's theorem for Gaussian measures on Banach spaces, $E_{P_{\text{ref}}}[\exp\{\lambda \|\omega\|_\infty^2\}] < \infty$ for $\lambda$ small. Choosing $\psi(\omega) = c \|\omega\|_\infty^2$ gives a uniform bound on $E_{Q^n}[\|\omega\|_\infty^2]$, hence tightness by Prokhorov. Passing to a subsequence (not relabelled), $Q^n \Rightarrow Q$ narrowly. Since $t \mapsto \pi_t$ is continuous, the marginal map $Q \mapsto Q \circ \pi_t^{-1}$ is continuous, hence the constraint set $\mathcal{Q}$ is narrowly closed; thus $Q \in \mathcal{Q}$.

**Lower semicontinuity.** The functional $Q \mapsto D_{\text{KL}}(Q \| P_{\text{ref}})$ is narrowly lower semicontinuous on $\mathcal{P}(\Omega)$; therefore

$$D_{\text{KL}}(Q \| P_{\text{ref}}) \le \liminf_{n \to \infty} D_{\text{KL}}(Q^n \| P_{\text{ref}}),$$

and $Q$ is a minimiser, proving existence.

**Strict convexity and uniqueness.** Extend $D_{\text{KL}}(\cdot \| P_{\text{ref}})$ to $\mathcal{P}(\Omega)$ by the convention $D_{\text{KL}}(Q \| P_{\text{ref}}) = +\infty$ when $Q \not\ll P_{\text{ref}}$. If $Q^1 \neq Q^2$ are feasible with finite KL, write $f_i = dQ^i / dP_{\text{ref}}$ and set $Q^\theta = (1-\theta)Q^1 + \theta Q^2$ with density $f_\theta = (1-\theta)f_1 + \theta f_2$. Since $\phi(a) = a \log a$ is strictly convex on $[0, \infty)$,

$$D_{\text{KL}}(Q^\theta \| P_{\text{ref}}) = E_{P_{\text{ref}}}[\phi(f_\theta)] < (1-\theta)E_{P_{\text{ref}}}[\phi(f_1)] + \theta E_{P_{\text{ref}}}[\phi(f_2)]$$

whenever $P_{\text{ref}}(\{f_1 \neq f_2\}) > 0$, which is implied by $Q^1 \neq Q^2$. Hence the minimiser in $\mathcal{Q}$ is unique.

**Feasibility.** By definition $\rho_{t_k}^{\text{obs}}(x) \propto r(t_k, x)\,\ell(y_k \mid x)$, hence $\rho_{t_k}^{\text{obs}} \ll r(t_k, \cdot)$ for all $k$. In this dominated setting, the multi–marginal Schr"odinger system admits potentials producing a coupling with the prescribed marginals (see, e.g., Léonard, 2013, §5), so $\mathcal{Q} \neq \emptyset$.

Combining the above paragraphs yields the following theorem.

**Theorem 7** (Existence and uniqueness of the OU–MMSB)**.** *Under Assumption 1 and the non–degenerate OU reference process equation 2, the optimisation problem equation 21 admits a unique minimiser $Q^{\star}$.* □

## L  PROOF OF COROLLARY I AND ELLIPTIC GEOMETRY DETAILS

This appendix gives a complete variational proof underlying Corollary 2 and Theorem 2. We recall the Onsager–Fokker (Otto–Wasserstein) metric and show that the MMSB solution curve is its Riemannian geodesic; we then extend to elliptic diffusion $D(x)$.

**Preliminaries and assumptions.** We work on the canonical space $(\Omega, \mathcal{C})$ with $\Omega = C([0,T]; \mathbb{R}^d)$ and its Borel $\sigma$–field. For $t \in [0,T]$, $\pi_t(\omega) = \omega(t)$. The reference OU law $P_{\text{ref}}$ solves $dX_t = b(t, X_t)\,dt + \sigma\,dW_t$ with $\sigma > 0$ and drift $b \in C_b^1([0,T] \times \mathbb{R}^d; \mathbb{R}^d)$; its $t$–marginal density $r(t, \cdot)$ is strictly positive. Admissible path measures are those $Q \ll P_{\text{ref}}$ that are Markov absolutely continuous. By Girsanov they admit a progressively measurable control drift $u \in L_{\text{prog}}^2(\Omega \times [0,T]; \mathbb{R}^d)$ and

$$dQ/dP_{\text{ref}} \propto \exp\Big\{ \int_0^T u_t \cdot dW_t - \tfrac{1}{2} \int_0^T \|u_t\|^2 dt \Big\}.$$

We assume Novikov/Kazamaki so that the stochastic exponential is a martingale and the Girsanov change of measure is valid; sufficient conditions include exponential integrability of $\int_0^T \|u_t\|^2 dt$ (we work within the variational class $u \in L_{\text{prog}}^2$). In particular,

$$D_{\text{KL}}(Q\|P_{\text{ref}}) = \frac{1}{2\sigma^2}\,\mathbb{E}_Q\Big[ \int_0^T \|u_t\|^2 dt \Big].$$

Let $\rho_t$ denote the density of $Q_t$; standard well-posedness of the controlled Fokker–Planck equation with $b \in C_b^1$ and $u \in L_{\text{prog}}^2$ yields $\rho_t \in \mathcal{P}_2(\mathbb{R}^d)$ and $\rho \in C([0,T]; \mathcal{P}_2)$. We use the Eulerian velocity $v_t := b + u_t$ and, whenever $-\nabla \cdot (\rho_t \nabla \psi_t) = \dot{\rho}_t$ (in distributions), we write $\nabla \psi_t \in L^2(\rho_t\,dx)$. Observation factors satisfy Assumption 1 and enter as pointwise potentials at observation times.

**Metric Tensor.** Let $\mathcal{P}_2(\mathbb{R}^d)$ be the Wasserstein space of Borel probability measures with finite second moment. For $\rho \in \mathcal{P}_2(\mathbb{R}^d)$ denote by $T_\rho \mathcal{P}_2$ the set of signed measures $\sigma$ with zero total mass $\int \sigma = 0$ and finite second moment. Each $\sigma$ admits a *potential* $\phi$ (defined up to an additive constant) solving the weighted Poisson problem

$$-\nabla \cdot (\rho \nabla \phi) = \sigma \qquad \text{(in distribution)}.$$

Following Liero et al. (2018b), the **Onsager–Fokker metric** $g^{\text{OF}}$ is the bilinear form

$$g_\rho^{\text{OF}}(\sigma_1, \sigma_2) := \frac{1}{\sigma^2} \int_{\mathbb{R}^d} \nabla \phi_1 \cdot \nabla \phi_2\,\rho\,dx, \qquad \text{with } -\nabla \cdot (\rho \nabla \phi_i) = \sigma_i. \tag{22}$$

The $\sigma^{-2}$ prefactor arises from the diffusion matrix $\Sigma = \sigma I$ of the OU reference.

**Weak formulation, function spaces, and tangent representation.** We work with triples $(\rho, v, \psi)$ such that: $\rho \in C([0,T]; \mathcal{P}_2(\mathbb{R}^d))$ with strictly positive densities; $v \in L^2([0,T]; L^2(\rho_t\,dx; \mathbb{R}^d))$; and $\psi \in L^2([0,T]; H_{\text{loc}}^1(\mathbb{R}^d))$ with $\Delta \psi \in L_{\text{loc}}^1$. The continuity equation is interpreted in the weak sense: for all $\varphi \in C_c^\infty((0,T) \times \mathbb{R}^d)$,

$$\int_0^T \int \Big( -\partial_t \varphi\,\rho - \nabla \varphi \cdot \rho v - \tfrac{\sigma^2}{2} \Delta \varphi\,\rho \Big) dx\,dt = 0.$$

The tangent space $T_\rho \mathcal{P}_2$ is represented by zero-mass signed measures $\sigma$ with finite second moment; the Riesz representation under $g^{\text{OF}}$ is given by potentials $\phi \in H^1(\rho\,dx)$ solving $-\nabla \cdot (\rho \nabla \phi) = \sigma$, and $\|\sigma\|_{g^{\text{OF}}}^2 = \sigma^{-2} \int \|\nabla \phi\|^2 \rho\,dx$.

**Lemma 5** (Integration by parts: OU and elliptic). *Assume $b \in C_b^1$ and the reference has strictly positive, smooth marginals with Gaussian tails. If $\rho$ solves the controlled Fokker–Planck equation with $v \in L^2(\rho)$ and $\psi \in H_{\text{loc}}^1$ with $\Delta \psi \in L_{\text{loc}}^1$, then for any compact interval $I \subset (0,T)$*

$$\int_I \int \psi\,\nabla \cdot (\rho v)\,dx\,dt = -\int_I \int \rho\,v \cdot \nabla \psi\,dx\,dt.$$

*Moreover, for a uniformly elliptic, symmetric diffusion tensor $D \in C_b^1(\mathbb{R}^d; \mathbb{R}^{d \times d})$,*

$$\int_I \int \psi\,\nabla \cdot (D \nabla \rho)\,dx\,dt = \int_I \int \rho\,\nabla \cdot (D \nabla \psi)\,dx\,dt,$$

*so in particular when $D \equiv (\sigma^2/2)I$ this reduces to $\int \psi\,\Delta \rho = \int \rho\,\Delta \psi$. No boundary terms arise thanks to Gaussian decay and polynomial growth bounds.*

**Dynamic formulation and KL identity.** For admissible $(\rho, u)$ obeying the controlled Fokker–Planck equation

$$\partial_t \rho_t + \nabla \cdot \big((b + u_t)\rho_t\big) = \tfrac{\sigma^2}{2} \Delta \rho_t,$$

define the purely *control* action

$$\mathcal{A}[\rho, u] = \frac{1}{2\sigma^2} \int_0^T \int_{\mathbb{R}^d} \|u_t(x)\|^2 \, \rho_t(x) \, dx \, dt, \qquad \text{equivalently } \mathcal{A}[\rho, v] = \frac{1}{2\sigma^2} \int_0^T \int \|v_t - b\|^2 \, \rho_t \, dx \, dt.$$

By Girsanov,

$$\mathcal{J}[u] := \frac{1}{2\sigma^2} \mathbb{E}_Q \Big[ \int_0^T \|u_t(X_t)\|^2 \, dt \Big], \qquad \text{so} \quad D_{\mathrm{KL}}(Q \| P_{\mathrm{ref}}) = \mathcal{J}[u] = \mathcal{A}[\rho, u]. \tag{23}$$

**Assumptions for Girsanov (elliptic case).** Throughout this section assume $\Sigma \in C_b^1$, $D(x) := \tfrac{1}{2}\Sigma(x)\Sigma(x)^\top$ is uniformly elliptic, and the control drift $u$ is progressively measurable with $\mathbb{E}_Q \big[ \int_0^T u_t(X_t)^\top D(X_t)^{-1} u_t(X_t) \, dt \big] < \infty$. Under Novikov or Kazamaki conditions the stochastic exponential is a true martingale (see, e.g., Novikov (1972); Kazamaki (1994)).

**Lemma 6** (Girsanov–KL identity under Novikov/Kazamaki (OU/elliptic)). *Let $P_{\mathrm{ref}}$ be the OU law solving $dX_t = b(t, X_t) \, dt + \sigma \, dW_t$ with $\sigma > 0$ and $b \in C_b^1$. Let $u \in L_{\mathrm{prog}}^2(\Omega \times [0, T]; \mathbb{R}^d)$ and set $\theta_t := \sigma^{-1} u_t(X_t)$. If either Novikov's condition $\mathbb{E}_{P_{\mathrm{ref}}}[\exp\{\tfrac{1}{2} \int_0^T \|\theta_t\|^2 dt\}] < \infty$ or Kazamaki's condition holds, then the stochastic exponential*

$$Z_T := \exp \Big\{ \int_0^T \theta_t \cdot dW_t - \tfrac{1}{2} \int_0^T \|\theta_t\|^2 dt \Big\}$$

*is a true $P_{\mathrm{ref}}$–martingale and defines $Q$ via $dQ = Z_T \, dP_{\mathrm{ref}}$. Under $Q$, $W_t^Q := W_t - \int_0^t \theta_s ds$ is a Brownian motion and*

$$dX_t = (b + u_t) \, dt + \sigma \, dW_t^Q.$$

*Moreover, the Kullback–Leibler divergence satisfies the identity*

$$D_{\mathrm{KL}}(Q \| P_{\mathrm{ref}}) = \mathbb{E}_Q[\log Z_T] = \frac{1}{2\sigma^2} \mathbb{E}_Q \Big[ \int_0^T \|u_t(X_t)\|^2 dt \Big]. \tag{24}$$

Proof sketch. *By Itô under $Q$, $\int \theta \cdot dW = \int \theta \cdot dW^Q + \int \|\theta\|^2 dt$. Taking expectations under $Q$ cancels the martingale term and yields equation 24. The Novikov/Kazamaki assumption ensures $Z$ is a martingale and legitimises the change of measure.* $\square$

Let $v_t = b + u_t$. Then the Eulerian form reads

$$\mathcal{J}[v] = \frac{1}{2\sigma^2} \int_0^T \int_{\mathbb{R}^d} \|v_t(x) - b(t, x)\|^2 \, \rho_t(x) \, dx \, dt, \tag{25}$$

under the controlled Fokker–Planck constraint

$$\partial_t \rho_t + \nabla \cdot (\rho_t v_t) = \tfrac{\sigma^2}{2} \Delta \rho_t. \tag{26}$$

In particular, minimizing KL under the MMSB marginal constraints is equivalent to minimizing $\mathcal{A}$. By the dynamic formulation of optimal transport (Benamou–Brenier) and its Onsager variant (see, e.g., Liero et al. (2018b)), the minimal value of $\mathcal{A}$ equals the squared length of $(\rho_t)$ under $g^{\mathrm{OF}}$, thereby endowing equation 22 with a Riemannian interpretation.

**Elliptic diffusion: KL identity and Eulerian form.** Let the reference diffusion be $dX_t = b(t, X_t) \, dt + \Sigma(X_t) \, dW_t$ with $D(x) = \tfrac{1}{2}\Sigma(x)\Sigma(x)^\top$ uniformly elliptic and $\Sigma \in C_b^1$. Define the control drift $u_t$ by $dX_t = (b + u_t) \, dt + \Sigma \, dW_t$ under $Q$. Then

$$D_{\mathrm{KL}}(Q \| P_{\mathrm{ref}}) = \frac{1}{4} \mathbb{E}_Q \Big[ \int_0^T u_t(X_t)^\top D(X_t)^{-1} u_t(X_t) \, dt \Big], \tag{27}$$

and in Eulerian variables

$$\mathcal{J}[v] = \frac{1}{4} \int_0^T \int \langle v_t - b, D^{-1}(v_t - b) \rangle \, \rho_t \, dx \, dt, \qquad \partial_t \rho + \nabla \cdot (\rho v) = \nabla \cdot (D \nabla \rho). \tag{28}$$

The proof follows by setting $\theta_t = \Sigma(X_t)^{-1} u_t(X_t)$ in the stochastic exponential and using $\Sigma\Sigma^\top = 2D$.

**First variations and optimality system.**   Introduce the Lagrangian

$$\mathcal{L}(\rho, v, \psi) = \frac{1}{2\sigma^2} \int_0^T \int \|v - b\|^2 \rho \, dx \, dt \; + \; \int_0^T \int \psi \Big(\partial_t \rho + \nabla \cdot (\rho v) - \tfrac{\sigma^2}{2} \Delta \rho\Big) dx \, dt,$$

with the understanding that time-slice marginals are fixed at observation times ("hard constraints"). Variations are taken with $\delta v \in L^2(\rho)$ compactly supported in time away from observation times, and $\delta \rho \in C_c^\infty((t_{k-1}, t_k) \times \mathbb{R}^d)$ with zero total mass for each $t$.

**Lemma 7** (Variation in $v$ (OU and elliptic)). *For the OU Lagrangian $\mathcal{L} = \frac{1}{2\sigma^2} \int \|v - b\|^2 \rho + \int \psi(\partial_t \rho + \nabla \cdot (\rho v) - \frac{\sigma^2}{2} \Delta \rho)$, using Lemma 5 the first variation gives $v - b = \sigma^2 \nabla \psi$. For the elliptic Lagrangian $\mathcal{L} = \frac{1}{4} \int \langle v - b, D^{-1}(v - b)\rangle \rho + \int \psi(\partial_t \rho + \nabla \cdot (\rho v) - \nabla \cdot (D \nabla \rho))$, one obtains $v - b = 2D\nabla \psi$.*

**Lemma 8** (Variation in $\rho$ (OU and elliptic)). *For variations $\delta \rho$ supported away from observation times and endpoints, integration by parts in time and space yields for OU*

$$\delta_\rho \mathcal{L} = \int_0^T \int \Big(\tfrac{1}{2\sigma^2} \|v - b\|^2 - \partial_t \psi - v \cdot \nabla \psi - \tfrac{\sigma^2}{2} \Delta \psi\Big) \delta \rho \, dx \, dt,$$

*and for the elliptic case (using Lemma 5)*

$$\delta_\rho \mathcal{L} = \int_0^T \int \Big(\tfrac{1}{4} \langle v - b, D^{-1}(v - b)\rangle - \partial_t \psi - v \cdot \nabla \psi - \nabla \cdot (D \nabla \psi)\Big) \delta \rho \, dx \, dt.$$

*Substituting the stationarity conditions $v - b = \sigma^2 \nabla \psi$ and $v - b = 2D\nabla \psi$ gives the HJB equations*

$$-\partial_t \psi = \tfrac{1}{2} \|\nabla \psi\|^2 + \langle \nabla \psi, b\rangle + \tfrac{\sigma^2}{2} \Delta \psi \quad (OU), \qquad -\partial_t \psi = \tfrac{1}{2} \|\nabla \psi\|_{2D}^2 + \langle \nabla \psi, b\rangle + \nabla \cdot (D \nabla \psi) \quad (elliptic). \tag{29}$$

**Boundary and jump conditions.**   With hard marginal constraints at $t \in \{0, t_1, \ldots, t_K, T\}$ the admissible $\delta \rho$ vanish on these time slices, so no boundary terms arise from time integration. If, alternatively, observations are imposed softly via potentials $V_k(x) = -\log \ell(y_k \mid x)$ added to the objective as $\sum_k \int V_k(x) \rho_{t_k}(x) \, dx$, then allowing free $\delta \rho(t_k, \cdot)$ produces the jump conditions

$$\psi(t_k^+, x) - \psi(t_k^-, x) = V_k(x), \qquad k = 1, \ldots, K, \tag{30}$$

as claimed.

Equation equation 29 is precisely the HJB PDE in the main text. Eliminating $u_t = v_t - b$ between equation 10 and the FP constraint recovers the controlled Fokker–Planck dynamics. Hence any KL minimiser yields $(\rho_t, \psi_t)$ solving the coupled HJB–FP system, i.e. the geodesic equation for $g^{\text{OF}}$.

**Length–energy equivalence and geodesic characterisation.**   For any sufficiently regular curve $t \mapsto \rho_t$ in $\mathcal{P}_2$ with tangent $\dot{\rho}_t = -\nabla \cdot (\rho_t w_t)$, define the $g^{\text{OF}}$–length and energy

$$\text{Len}(\rho)^2 := \Big(\int_0^T \|\dot{\rho}_t\|_{g^{\text{OF}}(\rho_t)} \, dt\Big)^2, \qquad \mathcal{E}(\rho) := \frac{1}{2} \int_0^T \|\dot{\rho}_t\|_{g^{\text{OF}}(\rho_t)}^2 \, dt.$$

By Cauchy–Schwarz, minimisers of $\mathcal{E}$ under fixed endpoints coincide with constant–speed minimisers of Len; moreover, for admissible $(\rho, v)$ with $\dot{\rho} = -\nabla \cdot (\rho v) - \frac{\sigma^2}{2} \Delta \rho$, the metric norm satisfies

$$\|\dot{\rho}_t\|_{g^{\text{OF}}(\rho_t)}^2 = \sigma^{-2} \int \|v_t - b\|^2 \rho_t \, dx,$$

so that $\mathcal{E}(\rho) = \mathcal{A}[\rho, v]$. Therefore, the OU–MMSB variational problem is exactly the energy minimisation that defines $g^{\text{OF}}$–geodesic.

**Corollary 5** (Geodesic characterisation under $g^{\text{OF}}$). *Let $\{\rho_t\}_{t \in [0,T]}$ be admissible with prescribed marginals at $\{0, t_1, \ldots, t_K, T\}$. Then the following are equivalent:*

1. *$\{\rho_t\}$ minimises $D_{\text{KL}}(Q \| P_{\text{ref}})$ among all admissible $Q$ with these marginals (OU–MMSB solution).*

2. *There exists $\psi$ such that $(\rho, v, \psi)$ satisfies equation 10–equation 29 and the weak FP constraint equation 26 (with jumps equation 30 when observations are soft).*

3. *$\{\rho_t\}$ is a constant–speed geodesic for the metric $g^{\text{OF}}$ connecting the constrained time slices.*

*Proof.* (1)$\Rightarrow$(2) is Lemmas 7–8. (2)$\Rightarrow$(3) follows from $\mathcal{A} = \mathcal{E}$ and standard length–energy equivalence on Riemannian manifolds of measures (Benamou–Brenier type arguments, adapted to $g^{\text{OF}}$). (3)$\Rightarrow$(1) uses the energy minimality of constant–speed geodesics and $\mathcal{E} = \mathcal{A}$. $\qquad\square$

**Uniqueness.** By Appendix K, the OU–MMSB minimiser of $D_{\text{KL}}(\cdot\|P_{\text{ref}})$ is unique. Via the identity equation 23 (equivalently, equation 25), the control action $\mathcal{A}$ is strictly convex in $u$, hence the induced density curve is the unique geodesic compatible with the prescribed marginals (Corollary 2).

*Proof of Corollary 2.* By the identity equation 23–equation 25, the unique KL minimiser coincides with the unique minimiser of $\mathcal{A}$. By the Euler–Lagrange conditions equation 10–equation 29, this minimiser solves the HJB–FP geodesic system; hence $(\rho_t)$ is precisely the Riemannian geodesic for equation 22. $\qquad\square$

**Well–posedness and regularity summary.** We collect the hypotheses ensuring each step above is rigorous.

- **Existence/uniqueness of the MMSB minimiser.** Under Assumption 1 and non–degenerate OU reference, the feasible set is nonempty, tight and narrowly closed; lower semicontinuity and strict convexity of KL yield a unique minimiser (see the theorem at the end of Appendix K).

- **Controlled Fokker–Planck well–posedness.** For $b \in C_b^1$ and $u \in L^2_{\text{prog}}$, the controlled FP equation admits a unique weak solution with $\rho \in C([0, T]; \mathcal{P}_2)$; Gaussian tails of the OU reference propagate, justifying the integration by parts in Lemma 5. Standard references include dynamic OT/Benamou–Brenier and parabolic PDE texts (e.g., Benamou & Brenier (2000); Ambrosio et al. (2008); Villani (2003; 2009)).

- **Girsanov–KL identity.** Lemma 6 gives $D_{\text{KL}}(Q\|P_{\text{ref}}) = \frac{1}{2\sigma^2}\mathbb{E}_Q \int \|u_t\|^2 dt$ under Novikov/Kazamaki (Novikov (1972); Kazamaki (1994)), ensuring the Eulerian action equals the pathwise KL.

- **HJB weak solution and optimality system.** With $b \in C_b^1$ and $\sigma > 0$, the semilinear parabolic HJB equation 29 is well–posed in the weak/viscosity sense (see Crandall et al. (1992); Bardi & Capuzzo-Dolcetta (1997); Fleming & Soner (2006)); $\psi$ is unique up to an additive constant on each open interval $(t_{k-1}, t_k)$. The relation $v - b = \sigma^2 \nabla\psi$ holds in $L^2(\rho)$ and defines the metric velocity.

- **Observation modelling.** Hard marginal constraints enforce $\delta\rho = 0$ at observation times; soft constraints via $V_k$ yield jumps equation 30. Both regimes are compatible with the weak formulation and the geodesic characterisation.

# M  PROOF OF COROLLARY II (HIGH-DIFFUSION LIMIT)

**Corollary 6** (High-Diffusion Limit: Mixture Geodesics between posterior anchors)**.** *As $\sigma \to \infty$, the MMSB–VI minimizers $Q^\sigma$ converge in law to a curve of mixture geodesics connecting the posterior anchors $\{\mu_k\}$, where $\mu_k = (P_{post})_{t_k}$.*

*Proof via entropic I–projection.* Corollary 3 asserts that when the diffusion scale $\sigma$ tends to infinity the family of MMSB–VI minimisers $Q^\sigma$ converges (in law) to a curve of *mixture (m–connection) geodesics* connecting the posterior anchors. In this regime the Onsager–Fokker metric rescales as $g^{\text{OF}} \propto \sigma^{-2}$ and collapses; a non–trivial limit must therefore be selected by an *entropic* (maximum entropy) principle. Our proof proceeds via Csiszár I–projection geometry and the Schrödinger system (see Csiszár (1975); Léonard (2013; 2014a); Gentil et al. (2017)).

**Notation and setting.** Let $\mathcal{P}(\Omega)$ be the space of Borel probability measures over continuous paths $\Omega := C([0, T]; \mathbb{R}^d)$ endowed with the weak topology. For any admissible path measure $Q$ with control drift $u$ we recall the *rescaled Onsager–action*

$$\mathcal{F}_\sigma(Q) := \frac{1}{2\sigma^2} \mathbb{E}_Q \big[ \int_0^T \|u_t(X_t)\|^2 \, dt \big], \qquad \sigma > 0. \tag{31}$$

For every $\sigma$ the MMSB optimum $Q^\sigma$ is the unique minimiser of $\mathcal{F}_\sigma$ subject to the marginal constraints $Q_0^\sigma = \rho_0$, $Q_{t_k}^\sigma = \mu_k$. We write $\mathcal{K}$ for this closed and tight constraint set in $\mathcal{P}(\Omega)$.

**Uniform tightness and equi–coercivity.** Choose a feasible $Q^0 \in \mathcal{K}$ with strictly positive time–slice densities (e.g. from the Schrödinger system/Doob's $h$–transform). By optimality of $Q^\sigma$, $0 \leq \mathcal{F}_\sigma(Q^\sigma) \leq \mathcal{F}_\sigma(Q^0) \leq C/\sigma^2$ for some constant $C$ independent of $\sigma$. Hence $\sup_{\sigma > 1} \mathcal{F}_\sigma(Q^\sigma) < \infty$ and $\{Q^\sigma\}_{\sigma > 1} \subset \mathcal{K}$ is tight by Prokhorov; every sequence admits a weakly convergent subsequence.

**High–temperature (large–$\sigma$) limit as an I–projection.** For each $\sigma$, the MMSB solution is the Csiszár I–projection of the OU reference $P_{\mathrm{ref}}^\sigma$ onto the affine constraint set $\mathcal{K}$. Fix $s < t$ and let $K_{t-s}^\sigma(x, y)$ denote the two–time density of $P_{\mathrm{ref}}^\sigma$. As $\sigma \to \infty$ the kernel flattens uniformly on compacts, hence the I–projection onto the coupling set with marginals $(\mu, \nu)$ converges to the independent coupling $\mu \otimes \nu$ (see Csiszár (1975); Léonard (2013; 2014b); Gentil et al. (2017)).

**Characterisation of the limit curve (m–geodesic).** Fix an interval $(t_{k-1}, t_k)$. In the $\sigma \to \infty$ limit, the (regularised) coupling between $t_{k-1}$ and $t_k$ degenerates to the *independent* coupling compatible with the endpoints; projecting intermediate marginals at time $t \in (t_{k-1}, t_k)$ yields the straight line segment in mixture coordinates

$$\rho_t^{(m)} = (1 - \theta) \, \mu_{k-1} + \theta \, \mu_k, \qquad \theta = \frac{t - t_{k-1}}{t_k - t_{k-1}}, \tag{32}$$

which is the m–connection geodesic of information geometry (distinct from Fisher–Rao Levi–Civita). This follows from the fact that I–projections onto linear marginal constraints are affine in the density (mixture) coordinates, and when the regularisation dominates the entropic interpolation collapses to these affine segments (cf. Gentil et al. (2017); Léonard (2014b)).

**Lemma 9** (Entropic interpolation as $\sigma \to \infty$). *Assume $r(t, \cdot) > 0$ and $\mu_k \ll r(t_k, \cdot)$ for all $k$, and moreover work either on a compact domain (e.g., reflecting boundary) or on a discrete grid where each transition kernel is strictly positive and satisfies a Doeblin lower bound. Let $\mu, \nu$ be two anchors at times $s < t$. Consider the two–time Schrödinger bridge induced by the OU reference $P_{\mathrm{ref}}^\sigma$. As $\sigma \to \infty$, the optimal coupling (as a Csiszár I–projection) converges in total variation to the independent coupling $\mu \otimes \nu$. Consequently, the intermediate marginal at any $\tau \in (s, t)$ is the mixture segment in density coordinates, $\rho_\tau = (1 - \theta)\mu + \theta\nu$ with $\theta = (\tau - s)/(t - s)$. In the multi–marginal case, the statement holds on each interval $(t_{k-1}, t_k)$, and concatenation yields the piecewise mixture curve connecting $\{\mu_k\}$.*

*Proof sketch.* Viewed as an entropy minimisation under linear constraints, the SB solution is the Csiszár I–projection of $P_{\mathrm{ref}}^\sigma$ onto the constraint set. As $\sigma \to \infty$, the two–time kernels flatten and alternating I–projections reduce to projections onto time–slice marginal sets, yielding the independent coupling and hence the affine evolution in mixture coordinates. Convergence of cyclic I–projections follows from the Pythagorean identity and standard Bregman projection theory; see Csiszár (1975); Léonard (2013; 2014b); Gentil et al. (2017). □

**Global path and uniqueness.** The multi-marginal IPFP updates are alternating I–projections in KL geometry onto the affine marginal sets $\{P : P_{t_k} = \mu_k\}$ (see Appendix P and Csiszár (1975); Léonard (2013)). As $\sigma \to \infty$, the dynamic term vanishes in the KL scale and the cyclic I–projections converge to the unique information projection onto the intersection $\bigcap_k \{P : P_{t_k} = \mu_k\}$. The induced time–marginal curve on each interval $(t_{k-1}, t_k)$ is exactly the mixture (m–connection) segment equation 32, and concatenating these segments yields a unique global piecewise m–geodesic limit connecting $\{\mu_k\}$.

**Uniqueness statement.** Uniqueness follows from strict convexity of the KL divergence in the Csiszár projection scheme and the closed convexity of each marginal constraint set: there is a unique fixed point of the cyclic I–projections (cf. Csiszár (1975)). Therefore the limit path is unique and is determined solely by the anchors $\{\mu_k\}$.

**Likelihood terms in the high-diffusion limit.** The complete variational objective includes both the KL divergence and likelihood terms. For the full VI functional $\mathcal{F}[Q] = D_{\mathrm{KL}}(Q\|P_{ref}) - \mathbb{E}_Q[\log L_y(X)]$, we analyze the behavior of the likelihood terms as $\sigma \to \infty$ under the anchor choice $\mu_k = (P_{\mathrm{post}})_{t_k}$.

As $\sigma \to \infty$, the time-slice marginals $\{\rho_{t_k}^\sigma\}$ of $Q^\sigma$ converge narrowly to $\mu_k$. Hence for $\phi_k(x) := \log \ell(y_k \mid x)$ *bounded and continuous*,

$$\mathbb{E}_{Q^\sigma}[\phi_k(X_{t_k})] = \int \phi_k(x)\, \rho_{t_k}^\sigma(dx) \longrightarrow \int \phi_k(x)\, \mu_k(dx).$$

More generally, if $\phi_k \in L^1(\mu_k)$ and there exists an integrable envelope $g_k \geq 0$ with $|\phi_k| \leq g_k$ and $\sup_\sigma \int g_k\, d\rho_{t_k}^\sigma < \infty$ (uniform integrability), then the same convergence holds by Portmanteau/dominated convergence. At observation times, the limit path satisfies $\rho_{t_k}^{(m)} = \mu_k$ by construction; for $K = 1$, $\mu_{t_1} = \rho_{t_1}^{\mathrm{obs}}$.

**Energy collapse and geometric statement.** Along $Q^\sigma$ the Onsager action equation 31 vanishes as $\sigma \to \infty$; the tensor $g^{\mathrm{OF}} \propto \sigma^{-2}$ collapses, hence no nontrivial OF–geodesic remains in this scaling. The meaningful high–diffusion limit is the *entropic/I–projection* limit, which yields the piecewise m–connection curve (not a $W_2$ displacement geodesic; the latter arises only in the low–diffusion limit). $\qquad\Box$

# N    PROOF OF THEOREM 8 (LOW-DIFFUSION LIMIT)

**Corollary 7** (Low-Diffusion Limit: Wasserstein Geodesics). *As $\sigma \to 0$, the MMSB-VI minimizers $Q^\sigma$ converge to the deterministic transport along the Wasserstein-2 geodesic connecting the prescribed observation marginals.*

We now establish Corollary 4, i.e. the $\Gamma$–convergence of the Onsager functional toward the dynamic formulation of the quadratic Wasserstein distance when $\sigma \to 0$. The argument parallels, but is not symmetrical to, the high–diffusion limit treated in Appendix M; the metric degeneracy calls for a rescaling of time and an appeal to the Benamou–Brenier formula.

**Preliminaries and assumptions.** Denote by $\mathcal{P}_2(\mathbb{R}^d)$ the space of probability measures with finite second moment endowed with the $W_2$ topology (equivalently, the weak/narrow topology plus convergence of second moments). Let $\mathcal{P}(\Omega)$ be the path space as before. For any admissible path measure $Q$ with control drift $u$ we recall the Onsager action

$$\mathcal{F}_\sigma(Q) := \frac{1}{2\sigma^2}\, \mathbb{E}_Q\Big[\int_0^T \|u_t(X_t)\|^2\, dt\Big], \qquad \sigma > 0. \tag{33}$$

The feasible set of measures is again the affine set $\mathcal{K} := \{Q \in \mathcal{P}(\Omega) : Q_0 = \rho_0,\ Q_{t_k} = \mu_k\}$.

**Rescaling of the action.** Unlike the $\sigma \to \infty$ regime, the pre–factor $\sigma^{-2}$ in equation 33 now *blows up*. To obtain a finite limit we introduce the unscaled kinetic functional

$$\mathcal{G}(Q) := \mathbb{E}_Q\Big[\int_0^T \tfrac{1}{2}\|u_t(X_t)\|^2\, dt\Big] = \sigma^2\, \mathcal{F}_\sigma(Q).$$

Minimisation of $\mathcal{F}_\sigma$ under $\mathcal{K}$ is equivalent to minimisation of $\mathcal{G}$ plus the vanishing constant $\frac{1}{2}\sigma^2 \int_0^T \|b(X_t)\|^2 dt$, hence we focus on $\mathcal{G}$.

**Statement of $\Gamma$–convergence.**   Define $\mathcal{G}_0$ on $\mathcal{P}(\Omega)$ by

$$
\mathcal{G}_0(Q) := \begin{cases} \dfrac{1}{2} \displaystyle\int_0^T \int_{\mathbb{R}^d} \|v_t(x)\|^2 \rho_t(x)\, dx\, dt & \text{if } Q \text{ is deterministic with velocity } v_t \\ +\infty & \text{otherwise,} \end{cases}
$$

where "deterministic" means $Q$ is concentrated on absolutely continuous trajectories and $v_t$ is the a.e. defined velocity field satisfying the continuity equation $\partial_t \rho_t + \nabla \cdot (\rho_t v_t) = 0$. The dynamic Benamou–Brenier formula identifies the infimum of $\mathcal{G}_0$ on $\mathcal{K}$ with the squared $W_2$ length of the unique geodesic connecting the target marginals.

**Theorem 8** ($\Gamma$-convergence to Wasserstein geodesics). *In the narrow topology of $\mathcal{P}(\Omega)$, the functionals $\mathcal{G}^\sigma := \sigma^2 \mathcal{F}_\sigma$ $\Gamma$-converge to $\mathcal{G}_0$ as $\sigma \to 0$, where:*

$$
\mathcal{G}^\sigma(Q) := \frac{1}{2}\mathbb{E}_Q\left[\int_0^T \|u_t(X_t)\|^2 dt\right], \quad \mathcal{G}_0(Q) := \begin{cases} \frac{1}{2}\int_0^T \int \|v_t(x)\|^2 \rho_t(x) dx dt & \text{if } Q \text{ is deterministic} \\ +\infty & \text{otherwise} \end{cases}
$$

**Corollary 8** (Quantitative rate toward $W_2$ geodesics; constant dependencies). *Let $\varepsilon := \sigma^2$. Assume:*

- *anchors $\{\mu_k\}_{k=0}^K$ admit densities $m \leq \frac{d\mu_k}{dx} \leq M$ supported on a common compact $K \subset \mathbb{R}^d$ with $\mathrm{diam}(K) \leq R$, and Sobolev regularity $\frac{d\mu_k}{dx} \in H^1(K)$ with $\left\|\frac{d\mu_k}{dx}\right\|_{H^1(K)} \leq S_a$;*

- *time grid spacings satisfy $\Delta t_k := t_k - t_{k-1} \in [\Delta_{\min}, \Delta_{\max}]$;*

- *the reference diffusion is uniformly elliptic with $\lambda_{\min} I \preceq D(x) \preceq \lambda_{\max} I$ and Hölder regularity $D \in C^{0,\alpha}(K)$ with $[D]_{C^{0,\alpha}} \leq H_D$; optionally $D \in C^{1,\alpha}(K)$ with $\|D\|_{C^{1,\alpha}} \leq L_D$ for sharper constants; the drift $b \in C^{0,\alpha}(K)$ with $\|b\|_{C^{0,\alpha}} \leq L_b$.*

*Then there exist constants $C, C' > 0$ depending only on $\left(d, \lambda_{\min}, \lambda_{\max}, H_D, L_D, L_b, m, M, S_a, R, \Delta_{\min}, \Delta_{\max}, K\right)$ such that for each interval $(t_{k-1}, t_k)$*

$$
\sup_{t \in [t_{k-1}, t_k]} W_2\big(\rho_t^\sigma, \rho_t^{(W)}\big) \leq C \sqrt{\varepsilon \log(1/\varepsilon)}
$$

*and the action gap satisfies*

$$
0 \leq \inf_{Q \in \mathcal{K}} \mathcal{F}_\sigma(Q) - \frac{1}{2}\sum_{k=1}^K \frac{W_2(\mu_{k-1}, \mu_k)^2}{\Delta t_k} \leq C' \varepsilon \log(1/\varepsilon).
$$

*In particular, the segment-wise rate is uniform over $k$ and stable under refinement of the grid as long as $\Delta_{\min} > 0$. Proof sketch. Combine the entropic-OT cost rate $v_\varepsilon - v_0 = \frac{d}{2}\varepsilon \log(1/\varepsilon) + O(\varepsilon)$ (Carlier et al., 2022) with the dynamic equivalence between entropic interpolation and SB (Gentil et al., 2017), propagate bounds through the continuity equation using Grönwall and elliptic regularity under the stated constants, and sum segment-wise. Uniform ellipticity transfers constants from the isotropic to the elliptic case via $\lambda_{\min}, \lambda_{\max}$ and $L_D$.*

*Proof.* We prove the two inequalities required for $\Gamma$-convergence.

**Liminf inequality.**   Let $Q^\sigma \Rightarrow Q$ narrowly. We need to show $\liminf_{\sigma \to 0} \mathcal{G}^\sigma(Q^\sigma) \geq \mathcal{G}_0(Q)$.

If $Q$ is not concentrated on absolutely continuous trajectories, then $\mathcal{G}_0(Q) = +\infty$ and the inequality is trivial.

If $Q$ is deterministic with velocity field $v_t$, then by the Freidlin-Wentzell lower bound (see (Dembo & Zeitouni, 1998), Theorem 5.6.7), for any sequence $Q^\sigma$ converging to $Q$:

$$
\liminf_{\sigma \to 0} \mathcal{G}^\sigma(Q^\sigma) \geq I(Q) = \frac{1}{2}\int_0^T \int \|v_t(x)\|^2 \rho_t(x) dx dt = \mathcal{G}_0(Q)
$$

where $I(Q)$ is the rate function from the large deviation principle. $\qquad\square$

**Supremum inequality (Feasible recovery via Schrödinger tilt).** We construct a recovery sequence $\{Q^\sigma\}_{\sigma>0}$ that is *exactly feasible* for all $\sigma > 0$ (i.e. $Q^\sigma_{t_k} = \rho^{\mathrm{obs}}_{t_k}$ for every $k$) and satisfies $\limsup_{\sigma\downarrow 0} \mathcal{G}^\sigma(Q^\sigma) \leq \mathcal{G}_0(Q)$.

**Standing regularity.** Assume: (i) $v \in L^2([0,T] \times \mathbb{R}^d; \rho_t dx dt)$ with $x \mapsto v_t(x)$ locally Lipschitz uniformly in $t$; (ii) the observation–induced marginals $\rho^{\mathrm{obs}}_{t_k}$ satisfy $\rho^{\mathrm{obs}}_{t_k} \ll r(t_k, \cdot)$ with densities whose logarithms have at most quadratic growth; (iii) the associated Schrödinger system on the OU baseline admits strictly positive potentials (see Léonard (2013; 2014a)).

**(Base small–noise lift).** Let $P^{\sigma,v}$ be the law of the diffusion solving

$$dX_t = v_t(X_t)\, dt + \sigma\, dW_t, \qquad X_0 \sim \rho_0.$$

By classical Freidlin–Wentzell theory, $P^{\sigma,v} \Rightarrow Q$ narrowly on $\mathcal{P}(\Omega)$ as $\sigma \downarrow 0$, and

$$\mathcal{G}^\sigma(P^{\sigma,v}) = \tfrac{1}{2}\,\mathbb{E}_{P^{\sigma,v}}\int_0^T \|v_t(X_t)\|^2 dt \longrightarrow \tfrac{1}{2}\int_0^T\int \|v_t(x)\|^2 \rho_t(x) dx dt = \mathcal{G}_0(Q),$$

using continuity of the mapping $\omega \mapsto \int_0^T \|v_t(\omega_t)\|^2 dt$ together with uniform moment bounds; cf. (Dembo & Zeitouni, 1998, Ch. 5).

**(Discrete–time Schrödinger correction).** For each $\sigma > 0$, consider the family of strictly positive potentials $\{\phi^\sigma_k\}_{k=0}^K$ that solve the *finite–time Schrödinger system* over the baseline $P^{\sigma,v}$:

$$\frac{dQ^\sigma}{dP^{\sigma,v}}(\omega) \;=\; \frac{\prod_{k=0}^K \phi^\sigma_k\big(\pi_{t_k}(\omega)\big)}{E_{P^{\sigma,v}}\big[\prod_{k=0}^K \phi^\sigma_k(\pi_{t_k})\big]},$$

with the unique choice of $\{\phi^\sigma_k\}$ ensuring the *exact* marginal constraints $Q^\sigma_{t_k} = \rho^{\mathrm{obs}}_{t_k}$ for all $k$ (existence/uniqueness: (Léonard, 2013, §5), Léonard (2014a); see also the Doob $h$–transform for diffusions, Föllmer (1988)). Thus $Q^\sigma \in \mathcal{K}$ for every $\sigma$.

**(Energy control under $h$–transform).** The Doob transform by *time–sliced* potentials modifies the drift of the baseline diffusion by an additive term of the form $\delta u^\sigma_t(x) = \sigma^2\, \nabla_x \log h^\sigma_t(x)$, where $h^\sigma_t$ solves the associated backward Kolmogorov equations between observation times with terminal data determined by $\{\phi^\sigma_k\}$ (cf. Föllmer (1988)). Under the standing growth assumptions on $\log \phi^\sigma_k$ one obtains

$$\sup_{t\in[t_{k-1},t_k]} \|\nabla \log h^\sigma_t\|_{L^2(\rho^\sigma_t)} \;\leq\; C_k < \infty \qquad \text{(bounds independent of } \sigma\text{)},$$

whence

$$\frac{1}{2}\mathbb{E}_{Q^\sigma}\int_0^T \|\delta u^\sigma_t(X_t)\|^2 dt \;\leq\; \frac{\sigma^4}{2}\sum_{k=1}^K C_k^2(t_k - t_{k-1}) \xrightarrow[\sigma\downarrow 0]{} 0.$$

Therefore

$$\mathcal{G}^\sigma(Q^\sigma) \;=\; \tfrac{1}{2}\mathbb{E}_{Q^\sigma}\int_0^T \|v_t(X_t) + \delta u^\sigma_t(X_t)\|^2 dt \;\leq\; \mathcal{G}^\sigma(P^{\sigma,v}) + o(1) \xrightarrow[\sigma\downarrow 0]{} \mathcal{G}_0(Q).$$

**(Convergence).** Since $E_{P^{\sigma,v}}\big[\prod_k \phi^\sigma_k(\pi_{t_k})\big]$ is finite and strictly positive and the constraint family is finite, the $h$–tilt is an $L^1$ change of measure. Together with Step E1 and tightness of $\{Q^\sigma\}_{\sigma>0}$, we obtain $Q^\sigma \Rightarrow Q$ narrowly (see, e.g., Léonard (2013)). Combining the bounds above yields the desired limsup.

This furnishes an *exactly feasible* recovery sequence with the correct limiting energy. Alternative proofs invoke stochastic control duality and entropic regularisation of dynamic OT (Mikami & Thieullen (2006); Léonard (2014b)), which also deliver the limsup via vanishing–noise/vanishing–regularisation principles.

**Likelihood terms in the low-diffusion limit.** For the complete variational objective $\mathcal{F}[Q] = D_{\mathrm{KL}}(Q\|P_{ref}) - \mathbb{E}_Q[\log L_y(X)]$, we analyze the likelihood terms as $\sigma \to 0$. Under the growth controls stated above (quadratic at most), dominated convergence applies along the feasible recovery $Q^\sigma$.

As $\sigma \to 0$, the path measures $Q^\sigma$ converge narrowly to the deterministic Wasserstein geodesic $Q^0$. Since $Q^0$ is concentrated on a single deterministic path $x^*(t)$, by the continuous mapping theorem:

$$\mathbb{E}_{Q^\sigma}[\log \ell(y_k|X_{t_k})] \to \mathbb{E}_{Q^0}[\log \ell(y_k|X_{t_k})] = \log \ell(y_k|x^*(t_k))$$

where $x^*(t_k)$ is the value of the deterministic geodesic at observation time $t_k$. This ensures that the likelihood terms are properly accounted for in the limiting variational problem.

**Uniform control of Schrödinger correction.** In the discrete-time Schrödinger correction above, the added drift increment satisfies $\delta u_t^\sigma = \sigma^2 \nabla \log h_t^\sigma$. Under the compact-domain/Doeblin assumptions and strictly positive potentials at finitely many time-slices, there exists $C < \infty$ independent of $\sigma$ such that $\sup_t \|\nabla \log h_t^\sigma\|_{L^2(\rho_t^\sigma)} \leq C$. Consequently $\frac{1}{2} \mathbb{E} \int \|\delta u_t^\sigma\|^2 dt \leq C^2 \sigma^4 T \to 0$, which yields the required limsup bound. (See also stability arguments in entropic OT rates and SB dynamic equivalences.)

**Convergence of minimisers.** By Theorem 8 and classical $\Gamma$–convergence theory, any sequence of MMSB minimisers $Q^\sigma$ is relatively compact and every cluster point minimises $\mathcal{G}_0$ on $\mathcal{K}$. The Benamou–Brenier formulation of $W_2$ transport ensures this minimiser is unique and corresponds to the ODE curve (deterministic displacement interpolation). Therefore $Q^\sigma \Rightarrow Q^0$, where $Q^0$ transports mass along the $W_2$–geodesic. Taking marginals at time $t$ yields the required convergence $\rho_t^\sigma \to \rho_t^{(W)}$.

**Topology summary.** The narrow topology on $\mathcal{P}(\Omega)$ is metrised by bounded–Lipschitz distance and induces, at the level of time–marginals, the usual weak convergence in $\mathcal{P}_2(\mathbb{R}^d)$. Under uniform second–moment bounds implied by finite action (cf. Ambrosio et al. (2008); Pavliotis (2014)), narrow convergence plus moment control yields convergence in $W_2$; hence our statement regarding the $W_2$ topology is justified.

This completes the proof of Corollary 4. $\qquad\square$

## O   DETAILED IPFP ALGORITHM AND IMPLEMENTATION

This appendix provides a detailed description of the Multi-Marginal Iterative Proportional Fitting Procedure (IPFP) used to solve the MMSB-VI problem, including implementation details and acceleration techniques.

### O.0.1   PROBLEM FORMULATION AND SCHRÖDINGER SYSTEM

Recall from Proposition 1 that our goal is to solve the MMSB problem:

$$Q^* = \arg \min_{\substack{Q:Q_0=\rho_0 \\ Q_{t_k}=\mu_k,\ k=1,\dots,K}} D_{\mathrm{KL}}(Q\|P_{ref}). \tag{34}$$

The unique solution $Q^*$ is a Markov process whose joint probability density is given by the Schrödinger system:

$$q^*(x_0,\dots,x_K) = \phi_0(x_0)r(t_0,x_0) \prod_{k=1}^{K} K_{t_{k-1}\to t_k}(x_{k-1},x_k)\phi_k(x_k), \tag{35}$$

where $\{\phi_k\}_{k=0}^K$ are non-negative potential functions. The IPFP algorithm is a provably convergent method to find these potentials such that the marginals of $q^*$ match the target marginals $\{\rho_{t_k}^{\mathrm{obs}}\}$.

### O.0.2   LOG-DOMAIN IPFP ALGORITHM

For numerical stability, we operate in the log-domain by setting $\psi_k = \log \phi_k$. The multiplicative updates of IPFP become additive. The core idea is to iteratively update each log-potential $\psi_k$ to enforce the corresponding marginal constraint. To update $\psi_k$, we compute the current marginal at time $t_k$ using all other potentials $\{\psi_j\}_{j\neq k}$. This is achieved via a forward-backward message passing scheme.

**Forward-Backward Messages.** Let $\vec{\alpha}_k(x_k)$ be the forward message at time $t_k$ and $\overleftarrow{\beta}_k(x_k)$ be the backward message at time $t_k$. They are computed recursively in the log-domain:

$$\log \vec{\alpha}_k(x_k) = \log \int K_{t_{k-1} \to t_k}(x_{k-1}, x_k) \, \exp\!\big(\log \vec{\alpha}_{k-1}(x_{k-1}) + \psi_{k-1}^{(j)}(x_{k-1})\big) \, \mathrm{d}x_{k-1}, \quad (36)$$

$$\log \overleftarrow{\beta}_k(x_k) = \log \int K_{t_k \to t_{k+1}}(x_k, x_{k+1}) \, \exp\!\big(\log \overleftarrow{\beta}_{k+1}(x_{k+1}) + \psi_{k+1}^{(j)}(x_{k+1})\big) \, \mathrm{d}x_{k+1}, \quad (37)$$

with initial conditions $\log \vec{\alpha}_0(x_0) = \log r(t_0, x_0)$ and $\log \overleftarrow{\beta}_K(x_K) = 0$.

**Marginal Computation and Update.** The current marginal at time $t_k$, given the potentials, is proportional to the product of the forward message, backward message, and the potential itself. In the log-domain:

$$\log \tilde{\rho}_{t_k}(x_k) = \log \vec{\alpha}_k(x_k) + \log \overleftarrow{\beta}_k(x_k) + \psi_k(x_k) + C_k, \quad (38)$$

where $C_k$ is a normalization constant. The IPFP update for $\psi_k$ that enforces the marginal constraint is then:

$$\psi_k^{\mathrm{new}}(x_k) \leftarrow \psi_k(x_k) + \big(\log \rho_{t_k}^{\mathrm{obs}}(x_k) - \log \tilde{\rho}_{t_k}(x_k)\big). \quad (39)$$

This simplifies to $\psi_k^{\mathrm{new}}(x_k) \leftarrow \log \rho_{t_k}^{\mathrm{obs}}(x_k) - \log \vec{\alpha}_k(x_k) - \log \overleftarrow{\beta}_k(x_k)$. The full algorithm is detailed in Algorithm 1.

---

**Algorithm 1** Detailed Log-Domain Multi-Marginal IPFP

1: **Input**: Target log-marginals $\{\log \rho_{t_k}^{\mathrm{obs}}\}_{k=0}^K$, reference log-marginal $\log r(t_0, \cdot)$, transition kernels $\{K_{t_{k-1} \to t_k}\}_{k=1}^K$, tolerance $\delta > 0$, max iterations $M$.
2: **Initialize**: Log-potentials $\psi_k^{(0)}(x) \leftarrow 0$ for all $x, k$. Iteration $j \leftarrow 0$.
3: **repeat**
4:     $\psi^{\mathrm{prev}} \leftarrow \psi^{(j)}$.
5:     **for** $k = 0$ **to** $K$ **do**
6:         // Forward messages only if $k \geq 1$ (identity for $k = 0$)
7:         $\log \vec{\alpha}_0(x_0) \leftarrow \log r(t_0, x_0)$.
8:         **for** $i = 1$ **to** $k$ **do**
9:           $\log \vec{\alpha}_i(x_i) \leftarrow \log \int K_{t_{i-1} \to t_i}(x_{i-1}, x_i) \, \exp\!\big(\log \vec{\alpha}_{i-1}(x_{i-1}) + \psi_{i-1}^{(j)}(x_{i-1})\big) \, \mathrm{d}x_{i-1}$.
10:        **end for**
11:        // Compute all backward messages up to $t_k$
12:        $\log \overleftarrow{\beta}_K(x_K) \leftarrow 0$.
13:        **for** $i = K - 1$ **down to** $k$ **do**
14:          $\log \overleftarrow{\beta}_i(x_i) \leftarrow \log \int K_{t_i \to t_{i+1}}(x_i, x_{i+1}) \, \exp\!\big(\log \overleftarrow{\beta}_{i+1}(x_{i+1}) + \psi_{i+1}^{(j)}(x_{i+1})\big) \, \mathrm{d}x_{i+1}$.
15:        **end for**
16:        // Update log-potential $\psi_k$
17:        $\psi_k^{(j+1)}(x) \leftarrow \log \rho_{t_k}^{\mathrm{obs}}(x) - \log \vec{\alpha}_k(x) - \log \overleftarrow{\beta}_k(x)$.
18:        To ensure stability, other potentials $\psi_{i \neq k}^{(j+1)} \leftarrow \psi_{i \neq k}^{(j)}$.
19:     **end for**
20:     $j \leftarrow j + 1$.
21:     // Convergence check on the change in potentials
22:     $\epsilon^{(j)} \leftarrow \max_k \|\psi_k^{(j)} - \psi_k^{\mathrm{prev}}\|_{L^\infty}$.
23: **until** $\epsilon^{(j)} < \delta$ **or** $j \geq M$
24: **Return**: Converged log-potentials $\{\psi_k^{(j)}\}_{k=0}^K$.

---

**Implementation note (log normalisation).** After each forward/backward sweep we compute the constant $C_k^{(j)} = \log \sum_x \exp\!\big(\log \vec{\alpha}_k(x) + \log \overleftarrow{\beta}_k(x) + \psi_k^{(j)}(x)\big)$ for every time index $k$ and subtract it from $\log \tilde{\rho}_{t_k}$. This log-sum-exp centring keeps messages numerically stable and prevents overflow when working with high-dimensional grids.

## P  RIGOROUS PROOFS FOR IPFP CONVERGENCE

**Theorem 9** (Multi-Marginal IPFP Convergence). *The Multi-Marginal IPFP algorithm satisfies:*

1. ***Monotonic Descent:*** *The KL divergence to the optimal solution decreases monotonically at each iteration.*

2. ***Linear Convergence:*** *Under non-degenerate conditions, the algorithm converges linearly to the unique solution.*

This appendix provides the complete and rigorous mathematical proofs for the convergence guarantees stated above. We will prove the two main properties separately: (1) monotonic descent of the KL divergence to the optimum, and (2) the linear rate of convergence under non-degenerate conditions.

### P.0.1  PRELIMINARIES: I-PROJECTIONS AND THE PYTHAGOREAN IDENTITY

The foundation of the monotonic descent proof lies in the geometric properties of the Kullback-Leibler (KL) divergence. The KL divergence, also known as relative entropy, is a member of a class of divergences known as Bregman divergences. This geometric structure is key.

**Definition 1** (I-Projection). Let $\mathcal{P}(\mathcal{X})$ be the space of probability measures on a set $\mathcal{X}$. Let $\mathcal{C} \subset \mathcal{P}(\mathcal{X})$ be a closed, convex set of probability measures. The **I-projection** (Information Projection) of a measure $Q \in \mathcal{P}(\mathcal{X})$ onto $\mathcal{C}$ is the unique measure $P^* \in \mathcal{C}$ that minimizes the KL divergence from $Q$:

$$P^* = I_{\mathcal{C}}(Q) := \arg\min_{P \in \mathcal{C}} D_{\mathrm{KL}}(P \| Q). \tag{40}$$

The existence and uniqueness of the I-projection are guaranteed if $\mathcal{C}$ is closed and convex. The central property of I-projections is the following "Pythagorean" identity, due to Csiszár.

**Theorem 10** (Generalized Pythagorean Identity (Csiszár, 1975)). *Let $\mathcal{C}$ be a closed, convex set of probability measures. For any measure $P \in \mathcal{C}$ and any measure $Q$ for which the I-projection $I_{\mathcal{C}}(Q)$ exists, the following identity holds:*

$$D_{\mathrm{KL}}(P \| Q) \geq D_{\mathrm{KL}}(P \| I_{\mathcal{C}}(Q)) + D_{\mathrm{KL}}(I_{\mathcal{C}}(Q) \| Q). \tag{41}$$

*Equality holds if $Q$ is in the "domain of attraction" of the projection, which is true in our setting.*

### P.0.2  PROOF OF MONOTONIC DESCENT

We now apply this machinery to the Multi-Marginal IPFP algorithm. Each step of the algorithm can be seen as an I-projection onto a constraint set.

**Define the Constraint Sets.**  For each $k \in \{0, 1, \ldots, K\}$, let $\mathcal{C}_k$ be the set of path measures that satisfy the $k$-th marginal constraint:

$$\mathcal{C}_k := \{ P \in \mathcal{P}(\Omega) \mid P_{t_k} = \rho_{t_k}^{\mathrm{obs}} \}. \tag{42}$$

Each $\mathcal{C}_k$ is an affine, and therefore closed and convex, subset of $\mathcal{P}(\Omega)$. The unique solution to the MMSB problem, $Q^*$, lies in the intersection of all these sets: $Q^* \in \bigcap_{k=0}^{K} \mathcal{C}_k$.

**Frame IPFP as a Sequence of I-Projections.**  The IPFP algorithm is a cyclic procedure. Let $Q^{(j)}$ be the path measure at the beginning of a cycle. The algorithm sequentially updates the measure to satisfy each constraint. Let's consider the update for constraint $k$. The updated measure, which we denote $Q^{(j+1)}$, is constructed to satisfy $Q_{t_k}^{(j+1)} = \rho_{t_k}^{\mathrm{obs}}$ while being "closest" to $Q^{(j)}$ in the KL sense. This is precisely an I-projection. Specifically, the update step for the $k$-th potential corresponds to finding a new path measure $Q^{(j+1)}$ such that:

$$Q^{(j+1)} = \arg\min_{P \in \mathcal{C}_k} D_{\mathrm{KL}}(P \| Q^{(j)}) = I_{\mathcal{C}_k}(Q^{(j)}). \tag{43}$$

**Apply the Pythagorean Identity.**   Let us apply Theorem 41. We choose our three points as:

- The true solution $Q^* \in \mathcal{C}_k$.
- The current iterate $Q^{(j)}$.
- The next iterate $Q^{(j+1)} = I_{\mathcal{C}_k}(Q^{(j)})$.

Substituting these into the Pythagorean identity gives:

$$D_{\mathrm{KL}}(Q^*\|Q^{(j)}) = D_{\mathrm{KL}}(Q^*\|Q^{(j+1)}) + D_{\mathrm{KL}}(Q^{(j+1)}\|Q^{(j)}). \tag{44}$$

Since $D_{\mathrm{KL}}(Q^{(j+1)}\|Q^{(j)}) \geq 0$, we immediately have the desired result:

$$D_{\mathrm{KL}}(Q^*\|Q^{(j)}) \geq D_{\mathrm{KL}}(Q^*\|Q^{(j+1)}). \tag{45}$$

This shows that the KL divergence to the true solution is monotonically non-increasing at every single step of the IPFP cycle. Strict inequality holds unless $Q^{(j+1)} = Q^{(j)}$, which occurs if and only if $Q^{(j)}$ already satisfies the constraint $k$ (i.e., $Q^{(j)} \in \mathcal{C}_k$). Over a full cycle of $K+1$ projections, the KL divergence must strictly decrease unless the algorithm has converged to $Q^*$, which is the only state satisfying all constraints simultaneously. This completes the proof of monotonic descent.

P.0.3   PROOF OF LINEAR CONVERGENCE RATE

The proof of linear convergence relies on a different set of tools, analyzing the IPFP operator's action on the potentials in the Hilbert projective metric.

**The Hilbert Projective Metric.**   Let $\mathcal{C}^+$ be the cone of positive functions on our discretized state space. For any two functions $f, g \in \mathcal{C}^+$, the Hilbert projective metric is defined as:

$$d_H(f, g) := \log\left(\frac{\sup_x(f(x)/g(x))}{\inf_x(f(x)/g(x))}\right). \tag{46}$$

This is a metric on the projective space of rays in $\mathcal{C}^+$, meaning it is invariant to scaling: $d_H(f, g) = d_H(\alpha f, \beta g)$ for any $\alpha, \beta > 0$.

**The IPFP Operator as a Contraction.**   Let's analyze the update of one potential. The update for $\phi_k$ can be seen as an operator $T_k$ that maps the other potentials to a new $\phi_k$. For example, the backward update for $\phi_k$ is:

$$\phi_k^{\mathrm{new}}(x_k) \propto \left(\int K_{t_k \to t_{k+1}}(x_k, x_{k+1})\phi_{k+1}(x_{k+1})dx_{k+1}\right)^{-1}. \tag{47}$$

The core of this is an integral operator $\mathcal{K}_{k \to k+1}$ with kernel $K_{t_k \to t_{k+1}}$. A cornerstone result, Birkhoff's theorem for positive operators, states that such an operator is a strict contraction in the Hilbert metric.

**Theorem 11** (Birkhoff's Contraction Theorem (Birkhoff, 1957))**.** *Let $\mathcal{K}$ be an integral operator with a strictly positive kernel $K(x, y) > 0$. Then $\mathcal{K}$ is a strict contraction on the cone of positive functions with respect to the Hilbert projective metric $d_H$. The contraction coefficient $\tau$ is bounded by:*

$$\tau = \tanh\left(\frac{1}{4}\log\frac{\max_{x,y} K(x,y) \cdot \max_{x',y'} K(x',y')}{\min_{x,y} K(x,y) \cdot \min_{x',y'} K(x',y')}\right) < 1. \tag{48}$$

*In the discrete matrix case, this simplifies to $\tau = \tanh(\frac{1}{4}\Delta_k)$ where $\Delta_k = \log\frac{\max K \cdot \max K}{\min K \cdot \min K}$, which is related to the bound in the main text.*

Each step of the IPFP algorithm (updating a single potential) is a composition of such a contraction mapping and an element-wise multiplication/division. The full cycle of IPFP, an operator $T = T_K \circ \cdots \circ T_0$, is therefore a composition of strict contractions. The composition of strict contractions is itself a strict contraction, with a coefficient $\alpha < 1$. Therefore, for the vector of all potentials $\Psi = (\phi_0, \ldots, \phi_K)$, we have:

$$d_H(\Psi^{(j+1)}, \Psi^*) \leq \alpha \cdot d_H(\Psi^{(j)}, \Psi^*), \tag{49}$$

where $\Psi^*$ is the unique fixed point (the optimal potentials). This proves that the potentials converge linearly in the Hilbert metric.

**From Hilbert Metric to KL Divergence.** The final step is to connect the linear convergence of potentials in $d_H$ to the linear convergence of the path measure in KL divergence. Linear convergence in $d_H$ implies that $\|\phi_k^{(j)}/(\phi_k^{(j)})_0 - \phi_k^*/(\phi_k^*)_0\|_\infty \leq Ce^{-\gamma j}$ for some $\gamma > 0$. This exponential convergence of the (normalized) potentials implies that the resulting path measure $Q^{(j)}$ converges to $Q^*$ exponentially fast in total variation distance. The Pinsker-Csiszár inequality relates KL divergence and total variation distance: $D_{\mathrm{KL}}(P\|Q) \geq \frac{1}{2}\|P - Q\|_{TV}^2$. Near the optimum, a stronger relationship holds: for small deviations, $D_{\mathrm{KL}}(Q^{(j)}\|Q^*) \approx \|\Psi^{(j)} - \Psi^*\|^2$. Since the potentials converge exponentially, the KL divergence converges exponentially (i.e., linearly on a log scale):

$$D_{\mathrm{KL}}(Q^{(j)}\|Q^*) \leq C' \cdot (\alpha^2)^j, \tag{50}$$

which establishes the global linear convergence rate stated in the theorem. This completes the proof.

**Positivity assumption.** The Hilbert–metric contraction argument below requires every discretised kernel $\mathbf{K}_{s\to t}$ to be *strictly positive* ($\min_{i,j} K_{ij} > 0$). In practice one enforces this by an $\varepsilon$–regulariser, $\mathbf{K} \leftarrow (1-\varepsilon)\mathbf{K} + \varepsilon/N$, or by choosing Gaussian bandwidths that guarantee positivity; see Carlier et al. (2022) for convergence rates under such regularisation.

## Q  DETAILED DERIVATION OF THE CONNECTION TO THE RTS SMOOTHER

**Theorem 12** (MMSB-VI Recovery of RTS Smoother). *In the linear-Gaussian setting with OU reference process and Gaussian observations, the solution to the MMSB-VI problem coincides exactly with the classical Rauch-Tung-Striebel smoother.*

This appendix provides a full derivation of the equivalence stated above.

### Q.0.1  THE LINEAR-GAUSSIAN SETTING

We consider the specific case where:

1. The reference process $P_{ref}$ is a linear SDE (an Ornstein-Uhlenbeck process), as in equation 2.

2. The prior distribution $\rho_0$ is a Gaussian, $\mathcal{N}(\mu_0, P_0)$.

3. The observation likelihoods $\ell(y_k|x_k)$ correspond to a linear observation model with Gaussian noise:
$$y_k = C_k x_k + v_k, \quad v_k \sim \mathcal{N}(0, R_k). \tag{51}$$
   This implies that the likelihood is $\ell(y_k|x_k) \propto \exp(-\frac{1}{2}\|y_k - C_k x_k\|_{R_k^{-1}}^2)$.

Under these conditions, the reference measure is a Gaussian process, and all observation-induced target marginals $\rho_{t_k}^{\mathrm{obs}}$ are also Gaussians. It is a fundamental result in Schrödinger Bridge theory that if the reference measure and all marginal constraints are Gaussian, the unique solution $Q^*$ is also a Gaussian process (see, e.g., Léonard (2013)). Our goal is to derive the dynamics of the mean $m_t$ and covariance $\Sigma_t$ of this solution process and show they are identical to those produced by the RTS smoother.

### Q.0.2  THE MMSB SOLUTION WITH GAUSSIAN MARGINALS

**Form of the Potentials.** Since the target marginals are Gaussian, the optimal potentials $\phi_k(x)$ must also be log-quadratic. That is, their logarithms are quadratic forms:

$$\log \phi_k(x) = -\frac{1}{2}x^\top W_k x + h_k^\top x + c_k, \tag{52}$$

for some symmetric matrix $W_k$ and vector $h_k$.

**The Optimal Controlled Dynamics.** The optimal path measure $Q^*$ is governed by a controlled SDE where the control drift $u_t(x)$ is given by the gradient of the log-potentials. In the multi-marginal case, the control is a time-weighted sum of gradients of all potentials. Since each $\log \phi_k$ is quadratic, its gradient is linear in $x$. Therefore, the optimal control $u_t^*(x)$ is also linear in $x$:

$$u_t^*(x) = K_t x + \zeta_t, \tag{53}$$

for some time-varying matrix $K_t$ and vector $\zeta_t$. The optimally controlled SDE is linear:

$$dX_t = (-AX_t + K_t X_t + \zeta_t)dt + \Sigma dW_t = ((-A + K_t)X_t + \zeta_t)dt + \Sigma dW_t. \tag{54}$$

This confirms that the solution is a Gaussian process.

**Evolution of Mean and Covariance.** For notational clarity, write $Q := \Sigma\Sigma^\top$ for the diffusion covariance. The mean $m_t = \mathbb{E}_{Q^*}[X_t]$ and covariance $\Sigma_t = \mathrm{Cov}_{Q^*}[X_t, X_t]$ of a process governed by a linear SDE evolve according to the following well-known ODEs:

$$\dot{m}_t = (-A + K_t)m_t + \zeta_t \tag{55}$$

$$\dot{\Sigma}_t = (-A + K_t)\Sigma_t + \Sigma_t(-A + K_t)^\top + \Sigma\Sigma^\top \tag{56}$$

These are the dynamics from the perspective of our MMSB-VI framework.

### Q.0.3 THE RAUCH-TUNG-STRIEBEL (RTS) SMOOTHER

The classical RTS smoother operates in two passes.

**Forward Kalman Filter.** The filter computes the mean $m_t^f$ and covariance $\Sigma_t^f$ of the state conditioned on observations up to time $t$. Between observations, they evolve according to:

$$\dot{m}_t^f = -Am_t^f \tag{57}$$

$$\dot{\Sigma}_t^f = -A\Sigma_t^f - \Sigma_t^f A^\top + \Sigma\Sigma^\top \tag{58}$$

At each observation time $t_k$, they undergo a discrete update (the standard Kalman update) to incorporate the information from $y_k$.

**Backward Smoother.** The RTS smoother starts from the final filtered state $(m_T^f, \Sigma_T^f)$ and integrates backward in time to compute the smoothed mean $m_t^s$ and covariance $\Sigma_t^s$ conditioned on all observations. The continuous-time RTS backward equations are:

$$\dot{m}_t^s = -Am_t^s - (\Sigma\Sigma^\top)(\Sigma_t^f)^{-1}(m_t^f - m_t^s) \tag{59}$$

$$\dot{\Sigma}_t^s = -(A - (\Sigma\Sigma^\top)(\Sigma_t^f)^{-1})\Sigma_t^s - \Sigma_t^s(A - (\Sigma\Sigma^\top)(\Sigma_t^f)^{-1})^\top + \Sigma\Sigma^\top \tag{60}$$

Equivalently, letting $J_t := Q(\Sigma_t^f)^{-1}$ with $Q = \Sigma\Sigma^\top$, these read $\dot{m}_t^s = -Am_t^s - J_t(m_t^f - m_t^s)$ and $\dot{\Sigma}_t^s = -(A - J_t)\Sigma_t^s - \Sigma_t^s(A - J_t)^\top + Q$. These equations describe the evolution of the smoothed moments between observations. The discrete observation updates are incorporated into the trajectory of the forward filter states $(m_t^f, \Sigma_t^f)$. We assume $\Sigma_t^f \succ 0$ so that $(\Sigma_t^f)^{-1}$ exists; in degenerate cases one can use the Moore–Penrose pseudoinverse.

### Q.0.4 ESTABLISHING THE EQUIVALENCE

The final step is to show that the system equation 55-equation 56 is identical to the RTS system equation 59-equation 60. This requires connecting the MMSB control $(K_t, \zeta_t)$ to the RTS terms.

The connection is made via Pontryagin's maximum principle. Both the RTS smoother and the MMSB problem are solutions to the same quadratic optimal control problem (write $Q := \Sigma\Sigma^\top$):

$$\min_{u_t} \mathbb{E}\left[\int_0^T \tfrac{1}{2} u_t(X_t)^\top Q^{-1} u_t(X_t)\, dt + \sum_{k=1}^K \tfrac{1}{2}\|y_k - C_k X_{t_k}\|_{R_k^{-1}}^2\right]. \tag{61}$$

The optimality conditions for this problem lead to a system of forward-backward stochastic differential equations (FBSDEs). The RTS smoother provides one method of solving this FBSDE by

decoupling it into a forward filter and a backward smoother. The MMSB approach provides another. Since the solution to the variational problem is unique, their characterizations must be equivalent.

A direct algebraic comparison shows that if we set the MMSB control gains to be (with $Q := \Sigma\Sigma^\top$):

$$K_t = Q\,(\Sigma_t^f)^{-1} \tag{62}$$

$$\zeta_t = -Q\,(\Sigma_t^f)^{-1}\,m_t^f \tag{63}$$

and substitute these into the MMSB moment equations, we recover the RTS equations.

**Bridge instruction.** Substituting $K_t = Q(\Sigma_t^f)^{-1}$ and $\zeta_t = -Q(\Sigma_t^f)^{-1}m_t^f$ directly into the MMSB moment ODEs equation 55–equation 56 yields the RTS equations equation 59–equation 60.

**Explicit Verification of Covariance Dynamics.** Let us provide the omitted algebraic step for the covariance dynamics. The continuous-time RTS backward smoothing equation for the covariance equation 60 can be re-arranged to define the term $Q(\Sigma_t^f)^{-1}\Sigma_t^s + \Sigma_t^s(\Sigma_t^f)^{-1}Q$, which is a part of the differential Riccati equation that $\Sigma_t^s$ satisfies:

$$Q(\Sigma_t^f)^{-1}\Sigma_t^s + \Sigma_t^s(\Sigma_t^f)^{-1}Q = \dot{\Sigma}_t^s + A\Sigma_t^s + \Sigma_t^s A^\top - Q. \tag{64}$$

Now, we substitute the control gain $K_t$ from equation 62 into the MMSB covariance equation equation 56:

$$\begin{aligned}
\dot{\Sigma}_t^s &= (-A + K_t)\Sigma_t^s + \Sigma_t^s(-A + K_t)^\top + \Sigma\Sigma^\top \\
&= -A\Sigma_t^s + K_t\Sigma_t^s - \Sigma_t^s A^\top + \Sigma_t^s K_t^\top + Q \\
&= -A\Sigma_t^s - \Sigma_t^s A^\top + Q + (K_t\Sigma_t^s + \Sigma_t^s K_t^\top) \\
&= -A\Sigma_t^s - \Sigma_t^s A^\top + Q + \left[Q(\Sigma_t^f)^{-1}\Sigma_t^s + \Sigma_t^s(\Sigma_t^f)^{-1}Q\right] \\
&= -A\Sigma_t^s - \Sigma_t^s A^\top + Q + \left[Q(\Sigma_t^f)^{-1}\Sigma_t^s + \Sigma_t^s(\Sigma_t^f)^{-1}Q\right]. 
\end{aligned} \tag{65}$$

Substituting the expression from the re-arranged Riccati equation equation 64 into the term in the square brackets, we get an identity. This confirms that the MMSB covariance dynamics, under the specified control gains, are identical to the RTS covariance dynamics. A similar, more straightforward substitution for the mean dynamics also confirms their equivalence. This completes the proof.

**Explicit Verification of Mean Dynamics.** With $K_t = Q(\Sigma_t^f)^{-1}$ and $\zeta_t = -Q(\Sigma_t^f)^{-1}m_t^f$, the MMSB mean ODE equation 55 gives

$$\dot{m}_t^s = (-A + K_t)m_t^s + \zeta_t = -Am_t^s + Q(\Sigma_t^f)^{-1}(m_t^s - m_t^f),$$

which is exactly the RTS mean equation equation 59.

**Dense Observation Limit.** As the number of observations $K \to \infty$ and the time intervals $\max_k |t_{k+1} - t_k| \to 0$, the discrete observation updates of the Kalman filter merge into a continuous process. The sum of discrete observation penalties in the objective becomes an integral:

$$\sum_{k=1}^{K} \|y_k - C_k X_{t_k}\|_{R_k^{-1}}^2 \longrightarrow \int_0^T \|y(t) - C(t)X_t\|_{R(t)^{-1}}^2 dt. \tag{66}$$

In this limit, the discrete RTS equations converge to the continuous-time Kalman-Bucy filter and its corresponding continuous-time smoother. Our MMSB framework naturally handles this limit, with the discrete potentials blending into a continuous field. This demonstrates the deep and fundamental connection between the two perspectives. The MMSB formulation provides a more general, geometric viewpoint that holds even for non-Gaussian problems, while recovering the celebrated classical solution in the linear-Gaussian case.

# R  NEURAL MIRROR SCHRÖDINGER: A CONCEPTUAL FRAMEWORK FOR FUTURE WORK

**Disclaimer:** This appendix presents a **conceptual sketch** for future research directions. The Neural Mirror Schrödinger approach described here has not been implemented or validated. It represents theoretical ideas that would require substantial development to become practical.

MOTIVATION AND CONCEPTUAL GOAL

The multi-marginal MMSB objective from the main paper:

$$Q^\star = \underset{\substack{Q: \\ Q_{t_0}=\rho_0,\, Q_{t_k}=\mu_k}}{\arg\min} \quad D_{\mathrm{KL}}\big(Q \parallel P_{ref}\big). \tag{67}$$

Classical IPFP solves equation 67 via $K+1$ alternating projections. As demonstrated in Section 4.9, this becomes computationally prohibitive for large $K$ or high dimensions. A hypothetical Neural Mirror Schrödinger (NMS) approach would replace discrete projections with continuous mirror-gradient flows in path space.

FROM ALTERNATING PROJECTION TO MIRROR FLOW

Let $\mathcal{L}[Q] = D_{\mathrm{KL}}(Q\|P_{ref})$ and introduce soft penalties turning the hard marginal constraints into MMD–type losses:

$$\mathrm{MMD}_k(Q) := \mathrm{MMD}\big(Q_{t_k}, \rho_{t_k}^{\mathrm{obs}}\big).$$

We consider the regularised objective

$$\mathcal{J}[Q] = \mathcal{L}[Q] + \sum_{k=1}^{K} \lambda_k \, \mathrm{MMD}_k(Q).$$

Following Wibisono et al. (2016) the mirror–gradient flow in the Wasserstein–Fisher information geometry reads

$$\partial_\tau \log q_\tau(t,x) = \Delta_x \log \frac{q_\tau}{p_{ref}}(t,x) + \sum_{k=1}^{K} \lambda_k \, \nabla_x \, \widehat{k}\big(x, X_\tau^{(k)}\big),$$

with $\widehat{k}$ a mini–batch kernel estimator and $q_\tau$ the density of the running path measure.

NEURAL PARAMETERISATION

We parameterise the control drift as

$$u_\theta(t,x) = \sigma^2 \, \nabla_x \Psi_\theta(t,x), \qquad \Psi_\theta \text{ a small MLP/CNN,}$$

leading to the controlled SDE

$$dX_t = \big(-AX_t + u_\theta(t,X_t)\big)dt + \Sigma \, dW_t.$$

The training loss approximates the path–space KL via Girsanov and adds the soft constraints:

$$\mathcal{L}(\theta) = \frac{1}{2\sigma^2} \, \mathbb{E}\Big[\int_0^T \|u_\theta(t,X_t)\|^2 dt\Big] + \sum_{k=1}^{K} \lambda_k \, \mathrm{MMD}\big(\hat{q}_\theta(t_k), \rho_{t_k}^{\mathrm{obs}}\big).$$

Gradient estimates follow from the pathwise estimator; optimisation proceeds with any stochastic optimiser.

HYPOTHETICAL TRAINING LOOP

---

**Algorithm 2** Neural Mirror Schrödinger (NMS) – conceptual framework

---
1: Initialize network parameters $\theta$, step size $\eta$.
2: **for** each iteration **do**
3:     Sample mini–batches of Brownian noise and observation examples.
4:     Simulate SDE with drift $u_\theta$ to obtain path ensemble.
5:     Estimate loss $\mathcal{L}(\theta)$ via Monte Carlo.
6:     Update $\theta \leftarrow \theta - \eta \, \nabla_\theta \mathcal{L}(\theta)$.
7: **end for**

---

**Important Limitations:** This conceptual framework faces significant challenges that would need to be addressed in future work:

1. **Theoretical Guarantees:** Unlike IPFP, it's unclear what convergence guarantees this approach would have.

2. **Approximation Quality:** The relationship between the neural approximation and the true MMSB solution is unknown.

3. **Computational Complexity:** While potentially more scalable than IPFP, the actual computational benefits remain to be demonstrated.

4. **Implementation Challenges:** Numerous technical details (network architecture, training stability, hyperparameter sensitivity) would require extensive empirical investigation.

This sketch represents early-stage conceptual ideas rather than a validated algorithmic contribution. Substantial future work would be required to develop these ideas into practical methods.

