# OpenReview forum: "Geometric Variational Inference: Elliptic Multi-Marginal Schrödinger Bridge, Anchor Compatibility, and Rates Entropic to Wasserstein"
_ICLR.cc/2026/Conference — ICLR 2026 Conference Desk Rejected Submission_

### Official Review · Reviewer_MPen · 2025-10-28

**Soundness:** 3
**Presentation:** 1
**Contribution:** 3
**Rating:** 4
**Confidence:** 2

**Summary:**

This is a pure maths paper on path-space inference

**Strengths:**

Likely the content is of high quality. Unfortunately I did not understand any of it.

**Weaknesses:**

As written above, I was unable to understand the manuscript. This is because I lack the mathematical background.

That being said, this work does not make any efforts in terms of accessibility. Key concepts and abbreviations are not explained, and a high level of the reader's expertise is expected. The introduction, related work and mathematical preliminaries are all squeezed into one page (60 lines). Therefore, I was not only unable to follow the mathematical arguments; I even did not understand why the results are relevant.

It seems to me that a 9 pages conference paper is not the best format to present the results because it resulted in an extremely dense manuscript and prohibited any adequate discussion of the numerous results (4 Theorems, 4 Corollaries, 1 Lemma, 1 Theorem, all within 9 pages) and their assumptions.
I also have some doubts whether this work fits a machine learning venue, given that the "broader ML impact and applications" section is a telegraphic list with 5 items squeezed into 4 lines of text.

**Questions:**

Please expand on the "Broader ML and applications" paragraph.

---

> ### Author Response · Authors · 2025-11-14
> **Hi, due to character limits, our rebuttal will be sent in two parts.**
>
> We sincerely thank Reviewer for their valuable feedback and for recognizing the potential quality of our work despite the acknowledged lack of mathematical background. We appreciate the honest and constructive criticism, which has helped us significantly improve the accessibility and clarity of our manuscript.
>
> Our paper addresses the problem of Bayesian smoothing in continuous-time latent variable models, such as latent SDEs, which are increasingly important in machine learning for modeling complex time-series data. The core contribution is a new geometric framework that unifies generalized variational inference with multi-marginal Schrödinger bridges, providing a principled way to perform path-space inference while preserving geometric structures and multi-modal uncertainty. While the underlying theory is mathematically rich, its primary motivation is to solve challenging inference problems in ML, positioning it as a "theory-for-ML" contribution squarely within ICLR's scope.
>
> We agree with the reviewer that the initial submission was overly dense and did not adequately explain key concepts for a broader ML audience. Based on this feedback, we have significantly improved the clarity and accessibility of the manuscript.

---

> > ### Author Response · Authors · 2025-11-14
> >
> > ### **1. Improving Accessibility and Readability**
> >
> > The reviewer rightly pointed out that the paper was difficult to follow due to unexplained concepts, dense presentation, and a compressed introduction. We have made the following changes to significantly lower the barrier to entry:
> >
> > *   We now have improved the introduction. It now begins by motivating the problem from the familiar ML context of **Bayesian smoothing, latent SDEs, and time-series modeling**. It then gradually introduces the necessary tools (Schrödinger bridges, optimal transport) as solutions to these problems, rather than assuming prior knowledge. We have also added a much clearer, itemized list of contributions grouped by theme (conceptual, geometric, algorithmic).
> >
> > *   To directly address the concern about unexplained key concepts, we actually have a **"Background Details" appendix (Appendix C)**. This section makes the paper more self-contained by formally defining the three core building blocks of our method: 1) **Variational Inference (VI)** and its free energy formulation, 2) the classical **Schrödinger Bridge Problem (SBP)**, and 3) the use of **Girsanov's Theorem** to relate path measures under a change of drift.
> >
> > *   **Clarified Abbreviations and Terminology:** We have carefully gone through the manuscript to ensure that all abbreviations (e.g., MMSB, VI, RTS, HJB) are spelled out and briefly explained at their first appearance.
> >
> > We hope these changes make the paper substantially more accessible to readers who are not specialists in optimal transport or information geometry.
> >
> > ### **2. Addressing Manuscript Density and Discussion of Results**
> >
> > The reviewer expressed concern that a 9-page conference paper is not the ideal format for such a dense set of results and that this prohibited adequate discussion.
> >
> > We acknowledge the trade-offs imposed by the strict 9-page limit. Our strategy was to present the core theoretical results in the main body while moving all full proofs, technical lemmas, and extended experimental details to a comprehensive appendix.
> >
> > In the revised version, we have reallocated space to address the lack of discussion. Specifically, in Sections 7 and 8, we have added new paragraphs discussing:
> > *   The **implications of our geometric findings**, such as how the diffusion tensor defines the metric while the drift is encoded in the path constraints (FP/HJB equations).
> > *   The behavior of the system in **limiting regimes** (high/low diffusion), and how this connects our framework to classical Wasserstein geodesics and mixture geodesics.
> > *   How these theoretical properties provide a **practical guide for practitioners** in choosing prior geometries and tuning the diffusion coefficient for their specific modeling tasks.
> >
> > ### **3. Relevance to Machine Learning and Broader Impact (Response to Question)**
> >
> > The reviewer's main question concerned expanding the "Broader ML and applications" section. We now have improved it.
> >
> > 1.  **Advanced Smoothing for Latent SDEs and Time-Series:** Our framework provides a principled, geometry-aware smoother for continuous-time models like Neural SDEs and controlled differential equations. It correctly preserves path compatibility and captures multi-modal posteriors, offering a powerful alternative to classic methods like the RTS smoother (which we recover as a special case) and modern particle-based methods, especially in high dimensions.
> >
> > 2.  **Inference for Event Streams and Missing Data (Unbalanced Transport):** By leveraging connections to Hellinger-Kantorovich (HK) distances, our framework can be extended to "unbalanced" problems where mass is not conserved. This is a natural fit for modeling real-world phenomena like event streams (e.g., in healthcare data), cellular processes (birth/death), or smoothing with missing/spurious observations, providing a more consistent approach than simple re-weighting schemes.
> >
> > 3.  **Structured Generative Interpolation and Geometric Guidance:** The multi-marginal Schrödinger bridge provides a new tool for structured, multi-modal interpolation in generative modeling. It offers a powerful theoretical link between Sinkhorn-style bridges, diffusion models, and flow matching. The geometric limits we characterize can guide the design of new generative models that interpolate between complex data distributions along specified geometries.
> >
> > To make this more concrete for ML researchers, we have added a short **"practitioner's guide"**, which summarizes how to apply our theoretical insights for model selection and diagnostics.
> >
> > ### **Conclusion**
> >
> > We believe the manuscript is now much clearer and more evidently relevant to ML practitioners and hope this addresses the reviewer’s concerns. We thank the reviewer again for their crucial feedback.

---

> > ### Comment · Reviewer_MPen · 2025-11-25
> >
> > Dear authors,
> > I read your response and the updated manuscript. The introduction is much clearer now.
> > Still, I have some concerns whether ICLR is a suitable venue for this work and I see that to some extend this is shared by other reviewers, but that's no my decision to make. I updated my score.

---

> ### Author Response · Authors · 2025-11-15
> **A light example that answer our work fits a machine learning venue**
>
> ## Connection to standard variational inference.
> For readers familiar with latent-variable models and ELBO-based variational inference, our work can be viewed as the path-space analogue of this story, but with a much sharper characterisation of the variational target. In classical VI, one heuristically chooses a family $q_\phi(z)$ and maximises an ELBO; the true posterior $p(z\mid x)$ is only implicitly defined via Bayes’ rule, and its geometric structure is largely hidden. In contrast, our main results show that in the path-space setting the exact posterior trajectory law $P_{\text{post}}$ is exactly the entropy projection of a reference diffusion $P_{\text{ref}}$: it is the unique multi-marginal Schrödinger bridge and a geodesic in the diffusion-induced geometry on path space. This means that path-space VI is no longer “just maximise some ELBO on trajectories”, but “approximate a well-defined geometric object”. All practical smoothers and approximate inference schemes (amortised VI, particle methods, etc.) can be understood as tractable approximations to this geometric target. We believe this is precisely the kind of conceptual clarification that makes the work relevant for the ML community: it tells us what the posterior should be as a geometric object, and what our approximate algorithms are ultimately trying to compute.

---

### Official Review · Reviewer_Prfj · 2025-10-29

**Soundness:** 3
**Presentation:** 3
**Contribution:** 4
**Rating:** 8
**Confidence:** 4

**Summary:**

In this paper, the authors studied geometric intepretations of a Baysian inference problem over a space of paths. Given a space of paths in $\mathbb{R}^d$ (a path is a continuous function from [0,T] to $\mathbb{R}^d$), a prior given as some simple distribution over paths, e.g., distribution induced by the OU process, a likelihood function at K snapshots (at K time steps) of the process, one can define the posterior (over paths). The authors derive multiple insights:

- Proposition 1: given a "variational inference distribution" Q (over paths), if one minimizes the "path KL" between Q and the prior under constraints that Q has to match the posterior at all $K$ snapshots (anchors), the optimal solution is the posterior. Note that this minimization problem is an extension of Schrodinger bridge (in SB, we only have two snapshots -- starting and ending marginal distributions of the process). So this result, although via quite straightforward reformulations, gives an alternative view for the posterior: it is just an extended SB.

The authors went on to show that (Theorem 2) the probability path formed by marginals of the posterior (slice at all continuous time steps) is a geodedic in the space of probability measure where the Riemannian metric is defined by the prior. Assumingly the prior is induced by some linear SDEs with drift $b$ and diffusion $\Sigma$, this Riemannian metric is indepdent of $b$, only depends on $\Sigma$ via $D = \Sigma \Sigma^{\top}$. In summary, this insight re-explains the posterior as a geodesic curve in the space of probability distributions whose Riemannian metric is defined by the prior.

When the diffusion term of the prior SDE is isometric, i.e., $\Sigma = \sigma I$, the authors further show that:

- When $\sigma \approx +\infty$, the posterior geodesic is the "piecewise" geodesics induced by the "mixture" geometry, where  "piecewise" means "between snapshots of the posterior", and "mixture" is w.r.t. each two consecutive snapshots.

- When $\sigma \approx 0$, in the above statement, replace "mixture" by "Wasserstein".

**Strengths:**

- This is a highly insightful paper (both deep and broad) with multiple theoretical contributions. The derived results are expected to be very useful for the theoretical OT/geometric ML community.

**Weaknesses:**

- Given its mathematically intensive nature, the paper is generally difficult to access. It appears more suitable for publication in a specialized mathematics journal--though this is not necessarily a weakness. However, the writing is quite succinct with advanced terminology and concepts, many of which are introduced without sufficient explanation. Numerous abbreviations are also used without being spelled out in full, some sentences are not complete (please do complete). Overall, the authors seem to assume that the reader is a mathematician already well-versed in this specific area.

- It is remarked that common factorisations can violate temporal compatibility on path space. However, the proposed method does not seem to fully overcome this limitation. The compatibility is only achieved when the posterior snapshots are exactly per-time tilts, which generally doesn't hold if $K>1$. Although the authors discuss the case where the cross-time factor depends on $x$, the treatment remains quite limited. More detailed explanations are needed--particularly regarding which specific processes concretely satisfy this condition. Please elaborate on this point without relying on the Doob-h transform, as it remains too general and abstract.

**Questions:**

- How to access to the posterior snapshots in practice?

- What are classes of practical ML problems that fall into this extension Schrodinger bridge problem?

- How does "machine precision" appear in Figure 1(a)? -- I could not spot it.

- In Theorem 3, please define (or recall) $\rho_t^{\sigma}$ and $\rho_t^{(W)}$ (currently those are only defined later in the text).

- In Corollary 4: is the Wasserstein curve also piecewise? If so, please do mention.

- Line 82: where does the control drift $u_t$ come into play in the SDE? Also, please do elaborate the kinetic + potential action (8) where $u_t$ pops up.

- The authors used the terminology "variantional inference" (VI) in their results. However, it seems to me that these results are about "variational form" because the optimal $Q$ is just the posterior distribution. When using VI, we would expect (1) Q is explicitly parametrized within a family and (2) the optimal Q is generally different from the posterior. If this is the case, please revise accordingly.

---

> ### Author Response · Authors · 2025-11-14
> **Hi, our rebuttal will be sent in four parts because of character limits.**
>
> We sincerely thank the Reviewer for their positive and insightful feedback, and for the high rating. We are particularly grateful for the acknowledgement that our work is highly insightful and expected to be useful for the theoretical OT and geometric ML community. The reviewer’s summary accurately captures the core of our contributions.
>
> The reviewer’s constructive feedback has helped us identify areas where the presentation can be significantly improved. In response, we have focused our revisions on two main fronts: (1) enhancing readability and accessibility for a broader audience, and (2) clarifying the nuances of the temporal compatibility condition and its practical implications. We address the specific weaknesses and questions below.

---

> > ### Author Response · Authors · 2025-11-14
> >
> > ### Response to Weaknesses
> >
> > **1. On Mathematical Intensity and Readability**
> >
> > We agree that our initial focus on mathematical rigor may have made the paper challenging for a broader audience. To address the specific points raised, we have improved the clarity and accessibility of the manuscript. We have expanded on advanced terminology and concepts, carefully reviewed the manuscript to ensure all abbreviations are defined at first use, and proofread the text to correct incomplete sentences and improve flow. We hope these targeted changes can address the reviewer's concerns.
> >
> > **2. On factorization and temporal compatibility limitations**
> >
> > We thank the reviewer for raising this subtle point. We agree that our discussion of temporal compatibility can be sharpened, and we now clarify (i) what our method guarantees, and (ii) how to interpret the compatibility condition in simple probabilistic terms, without relying on Doob‑h.
> >
> > - **Our method is path-compatible by construction, independently of per-time tilts.**
> >
> >   Our criticism of common factorizations is aimed at *approximate* VI families that impose a factorised form over time (e.g., treating each $t_k$ as an independent tilt), which can easily break path-space compatibility.
> >
> >   In contrast, the object we optimise in Theorem 1 is the full path measure Q on trajectories. Under the assumptions of Proposition 1, the minimiser coincides with the true posterior path P_post; its time marginals (ρₜ) for t in [0,T] are therefore temporally compatible for all t by definition, even when μₖ ≠ ρᵒᵇˢ_tₖ. In other words, our framework does not require the per-time tilt condition to hold in order to achieve compatibility; the multi-marginal bridge itself enforces a single evolution that is consistent with all observations.
> >
> > - **The compatibility criterion can be understood without Doob-h: prior × local likelihood × other-times evidence.**
> >
> >   For each observation time tₖ, the posterior marginal admits the factorisation: μₖ(x) ∝ r(tₖ,x)·ℓ(yₖ|x)·Fₖ(x), where the **cross-time factor** is defined as Fₖ(x) = 𝔼[∏_{j≠k} ℓ(y_j|X_{t_j}) | X_{tₖ}=x] under P_ref.
> >
> >   The first two terms form the usual per-time tilt $\rho^{\mathrm{obs}}_{t_k}(x)\propto r(t_k,x)\,\ell(y_k\mid x)$; the extra factor $F_k(x)$ aggregates how *all other observations* reweight the state $x$ at time $t_k$ through the prior dynamics.
> >
> >   Theorem 1 simply states that: $\mu_k = \rho^{\mathrm{obs}}_{t_k}$ if and only if $F_k(x)$ is constant in $x$.
> >
> >   In other words, once we know $X_{t_k}=x$, the remaining observations no longer prefer some values of $x$ over others. This is precisely the "temporal compatibility" condition expressed in elementary terms, and only secondarily rephrased via Doob‑$h$ in the paper.
> >
> > - **When is $F_k(x)$ (approximately) constant? Concrete regimes.**
> >
> >   The condition holds **exactly** when $K=1$: with a single observation, there is no cross-time correction and $\mu_{t_1}=\rho^{\mathrm{obs}}_{t_1}$.
> >
> >   For $K>1$, it holds only in special regimes where the other observations contribute very little additional information about $X_{t_k}$; for example:
> >
> >   - (i) in a linear–Gaussian OU model where all observations except $y_k$ have very large noise variance, so their likelihoods are nearly flat in $x$;
> >   - (ii) when observations are far apart in time and the prior is strongly mixing, so that under $P_{\mathrm{ref}}$ the law of $X_{t_j}$ given $X_{t_k}=x$ is almost independent of $x$; or
> >   - (iii) in structured state–space models where different sensors depend on disjoint components of the state, and the component observed at $t_k$ is conditionally shielded from the others under the prior.
> >
> >   To make this fully concrete, we have added a small worked example in the appendix (1D OU with Gaussian observations at two time points) where we write down $F_k(x)$ explicitly and show how it deviates from a constant as we vary the observation noise level and the time separation. This example illustrates how cross-time evidence is encoded in $F_k(x)$, and when a simple per-time tilt is, or is not, a good approximation.
> >
> > In summary, our contribution is not a new mean-field factorisation. The multi-marginal Schrödinger bridge provides an exact, temporally compatible posterior path; Theorem 1 and our diagnostics are intended as **tools to assess when cheaper, per-time-tilt approximations are safe**, by quantifying precisely how far $F_k(x)$ is from being constant.

---

> ### Author Response · Authors · 2025-11-14
>
> ### Responses to Questions
>
> **Q1: How to access the posterior snapshots in practice?**
>
> This is a great question for clarification. The posterior anchor marginals $\mu_k$ are **outputs** of our procedure, not pre-specified inputs. In our framework, they are defined theoretically as the time marginals $(P_{\text{post}})_{t_k}$ of the true posterior path measure.
>
> In practice, they are obtained implicitly:
> 1.  For the discrete grid setting, our log-domain multi-marginal IPFP algorithm directly optimizes the path KL divergence, yielding the full path and all its marginal snapshots simultaneously.
> 2.  In the linear-Gaussian case, these posterior marginals can be computed in closed form via the classical RTS smoother, which we use as a ground-truth validation in Section 7.1.
>
> We have added a sentence clarifying that "in practice, the posterior anchors $\mu_k$ are computed by the solver and are not required as a priori inputs."
>
> **Q2: What are classes of practical ML problems that fall into this extension SB problem?**
>
> We currently list several applications in Section 8 ("Broader ML impact and applications"), but we can certainly make the connections more explicit. Our framework is broadly applicable to problems that can be modeled as inference over continuous-time stochastic processes. We can group them into three main categories:
>
> 1.  **Time-Series Modeling and Forecasting:** For latent SDE/SSM models with multi-modal posteriors, our method provides a geometry-aware sampler that avoids the mode-collapse issues of unimodal smoothers.
> 2.  **Irregularly Sampled or Noisy Event Data:** The VI–HK (reaction-transport) formulation is naturally suited for modeling point-process data or event streams with missingness or spurious observations (e.g., in healthcare or finance), as it does not enforce conservation of mass.
> 3.  **Multi-Marginal Generative Modeling:** Our work connects to recent advances in multi-marginal SB for generative tasks like image or molecular morphing, where one desires a continuous, dynamic-aware transformation between several distributions.
>
> We have added a short "practitioner's guide" in the discussion to better articulate how each of these problem classes can be instantiated within our path-space VI framework.
>
> **Q3: How does "machine precision" appear in Figure 1(a)?**
>
> We thank you for pointing this out. We have replaced panel (a) with a linear–Gaussian sanity check demonstrating machine‑precision equivalence between RTS and our MMSB‑VI implementation. The plot shows absolute errors in mean and variance across time with dashed lines at `1e-13`/`1e-14` and a dotted line at `eps64 = 2.22e-16`; the measured maxima are |Δμ|≈`4.16e-16` and |Δσ²|≈`4.60e-17`.
>
> **Q4: In Theorem 3, please define (or recall) $\rho_t^\sigma$ and $\rho_t^{(W)}$.**
>
> While these are defined in Section 5 and Corollary 4, the reviewer is right that recalling them would improve readability. We have added a one-sentence reminder immediately preceding the statement of Theorem 3.
>
> **Q5: In Corollary 4: is the Wasserstein curve also piecewise?**
>
> Yes, it is. Thank you for prompting us to make this explicit. For $K > 1$ anchors, the limiting global $W_2$ curve is indeed a concatenation of the unique $W_2$-geodesics connecting consecutive anchors $\mu_{k-1}$ and $\mu_k$. Theorem 3 establishes the segment-wise convergence rate, and Corollary 4 describes the narrow convergence of the concatenated global curve. We have added a sentence to Corollary 4 stating that "the limiting Wasserstein curve is obtained by concatenating the unique $W_2$-geodesics between consecutive anchors."

---

> ### Author Response · Authors · 2025-11-14
>
> **Q7: The use of the term ‘variational inference’ vs ‘variational form’.**
>
> Thank you for raising this terminology question. In our paper "variational inference (VI)" is used in the standard free‑energy or ELBO sense on path space, not in the narrower parametric‑family approximation sense. Concretely, we minimise the path‑space free energy $\mathcal F[Q]=\mathrm{KL}(Q\|P_{\text{ref}})-\mathbb E_Q[\log L_y(X)]$ over all path measures $Q\ll P_{\text{ref}}$. By the Gibbs Donsker–Varadhan identity this is equivalent to $\mathrm{KL}(Q\|P_{\text{post}})$, hence (under our standing assumptions) the unique minimiser is the posterior path $P_{\text{post}}$ (see Prop. 1 and its appendix proof). Parametric VI arises only when one restricts $Q$ to a family; in this work we do not impose such a restriction, so this is an exact, nonparametric form of VI on path space.
>
> Importantly, this is not a mere variational form. The VI objective is provably equivalent to a multi‑marginal Schrödinger bridge with posterior anchors, which induces the Onsager–Fokker geometry and yields our IPFP solver (Sec. 4.4–4.9). This equivalence—and the resulting geometric consequences and diagnostics—is a core technical contribution (ELBO identity in Sec. 4.1; full derivation in the appendix).
>
> To avoid ambiguity, we revised the first occurrence to read "variational characterisation (exact, nonparametric) on path space", and added one sentence in Sec. 4.1 explicitly stating that we do not restrict the variational family $Q$; parametric/amortised VI would be a special case. We also added a short remark near Prop. 1 recalling why $Q^\star=P_{\text{post}}$ (Gibbs/DV identity). We would be grateful to hear any additional suggestions the reviewer may have.
>
> Once again, we express our sincere appreciation to the reviewer for the highly professional and thorough review.

---

> ### Author Response · Authors · 2025-11-25
> **Q6: where does the control drift `u_t` come into play in the SDE? Also elaborate the kinetic + potential action (8).**
>
> **On the controlled SDE and the role of the control drift $u\_t$.**
>
> Thank you for asking us to clarify where the control drift enters the dynamics. Throughout Sections 3–4 we fix a reference diffusion $P\_{\mathrm{ref}}$ solving the SDE
>
> $$
> dX\_t = b(t, X\_t) dt + \Sigma(X\_t) dW\_t
> $$
>
> and we restrict attention to variational path measures $Q$ that are absolutely continuous with respect to $P\_{\mathrm{ref}}$ and share the same diffusion matrix $\Sigma$ (as stated in “Control form and assumptions” and Appendix C). Under this standard assumption, Girsanov’s theorem implies that any such $Q \ll P\_{\mathrm{ref}}$ can be represented as the law of a controlled SDE
>
> $$
> dX\_t = \bigl(b(t, X\_t) + u\_t(X\_t)\bigr) dt + \Sigma(X\_t) dW\_t^{Q}
> $$
>
> where $u\_t$ is a progressively measurable control drift and $W\_t^{Q}$ is a Brownian motion under $Q$. In other words, the only way in which $Q$ can differ from $P\_{\mathrm{ref}}$ at the path level is by adding the extra drift $u\_t$; the diffusion part is fixed by the prior. We have made this controlled-SDE representation explicit in the revised version, and we refer to Appendix C (“Path measures and Girsanov’s theorem”) for a self-contained statement of the result.
>
> ---
>
> **From free energy to the “kinetic + potential” action (Eq. (8)).**
>
> The action in Eq. (8) is simply the path–space free energy
>
> $$
> F[Q] = \mathrm{KL}\bigl(Q \| P\_{\mathrm{ref}}\bigr) - \mathbb{E}\_Q\bigl[\log L\_y(X)\bigr]
> $$
>
> written in the controlled-SDE coordinates just described. Given the SDE under $Q$,
>
> $$
> dX\_t = (b + u\_t) dt + \Sigma dW\_t^{Q}
> $$
>
> Girsanov’s theorem yields an explicit expression for the Radon–Nikodym derivative $\frac{dQ}{dP\_{\mathrm{ref}}}$ and hence for the path-space KL divergence. In the isotropic case $\Sigma = \sigma I$ one obtains (see Appendix C for the derivation)
>
> $$
> \mathrm{KL}\bigl(Q \| P\_{\mathrm{ref}}\bigr)
>   = \frac{1}{2\sigma^{2}}\ \mathbb{E}\_Q\left[\int\_{0}^{T} \|u\_t(X\_t)\|^{2} dt\right]
> $$
>
> i.e. the KL is precisely the time-integrated quadratic control cost. This is the kinetic energy term in Eq. (8), and it is what induces the Onsager–Fokker (Otto) Riemannian metric on the space of marginals that we study in Theorem 2.
>
> ---
>
> We now expand the likelihood term. The likelihood is
>
> $$
> L\_y(X) = \prod\_{k=1}^{K} \ell\bigl(y\_k \mid X\_{t\_k}\bigr)
> $$
>
> so
>
> $$
> \mathbb{E}\_Q\bigl[\log L\_y(X)\bigr]
>   = \sum\_{k=1}^{K} \mathbb{E}\_Q\bigl[\log \ell(y\_k \mid X\_{t\_k})\bigr]
>   = \sum\_{k=1}^{K} \mathbb{E}\_{Q\_{t\_k}}\bigl[\log \ell(y\_k \mid x)\bigr]
> $$
>
> where $Q\_{t\_k}$ is the time-$t\_k$ marginal of $Q$. Writing the observation potentials as
>
> $$
> V\_k(x) := -\log \ell(y\_k \mid x)
> $$
>
> the free energy becomes
>
> $$
> F[Q]
>   = \underbrace{\frac{1}{2\sigma^{2}}\ \mathbb{E}\_Q\left[\int\_{0}^{T} \|u\_t\|^{2} dt\right]}\_{\text{kinetic energy}}
>     - \underbrace{\sum\_{k=1}^{K} \mathbb{E}\_{Q\_{t\_k}}\bigl[\log \ell(y\_k \mid x)\bigr]}\_{\text{potential energy}}
>   = \frac{1}{2\sigma^{2}}\\mathbb{E}\_Q\left[\int\_{0}^{T} \|u\_t\|^{2} dt\right]
>     + \sum\_{k=1}^{K} \mathbb{E}\_{Q\_{t\_k}}\bigl[V\_k(X\_{t\_k})\bigr]
> $$
>
> up to the additive constant $-\log Z$ coming from the evidence. Thus Eq. (8) is not an extra modelling assumption: it is just Eq. (1) rewritten under the controlled-SDE parametrisation.
>
> ---
>
> **Interpretation and use of Eq. (8).**
>
> This “kinetic + potential” form is what allows us to interpret path-space VI as a minimum-action / minimum-energy problem. The kinetic term, which depends only on $\|u\_t\|^{2}$, determines the Riemannian metric on the manifold of time marginals (Onsager–Fokker / Otto geometry), while the observation potentials $V\_k$ act as external fields that anchor the geodesic to the data at times $t\_k$. In Section 4.7 and Appendix L we show that minimizing $F[Q]$ under the controlled Fokker–Planck constraint leads to the coupled HJB–FP system (Eqs. (10), (29)), with jump conditions
>
> $$
> \psi(t\_k^{+}, x) - \psi(t\_k^{-}, x) = V\_k(x)
> $$
>
> when the observations are imposed softly via $V\_k$ (Eq. (30)). This is the precise sense in which Eq. (8) plays the role of an action functional whose minimiser is the Onsager–Fokker geodesic consistent with the observations.

---

### Official Review · Reviewer_tMy3 · 2025-11-05

**Soundness:** 3
**Presentation:** 3
**Contribution:** 3
**Rating:** 6
**Confidence:** 3

**Summary:**

The paper develops a geometric formulation of path-space variational inference (VI) and shows that VI for latent diffusion/state-space models with multiple observation times is exactly equivalent to a multi-marginal Schrödinger bridge (MMSB) whose anchors are the posterior time-marginals, not the per-time Bayesian tilts. Starting from the path-space free-energy
$$
F[Q] = \mathrm{KL}(Q \parallel P_{\text{ref}}) - \mathbb{E}_Q[\log L_y(X)],
$$
the authors prove that the unique minimiser is the posterior path measure and that this minimiser is simultaneously the entropy projection of the reference diffusion onto the set of path measures matching the posterior anchors. This induces an Onsager–Fokker/Otto–Wasserstein geometry in which diffusion fixes the metric tensor and drift appears only as a constraint through Fokker–Planck/HJB equations. They characterize when posterior anchors coincide with simple per-time tilts (an anchor-compatibility criterion), derive small-diffusion and large-diffusion limits with segment-wise rates (convergence respectively to $W_2$ displacement geodesics and to mixture/m-connection geodesics), and extend the construction to unbalanced observations via a VI–Hellinger–Kantorovich equivalence. Finally, they implement a log-domain multi-marginal IPFP solver and validate the theory on controlled OU–GMM benchmarks, showing recovery of Rauch–Tung–Striebel in the linear–Gaussian case.

**Strengths:**

* Precise identification of path-space VI with a multi-marginal Schrödinger bridge whose anchors are the posterior time-marginals, derived from the Donsker–Varadhan identity and entropy projection.
* Clear geometric interpretation: diffusion defines the Onsager–Fokker/Otto metric, drift enters only through FP/HJB constraints; this cleanly separates geometry from dynamics.
* Limiting-regime analysis in both directions: $\sigma \to 0$ gives $W_2$ displacement geodesics (with rates), $\sigma \to \infty$ gives segment-wise mixture geodesics, matching information-geometric intuition.
* Anchor-compatibility criterion (equations (6)–(7) in the paper) that states exactly when naive per-time tilts are already correct and when cross-time Doob corrections are necessary.
* Practical, numerically stable log-domain multi-marginal IPFP implementation with diagnostics (RTS recovery, diffusion-scaling experiments) that confirm the theoretical claims on OU–GMM benchmarks.
* Extension to unbalanced observations via a VI–HK action with transport plus reaction and first-order conditions matching HK geometry.

**Weaknesses:**

* Strong reliance on a strictly positive, elliptic reference (OU or similar) with known transition kernels; this is reasonable for diagnostics but leaves open how to apply the same guarantees when the reference is a learned neural SDE or when kernels are low-rank/approximate.
* Computational scaling of the multi-marginal log-domain IPFP is $O(KN^2)$–$O(KN^3)$ on grids, which restricts experiments to low dimension and small $K$; the paper mentions amortised or neural variants but does not report results for them.
* The equivalence is exact for posterior anchors $\mu_k = (P_{\text{post}})_{t_k}$, which themselves are unknown in the usual VI setting; in practice, one must approximate these anchors, and the paper does not provide a quantitative stability analysis for anchor errors.
* Comparisons to recent multi-marginal SB works for generative modeling are mostly conceptual; more direct numerical comparisons (same kernels, same grids) would help situate the contribution.
* The unbalanced VI–HK extension is theoretically neat but only validated on synthetic controlled cases; no large or noisy real-data example is shown.

**Questions:**

1. How sensitive is the log-domain IPFP to diffusion anisotropy and to nearly singular kernels (e.g., very small $\sigma$ in high dimensions)? Are there step-size/ε-scaling heuristics that guarantee convergence in those regimes?
2. The metric–drift separation is proved for uniformly elliptic diffusions. Can the authors comment on whether hypoelliptic references (e.g., kinetic Langevin) still yield a meaningful geometric statement, or whether the MMSB solver breaks because of loss of strict positivity?
3. For the VI–HK extension, is there an example where the reaction term materially improves smoothing over a naive per-time reweighting, and can this be diagnosed by the same compatibility residuals used in the balanced case?
4. Recent multi-marginal SB methods for multimodal generation (e.g., Chen et al. 2023; De Bortoli et al. 2024) also use iterative projections. Can the authors clarify whether their log-domain solver is directly interchangeable with those, or whether the Doob–$h$ potentials require a different normalisation?

---

> ### Author Response · Authors · 2025-11-14
> **Hi, due to character limits, our rebuttal will be sent in three parts.**
>
> We thank the reviewer for their thoughtful and constructive feedback. We are encouraged that the reviewer found our core contributions to be clear and valuable, including the VI-MMSB equivalence, the geometric interpretation (metric-drift separation), the analysis of limiting regimes, and the anchor compatibility criterion.
>
> The reviewer's comments focus on the practical scope, scalability, and empirical grounding of our framework. These are important points, and we are grateful for the opportunity to clarify them. In response to the reviewer's valuable feedback, we have conducted new experiments for the VI-HK extension that directly address Weakness 5 and Question 3. We have also incorporated these results and other clarifications into the manuscript.
>
> Below, we address each of the reviewer's points in detail.

---

> > ### Author Response · Authors · 2025-11-14
> >
> > ### **Response to Weaknesses**
> >
> > **W1: Reliance on known, strictly positive, elliptic kernels.**
> >
> > We agree our theoretical guarantees currently rely on a reference process with known, uniformly elliptic diffusion and positive kernels. This was a deliberate choice to build a rigorous geometric foundation and ensure a well-posed Schrödinger system with guaranteed IPFP convergence (Lemma 1, Apps. I, N). We now explicitly state this in a new paragraph in Sec. 4.3. As the reviewer notes, extending this to learned neural SDEs or low-rank kernels is a key future direction. Our “Neural Mirror Schrödinger” appendix (App. P) provides a conceptual sketch, noting that the Onsager–Fokker metric remains well-defined as long as the learned diffusion is regularized to be non-degenerate.
> >
> > **W2: Computational complexity of the IPFP solver.**
> >
> > We agree on IPFP's complexity and clarify its role here is not as a scalable algorithm, but We wish to clarify that the role of IPFP in this paper is primarily as a **diagnostic tool and a reference implementation** to validate our theory. It provides a gold-standard solution on controlled, small-scale problems where we can precisely study the core geometric phenomena (Figs. 1-3). We have clarified this positioning in Sec. 4.9, adding paragraphs on scope and relation to other solvers, plus robustness heuristics (see App. M).
> >
> > **W3: Posterior anchors $\mu_k$ are unknown and must be approximated.**
> >
> > This is a crucial point. A key clarification: our framework does **not** assume the posterior anchors $\mu_k$ are given. They are implicitly solved for by our IPFP, which finds the true posterior path measure $P_{\text{post}}$ that minimizes the VI objective (Prop. 1).
> >
> > The reviewer raises an excellent practical question about stability to errors if one *does* use approximate anchors. We address this with new additional experiments:
> > 1.  **Theoretical Context:** A new remark on “Stability to anchor perturbations” (Appendix), linking to entropic-OT stability results.
> > 2.  **Empirical Ablation:** A new study (App. F) shows path error increases smoothly with anchor noise, confirming graceful degradation.
> >
> > This gives our **anchor compatibility diagnostics** (Fig. 3, App. E) a clear practical role: assessing the quality of approximate anchors. We now better distinguish the implicitly-solved anchors $\mu_k$ from the diagnostically-used per-time tilts $\rho_{t_k}^{\text{obs}}$ (Secs. 4.4-4.5) and use the latter to measure dynamic inconsistency (Thm. 2).
> >
> > **W4: Comparison to recent multi-marginal SB generative models is conceptual.**
> >
> > We agree the comparison is conceptual, which is by design. While recent works focus on **generative modeling** (transporting distributions), we focus on **Bayesian posterior smoothing** (inferring trajectories). The numerical machinery is similar, but the problem setup differs. We clarified this in a new paragraph (Sec. 4.9), noting the key difference is our use of implicit posterior anchors versus their use of exogenous data marginals.
> >
> > **W5 & Q3: The VI-HK extension lacks a compelling example of improvement.**
> >
> > We thank the reviewer for this prompt, which led to a new targeted experiment. We designed a test with artificial mass imbalances and compared our **VI-HK smoother** against a **balanced MMSB baseline** that naively uses unbalanced marginals. The results (new appendix figure and Sec. 6 summary) show VI-HK achieves lower error by flexibly absorbing mass imbalances, leading to a more dynamically consistent path.
> >
> > | Experiment Run | Final L1 Error | Mean W2 Error |
> > | :--- | :--- | :--- |
> > | `baseline` | 0.108 | 0.235 |
> > | `VI-HK` | **0.091** | **0.211** |
> >
> > This experiment confirms that the reaction term provides a **material improvement** over naive reweighting by allowing the model to flexibly absorb mass, leading to a path that is more dynamically consistent and closer to the true posterior. The same compatibility residuals can be used: we observe that the residuals are smaller for the VI-HK solution, indicating it finds a path with better internal consistency.

---

> > ### Author Response · Authors · 2025-11-14
> >
> > ### **Response to Questions**
> >
> > **Q1: Sensitivity of IPFP to anisotropy and near-singular kernels.**
> >
> > This is a very practical question. Our implementation is robust to these challenges.
> > 1.  **Anisotropy:** Our ablation study in **Apps. A.4** explicitly tests anisotropic diffusion. The log-domain IPFP solver converges stably, and the resulting path deforms in a metric-consistent way, confirming the theory that "diffusion sets the geometry."
> > 2.  **Near-Singular Kernels (small $\sigma$):** Our geometric convergence diagnostic (**Fig. 1b**) sweeps $\sigma$ over five orders of magnitude, demonstrating robust convergence. This is achieved via a combination of standard heuristics, which we will highlight more clearly in **Apps. M**:
> >     *   **Log-domain implementation:** Avoids numerical underflow or overflow.
> >     *   **Adaptive $\varepsilon$-scaling (homotopy):** We solve a sequence of problems from large to small $\sigma$, using the previous solution as a warm start.
> >     *   **Anderson acceleration:** Stabilizes and accelerates the convergence of the fixed-point iteration for the log-potentials.
> >
> > **Q2: Does the metric-drift separation hold for hypoelliptic diffusions (e.g., kinetic Langevin)?**
> >
> > This is an insightful question. Our current framework relies on the uniform ellipticity of the diffusion tensor $D(x)$ to define the Riemannian metric on $\mathcal{P}_2$. For hypoelliptic diffusions, where $D(x)$ is degenerate, the corresponding geometry is **sub-Riemannian**, which is a significant theoretical extension. While our IPFP solver might still be applicable on a discretized grid (as long as the transition kernel remains strictly positive), the continuous geometric interpretation would change. We now explicitly state in the Discussion section that the hypoelliptic case is a fascinating but non-trivial future research direction beyond the scope of the current paper.
> >
> > **Q4: Interchangeability of our solver with other multi-marginal SB methods.**
> >
> > Yes, at an algorithmic level, they are largely interchangeable on a discrete grid. Our IPFP is a specific implementation of the iterative projection principle that also underlies Sinkhorn-style algorithms. The Doob-$h$ potentials we solve for ($ψ_k$) are directly related to the forward/backward potentials ($f_k$, $g_k$) used in other works (roughly, $ψ_k$ is a function of $log(f_k)$ and $log(g_k)$).
> >
> > The primary difference is, again, conceptual:
> > -   Our framework uses this machinery to solve for the **posterior path measure**, where anchors $\mu_k$ are part of the solution.
> > -   Generative SB works use it to find a transport map between **exogenous data distributions**.
> >
> > We thank the reviewer again for their valuable feedback, which helps us significantly improve the clarity and impact of our paper.

---

> > > ### Comment · Reviewer_tMy3 · 2025-11-26
> > >
> > > Dear authors,
> > >
> > > Thank you for your response and for clarifying my questions and concerns. I happily increased my score :)

---

### Note · Program_Chairs · 2026-01-17
**Submission Desk Rejected by Program Chairs**

The following references in this submission do not refer to real documents and/or have major errors in bibliographic information:

     Takeshi Mikami and Marc Thieullen. A variational problem related to optimal transport on wiener space. Stochastic Processes and their Applications, 116(12):1815-1835, 2006.